# A frontal transcallosal inhibition loop mediates interhemispheric balance in visuospatial processing

Yanjie Wang[1,2,4], Zhaonan Chen[1,2,4], Guofen Ma[1,2,4], Lizhao Wang[1,2], Yanmei Liu[1,2], Meiling Qin[3], Xiang Fei[3], Yifan Wu [1,2], Min Xu [3] ✉ & Siyu Zhang [1,2] ✉

Interhemispheric communication through the corpus callosum is required for both sensory and cognitive processes. Impaired transcallosal inhibition causing interhemispheric imbalance is believed to underlie visuospatial bias after frontoparietal cortical damage, but the synaptic circuits involved remain largely unknown. Here, we show that lesions in the mouse anterior cingulate area (ACA) cause severe visuospatial bias mediated by a transcallosal inhibition loop. In a visual-change-detection task, ACA callosal-projection neurons (CPNs) were more active with contralateral visual field changes than with ipsilateral changes. Unilateral CPN inactivation impaired contralateral change detection but improved ipsilateral detection by altering interhemispheric interaction through callosal projections. CPNs strongly activated contralateral parvalbumin-positive (PV+) neurons, and callosal-input-driven PV+ neurons preferentially inhibited ipsilateral CPNs, thus mediating transcallosal inhibition. Unilateral PV+ neuron activation caused a similar behavioral bias to contralateral CPN activation and ipsilateral CPN inactivation, and bilateral PV+ neuron activation eliminated this bias. Notably, restoring interhemispheric balance by activating contralesional PV+ neurons significantly improved contralesional detection in ACA-lesioned animals. Thus, a frontal transcallosal inhibition loop comprising CPNs and callosal-input-driven PV+ neurons mediates interhemispheric balance in visuospatial processing, and enhancing contralesional transcallosal inhibition restores interhemispheric balance while also reversing lesion-induced bias.

Animals actively process a large amount of environmental information at any given moment, and appropriate motor responses to relevant sensory stimuli are important for survival. Both lower-level sensory processes and higher-order cognitive functions require balanced interhemispheric interactions via the corpus callosum to transfer and integrate information between the brain's two hemispheres[1–4]. The callosal fibers mostly project to homotopic regions of the hemispheres (e.g., frontal lobe to frontal lobe, etc.) along an anterior-posterior axis[3]. Clinical observations of patients with callosal degradation suggested distinctions between posterior and anterior regions: the posterior

[1]Songjiang Research Institute, Shanghai Songjiang District Central Hospital, Shanghai Jiao Tong University, School of Medicine, Shanghai 200025, China. [2]Center for Brain Science of Shanghai Children's Medical Center, Department of Anatomy and Physiology, School of Medicine, Shanghai Jiao Tong University, Shanghai 200127, China. [3]Institute of Neuroscience, CAS Center for Excellence in Brain Science and Intelligence Technology, State Key Laboratory of Neuroscience, Chinese Academy of Sciences, Shanghai 200031, China. [4]These authors contributed equally: Yanjie Wang, Zhaonan Chen, Guofen Ma. ✉e-mail: mxu@ion.ac.cn; zhang_siyu@sjtu.edu.cn

callosal areas linking sensory regions of the two hemispheres were related to interhemispheric facilitation of binocular vision, while anterior callosal regions linking frontal and parietal regions were related to maintaining balanced interhemispheric inhibition in visuospatial processing[3,5,6].

In mammals, the function of the primary visual cortex (V1) is highly lateralized, with the left cerebral hemisphere processing visual information from the right visual field and the right cerebral hemisphere processing information from the left visual field[7]. The callosal projections allow the sensory inputs from both hemispheres to be integrated and selected for further higher-order cognitive processing and for a unified motor response[3,8–11]. Previous studies in cats have shown that V1 callosal projections mainly activate contralateral excitatory neurons and enhance the gain of visual response in relation to the retino-geniculo-cortical input[12–15].

Recent studies have shown that also in rodents V1 callosal projections contribute to visual information processing in multiple ways. First, V1 callosal projections drive the developmental elimination of specific cortical interneurons (chandelier cells) in the binocular zone to facilitate binocular vision[16]. Second, V1 callosal synapses cluster with non-callosal synapses with similar orientation preferences to enhance binocular visual responses[17]. Third, V1 callosal projections work in concert with thalamocortical inputs to enhance both monocular and binocular visual responses[18]. Finally, V1 CPNs enhance synchronous oscillations between hemispheres to facilitate visual novelty discrimination[19]. However, very little is known about which transcallosal synaptic circuits mediate interhemispheric inhibition in visuospatial processing (i.e., a demonstrated function of anterior callosal regions linking the parietal and frontal cortices in the two hemispheres[6]).

Clinical observations in stroke patients, as well as lesion studies in monkeys, have shown that unilateral damage in a dorsal frontoparietal network that provides top-down control of visual processing results in interhemispheric imbalance and leads to severe visuospatial bias towards the ipsilesional side, even leading to neglect of visual stimuli on the contralesional side[6,20–24]. Specifically, unilateral lesions were found to reduce ipsilesional network activity, and a contralesional network with relatively higher activity can lead to a visuospatial bias towards the ipsilesional side[6,25,26]. Impaired mutual transcallosal inhibition causing interhemispheric imbalance has been proposed as a pathophysiological mechanism underlying visuospatial bias after frontoparietal cortical damage[6,27,28].

Previous studies examining the medial and frontal cortices in the mouse visual network have established that the retrosplenial (RSP), posterior parietal (PTLp), and cingulate (ACA) cortices directly innervate visual cortices, an arrangement similar to the cortical areas known for the primate dorsal frontoparietal network[29–34]. These higher-level cortical areas provided top-down modulation of visual information processing, and modulate different aspects of visually-guided goal-directed behavior in mice[33,35–43].

Here, we examined the transcallosal synaptic circuits mediating the interhemispheric inhibition underlying unilateral cortical-damage-induced visuospatial bias in the mouse brain. Examining all of the medial and frontal cortices directly innervating the V1 revealed that unilateral lesion of the ACA induced the most severe detected visuospatial bias in a visual-change-detection task. We then examined the roles of the ACA callosal-projection neurons (CPNs) and their downstream contralateral local circuits that enable mutual transcallosal inhibition between the two hemispheres in a healthy brain. Combining cell-type-specific calcium imaging, optogenetic manipulation, and synaptic-circuit dissection, we found a transcallosal loop wherein ACA CPNs strongly activate contralateral PV+ neurons and callosal-input-driven PV+ ($PV+_{cal}$) neurons preferentially inhibit nearby CPNs within the same hemisphere, thus mediating transcallosal inhibition in visuospatial processing. Additionally, we show that lesion-induced visuospatial bias can be reversed by restoring interhemispheric balance through increased transcallosal inhibition in the contralesional hemisphere.

## Results

### ACA lesions result in prolonged impairment in detecting contralesional visual changes

To investigate which synaptic circuits mediate the transcallosal inhibition underlying unilateral cortical-damage-induced visuospatial bias, we first examined whether lesions of the medial and frontal cortices directly innervating the V1 in the mouse visual network induce visuospatial bias. We injected an adeno-associated virus (AAV-EF1α-DIO-taCasp3) into the RSP, PTLp, or ACA in one hemisphere of CaMKIIα-Cre mice to induce cell death of pyramidal neurons, and then trained these animals to perform a two-alternative unforced-choice (2AUC) change-detection task (Fig. 1a, adapted from Burgess et al.[44], see "Methods"). Given reports that a visual stimulus presented to a mouse's monocular field is processed in the contralateral hemisphere[7], we placed the two initial visual stimuli in the animal's left and right monocular field to facilitate the study of interhemispheric interaction.

In left-go trials, head-fixed mice were rewarded with water for selecting a changed visual stimulus (from initial "visual white noise" to the appearance of Go cue--drifting grating) on the left screen by turning a steering wheel under their front paws clockwise to bring the grating to the center of the middle screen (left-hit). In right-go trials, mice earned water by selecting a changed visual stimulus on the right screen with counterclockwise wheel turn (right-hit). In no-go trials, constant visual white noise was presented, and mice were punished with a timeout (2 s) for turning the wheel in any direction (false alarm [FA]; clockwise turn, left-FA; counterclockwise turn, right-FA).

In learning phase1, we trained mice with interleaved left-go and right-go blocks (Fig. 1b). Each block comprised 30 left-go or right-go trials with visual stimuli at 100% contrast. We found that a mouse's correct hit rate stabilized within two weeks, and control mice achieved similar left-hit and right-hit rate (Fig. 1c, Day10, left-hit, 94.4 ± 2.2%; right-hit, 92.6 ± 1.5%; $P = 0.52$, two-sided paired $t$-test). Unilateral ACA lesion caused failure to learn contralesional tasks (Fig. 1d, e). At the end of learning phase1, the contralesional hit rate is significantly lower than the ipsilesional hit rate (Day 12, $P = 1 \times 10^{-9}$, two-sided paired $t$-test). Conversely, unilateral RSP and PTLp lesions only slowed down the learning of contralesional tasks, while the contralesional hit rates caught up with the ipsilesional hit rates by the end of learning phase1 in both groups (Day12, RSP-lesioned, $P = 0.96$; PTLp-lesioned, $P = 0.36$, two-sided paired $t$-test; Fig. 1f,g, and Supplementary Fig. 1). These results indicate that ACA lesion induces more severe visuospatial bias than RSP and PTLp lesions, so we subsequently focused on the interhemispheric balance between the ACA in the two hemispheres.

After learning phase1, we introduced no-go trials to control mice and ACA-lesioned mice. Once their behavioral performance stabilized, we varied task difficulty by adjusting the contrast levels of visual stimuli (test phase, Fig. 1h). In contrast to the similar left and right performance in control mice (hit rates, $F_{side}(1,7) = 0.08$, $P_{side} = 0.79$; FA rates, $F_{side}(1,7) = 0.009$, $P_{side} = 0.93$, two-way repeated ANOVA), ACA lesion led to prolonged deficits of contralesional change detection at multiple contrast levels (hit rates, $F_{side}(1,10) = 124$, $P_{side} = 6 \times 10^{-7}$; FA rates, $F_{side}(1,10) = 8.94$, $P_{side} = 0.01$, two-way repeated ANOVA; Fig. 1i–j), indicating a severe visuospatial bias towards the ipsilesional side following ACA lesion.

### ACA CPNs exhibit selective visual-related activity with a contralateral bias

To assess the physiological contributions of callosal projections from the ACA to interhemispheric balance in visuospatial processing, we measured the task-related activity of ACA CPNs. The calcium indicator GCaMP6s was expressed in ACA CPNs by injecting retrograde AAV

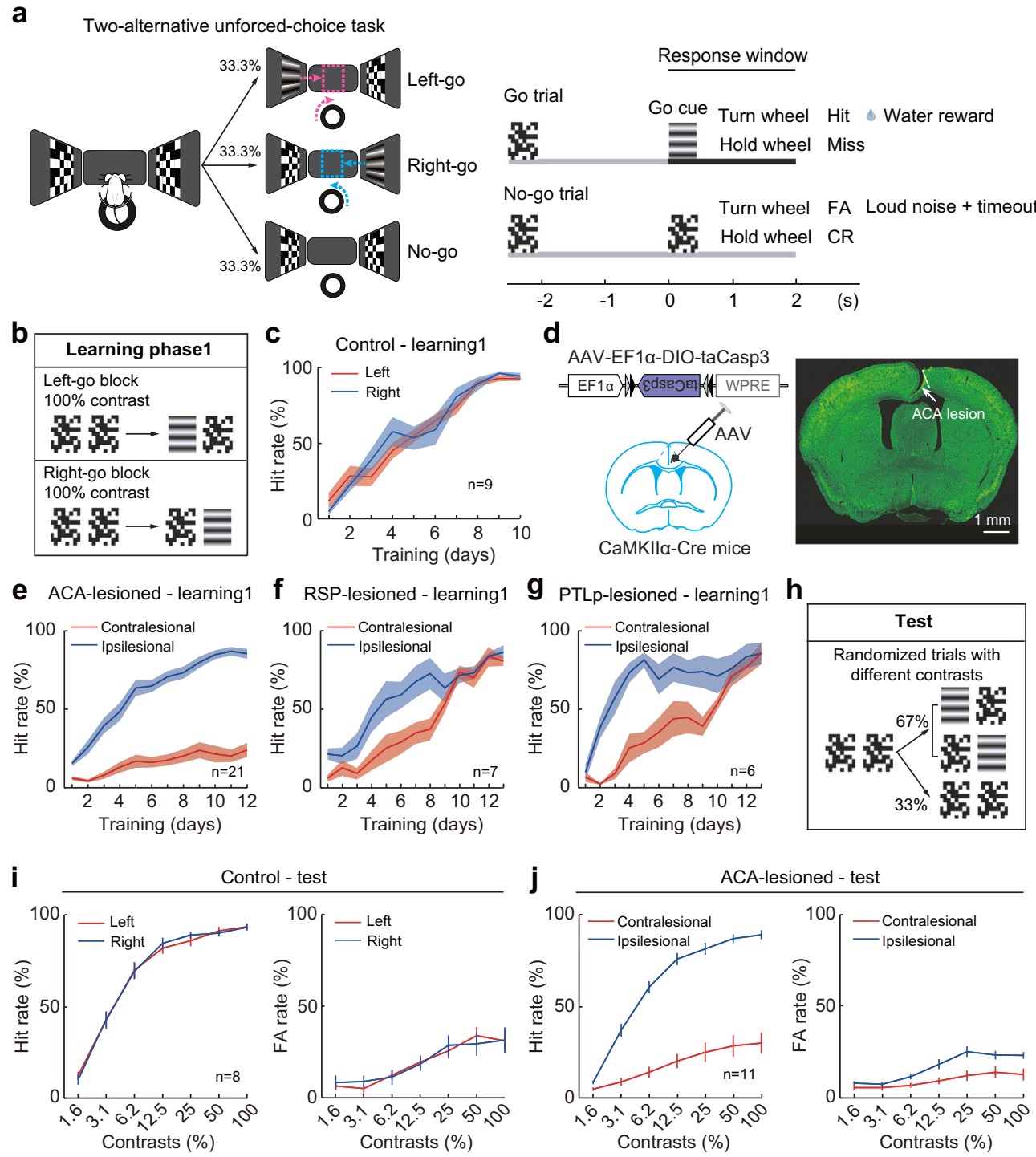

**Fig. 1 | Performance of ACA-lesioned mice in the 2AUC change-detection task.**
**a** Schematic for the behavioral tasks. FA false alarm, CR correct rejection.
**b** Schematic of the behavioral paradigms for learning phase1. **c** Performance of the
control group in learning phase1. The performance was not significantly different
between the left-go (red) and right-go (blue) trials ($n = 9$ mice, $F_{side}(1,8) = 0.07$,
$P_{side} = 0.8$, $F_{side*day}(9,72) = 1$, $P_{side*day} = 0.45$, two-way repeated ANOVA). Colored
shading, $\pm$ SEM. **d** Schematic diagram of the viral strategy for ACA lesion (left) and
fluorescence image of a coronal section showing an ACA lesion (right). Green,
fluorescent Nissl staining. $n = 21$ mice. **e** Similar to **c**, but for the performance of the
ACA-lesioned group. ACA-lesioned mice failed to learn the contralesional detection
task. The hit rates in contralesional trials were significantly lower than in ipsilesional
trials during learning1 ($n = 21$ mice, $F_{side}(1,20) = 161.66$, $P_{side} = 5 \times 10^{-11}$,

$F_{side*day}(11,220) = 19.85$, $P_{side*day} = 2 \times 10^{-27}$, two-way repeated ANOVA). **f, g** Similar to
**c**, but for performance of the RSP-lesioned and PTLp-lesioned groups. RSP and
PTLp lesions slowed down the learning of the contralesional change-detection task
(RSP-lesioned, $n = 7$ mice, $F_{side*day}(12,72) = 1.92$, $P_{side*day} = 0.046$; PTLp-lesioned,
$n = 6$ mice, $F_{side*day}(12,60) = 5.44$, $P_{side*day} = 4 \times 10^{-6}$, two-way repeated ANOVA).
**h**, Schematic of the behavioral paradigms for the test phase. **i** Performance of the
control group in the test phase ($n = 8$ mice). Red, left-go trials; blue, right-go trials.
At each contrast level, left-hit + left-miss = 100%; right-hit + right-miss = 100%; left-
FA + right-FA + CR = 100%. **j** Similar to **i**, but for performance of the ACA-lesioned
group ($n = 11$ mice). Red, contralesional trials; blue, ipsilesional trials. Data are
presented as the mean $\pm$ SEM. See Supplementary Data 2 for ANOVA parameters.
Source data are provided as a Source Data file.

expressing Cre (Retro-AAV-hSyn-Cre) into the ACA of one hemisphere and injecting Cre-inducible AAV expressing GCaMP6s (AAV-EF1α-DIO-GCaMP6s) into the opposite ACA in wild-type mice (Fig. 2a). Imaging was performed through a gradient-index (GRIN) lens coupled to a fluorescent microscope during the 2AUC change-detection task (Supplementary Fig. 2). To acquire sufficiently replicated data to support inferential statistical analysis, we reduced the number of contrast levels from seven to two in subsequent experiments (100% and 12.5%). Two-way repeated-measures ANOVA of the hit rate, FA rate, and reaction time revealed a significant effect for the contrast level, no significant effect for the left/right spatial location, and no significant interaction between contrast condition and spatial location. As expected, we observed increased difficulty of change detection with decreased contrast level: for trials with 12.5% contrast, mice showed significantly lower hit rates, lower FA rates, and longer reaction time than for trials with 100% contrast (Supplementary Fig. 3).

We detected clear task-related neural activities that were time-locked to the visual stimuli, to wheel rotation, and to trial outcomes in 77% of ACA CPNs (see Methods, Fig. 2b–d, and Supplementary Fig. 4), findings consistent with previous studies reporting that the ACA encodes both sensory- and motor-related signals[36–38,45]. Although the behavioral performance differed significantly between visual stimuli at

100% and 12.5% contrast, we observed that fewer than 6% of ACA CPNs showed a significant difference in response to high- or low-contrast trials during the visuomotor period (i.e., from the appearance of the drifting grating to the end of wheel rotation; see Methods). Given the small population of ACA CPNs exhibiting a contrast-dependent response, in subsequent analyses we grouped the 100% and 12.5% contrast trials together. During the visuomotor period, 50% of ACA CPNs responded differentially in the contralateral-hit (visual change on the side opposite to recorded CPNs) vs. the ipsilateral-hit trials, with significantly more contralateral-preferring CPNs than ipsilateral-preferring CPNs ($P = 8 \times 10^{-5}$, chi-square test, Fig. 2e). Although the single neuron activity was heterogeneous, the averaged responses of ACA CPNs were significantly stronger in the contralateral-hit trials than in the ipsilateral-hit trials during the visuomotor period, indicating that the overall strength of callosal inputs is stronger with contralateral visual change than with ipsilateral visual change (Fig. 2f).

We further investigated the neuronal encoding of visual stimulus and motor action from the CPN responses in miss and FA trials. At the population level, both contralateral-preferring and ipsilateral-preferring CPNs showed selectivity for visual stimuli (contralateral vs. ipsilateral drifting gratings) in miss trials (visual change without motor action), indicating that they encoded the properties of visual

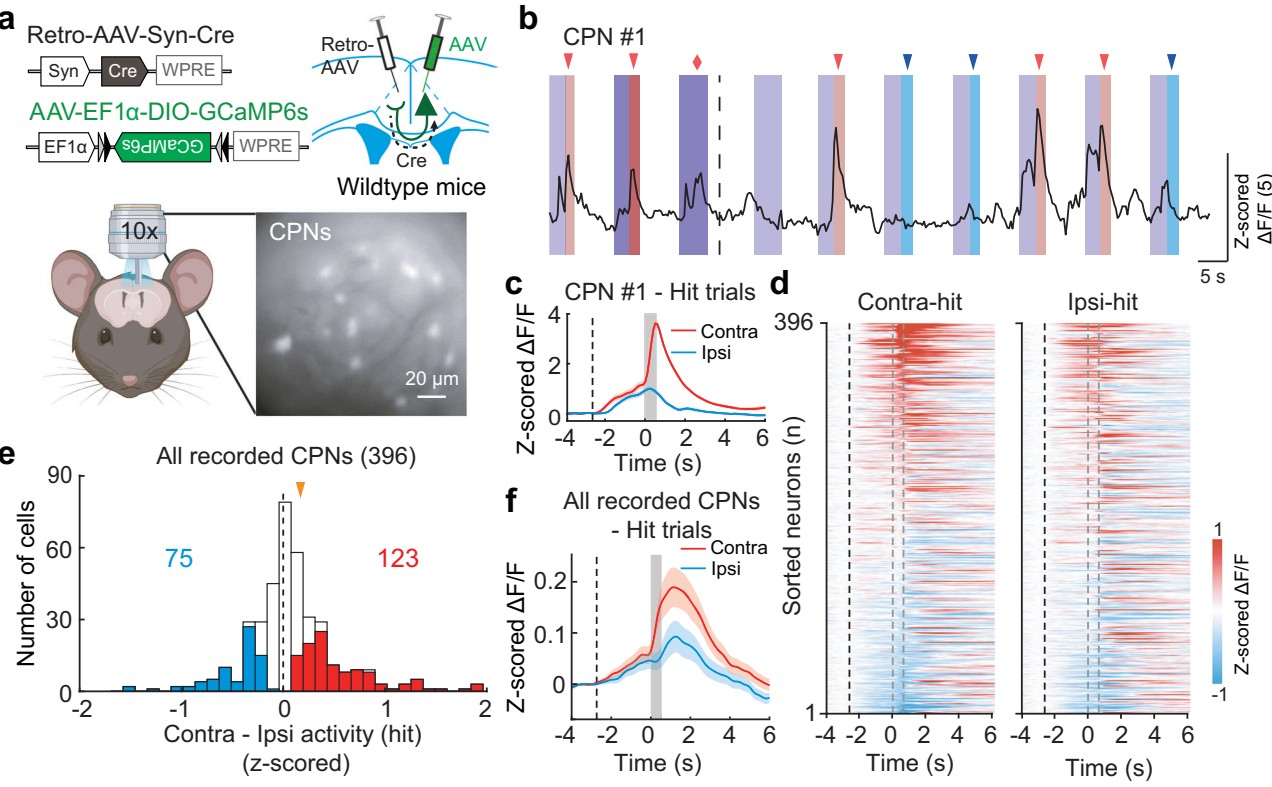

**Fig. 2 | ACA CPNs show a contralateral bias in the 2AUC change-detection task.** **a** Schematic diagrams of the viral strategy (top) and calcium imaging of CPNs (bottom). **b** Raw fluorescence trace of an example CPN. Same behavior paradigm as Fig. 1h, containing the trials at 100% and 12.5% contrast levels. Dark color, high contrast; light color, low contrast. Purple shading, visual white noise; red shadings, contralateral gratings; blue shading, ipsilateral gratings. Red and blue arrowheads, end of wheel rotation in the contralateral and ipsilateral-hit trials, respectively. Red diamond, end of contralateral wheel rotation in the FA trial. Dashed line, end of timeout in the FA trial. **c** Averaged ΔF/F (z-scored) of an example CPN in hit trials. Colored shading, ±SEM. Black dashed line, the start of visual white noise. Gray shading, window of the visuomotor period (from the start of drifting grating to the end of wheel rotation). **d** Color-coded averaged ΔF/F (z-scored) of all recorded CPNs in contralateral-hit (left) and ipsilateral-hit (right) trials ($n$ = 8 mice, 8 sessions, 396 neurons). Each row shows the average activity of one neuron, sorted by peak

response during the visuomotor period in contralateral-hit trials. Black dashed line, the start of visual white noise. The left and right gray dashed lines present the start of drifting grating and the end of wheel rotation, respectively. **e** Distribution of the difference in peak responses of z-scored ΔF/F during the visuomotor period between the contralateral-hit and ipsilateral-hit trials for the imaged CPNs. Blue, ipsilateral-preferring cells; red, contralateral-preferring cells. Orange arrowhead, median. There were significantly more contralateral-preferring CPNs than ipsilateral-preferring CPNs ($P = 8 \times 10^{-5}$, two-sided chi-square test). **f** Similar to **c**, but for all recorded CPNs. The averaged responses of CPNs were significantly stronger in the contralateral-hit trials than in the ipsilateral-hit trials during the visuomotor period ($P = 0.001$, two-sided paired $t$-test). See also Supplementary Fig. 4 and "Methods". Source data are provided as a Source Data file. Created with BioRender.com.

stimulus (Fig. 3 and Supplementary Fig. 5). The differences in neuronal responses to contralateral versus ipsilateral visual change were significantly greater in hit trials compared to miss trials (contralateral-preferring CPNs, $P = 4.5 \times 10^{-6}$; ipsilateral-preferring CPNs, $P = 1.2 \times 10^{-4}$, paired $t$-test), likely owing to differences in behavioral states such as task engagement and/or attention levels. Additionally, none of these neuron populations showed selectivity for the direction of motor actions (clockwise vs. counterclockwise wheel rotation) in FA trials (motor action without visual change), indicating that they did not encode the properties of motor action. Together, these results demonstrate that ACA CPNs exhibit selective visual-related activity with a contralateral bias.

## ACA CPNs mediate interhemispheric inhibition in visuospatial processing via callosal projections

To directly test the functional impact of ACA CPNs in the 2AUC change-detection task, we both optogenetically inactivated and activated their activities. For unilateral inactivation of ACA CPNs, we injected a retrograde AAV expressing Cre (Retro-AAV-hSyn-Cre) into the ACA of one hemisphere and a Cre-inducible AAV expressing halorhodopsin (NpHR) (AAV-EF1α-DIO-NpHR-EYFP) into the opposite ACA in wild-type mice (Fig. 4a and Supplementary Figs. 2, and 6a–d). Compared with the control (laser-off) trials, unilateral inactivation of ACA CPNs impaired contralateral performance, with significantly decreased hit rates and increased FA rates (Fig. 4b, c).

We observed similar CPN-inactivation induced behavioral bias at the 100% and 12.5% contrast levels, consistent with the contrast-independent activity of CPNs (Supplementary Data 1 and 2). Interestingly, we noted that unilateral inactivation of the ACA CPNs significantly improved ipsilateral performance (i.e., increased hit rates and decreased FA rates; right panels, Fig. 4c). Further, unilateral inactivation of the ACA CPNs at low contrast caused a significant increase in reaction time in contralateral-hit trials alongside a simultaneous significant decrease in reaction time in ipsilateral-hit trials (Fig. 4d). In contrast, ChR2-mediated unilateral activation of ACA CPNs resulted in significant improvements in contralateral performance (i.e., increased hit rates, decreased FA rates, and decreased reaction time) while simultaneously causing impairment in ipsilateral performance (i.e., decreased hit rates, increased FA rates, and increased reaction time) (Fig. 4e, f, and Supplementary Fig. 5e–h), supporting the notion that the hemisphere with higher activity biases the visuospatial processing towards its contralateral side. In control mice injected with AAV-CaMKIIα-EYFP, laser had no effect (Supplementary Fig. 7). Together, these results indicated that ACA CPNs mediate competitive interactions between the brain's two hemispheres in visuospatial processing.

Cortical pyramidal neurons typically send axon collaterals to multiple target areas[46–48]. To determine whether the behavioral bias induced by inactivation of ACA CPNs is mediated by altering interhemispheric interaction via their callosal projections, we performed

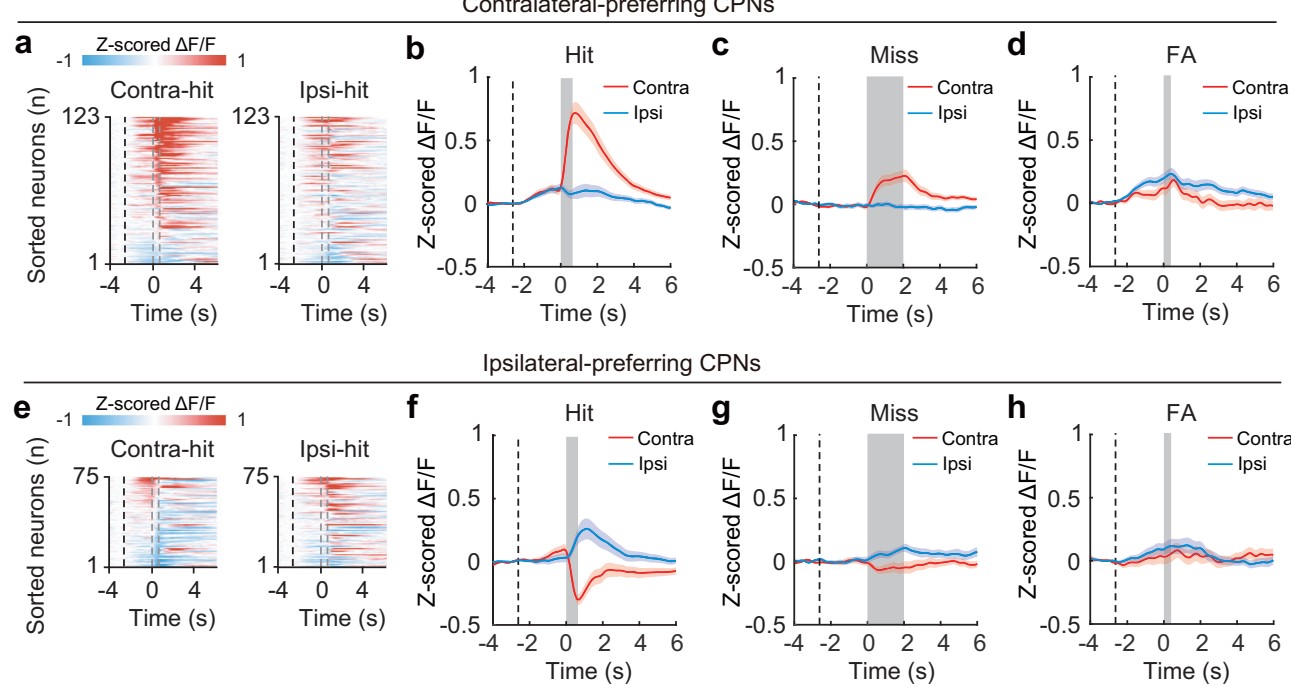

**Fig. 3 | ACA CPNs show selective visual-related activity in the 2AUC change-detection task. a** Color-coded averaged $\Delta F/F$ (z-scored) of the contralateral-preferring CPNs in contralateral-hit (left) and ipsilateral-hit (right) trials ($n = 8$ mice, 8 sessions, 123 neurons). Each row shows the average activity of one neuron, sorted by peak response during the visuomotor period in contralateral-hit trials. Black dashed line, the start of visual white noise. The left and right gray dashed lines present the start of drifting grating and the end of wheel rotation, respectively. **b**–**d** Averaged $\Delta F/F$ (z-scored) of contralateral-preferring CPN in different types of trials. Black dashed line, the start of visual white noise. Colored shading, ± SEM. **b** Activity in hit trials. Gray shading, window of the visuomotor period. The activity of contralateral-preferring CPNs was significantly higher in the contralateral-hit trials than in the ipsilateral-hit trials during the visuomotor period ($P = 1 \times 10^{-17}$, two-sided paired $t$-test). **c** Activity in miss trials. Gray shading, window during the presentation of drifting grating. The activity was significantly higher in the

contralateral-miss trials than in the ipsilateral-miss trials during the presentation of drifting grating ($P = 3 \times 10^{-8}$, two-sided paired $t$-test). **d** Activity in FA trials. Gray shading, window during the wheel rotation. The activity did not differ between the contralateral-FA trials and ipsilateral-FA trials during the wheel rotation ($P = 0.14$, two-sided paired $t$-test). **e**–**h**, Similar to **a**-**d**, but for ipsilateral-preferring CPNs ($n = 75$ neurons). The activity of ipsilateral-preferring CPNs was significantly lower in the contralateral-hit trials than in the ipsilateral-hit trials during the visuomotor period ($P = 8 \times 10^{-17}$, two-sided paired $t$-test). Similarly, the activity of ipsilateral-preferring CPNs was significantly lower in the contralateral-miss trials than in the ipsilateral-miss trials during the presentation of drifting grating ($P = 6 \times 10^{-8}$, two-sided paired $t$-test). However, there was no significant activity difference between the contralateral-FA trials and the ipsilateral-FA trials during the wheel rotation ($P = 0.10$, two-sided paired $t$-test). See also Supplementary Fig. 5 and "Methods". Source data are provided as a Source Data file.

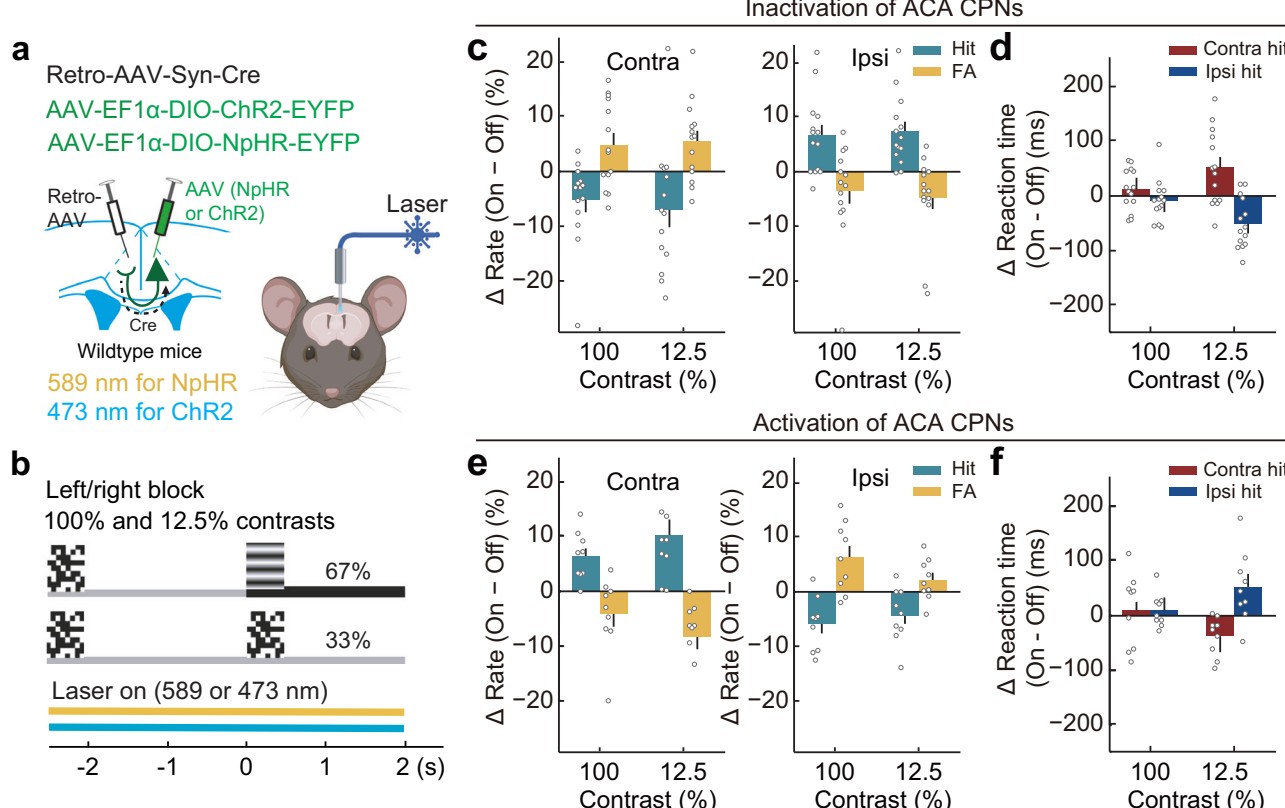

**Fig. 4 | Manipulation of ACA CPN activity has opposite effects on contralateral and ipsilateral performance. a**, **b**, Schematic diagrams of the viral strategy (**a**) and behavioral paradigm (**b**) for optogenetic manipulation of CPNs. Yellow laser, inactivation; blue laser, activation. **c** Effect of unilateral CPN inactivation on visual-change detection (*n* = 14 mice). Left, changes in contralateral performance. Right, changes in ipsilateral performance. Inactivation of CPNs (589 nm, 3–8 mW, constant light) significantly impaired contralateral detection (decreased hit rates, green bars, $F_{laser}(1,13) = 30.04$, $P_{laser} = 1 \times 10^{-4}$; increased FA rates, yellow bars, $F_{laser}(1,13) = 9.85$, $P_{laser} = 0.008$; two-way repeated ANOVA) and improved ipsilateral detection (increased hit rates, green bars, $F_{laser}(1,13) = 40.11$, $P_{laser} = 3 \times 10^{-5}$; decreased FA rates, yellow bars, $F_{laser}(1,13) = 8.27$, $P_{laser} = 0.01$; two-way repeated ANOVA). **d** Effect of unilateral CPN inactivation on reaction times in hit trials (*n* = 14 mice). CPN inactivation significantly increased contralateral reaction time and decreased ipsilateral reaction time at the 12.5% contrast level (contralateral reaction

time, $F_{laser}(1,13) = 7.17$, $P_{laser} = 0.02$; ipsilateral reaction time, $F_{laser}(1,13) = 17.255$, $P_{laser} = 0.001$; one-way repeated ANOVA). **e**, **f** Similar to **c**, **d**, but for optogenetic activation of CPNs (*n* = 9 mice). **e** Activation of CPNs (blue laser, 473 nm, 1–3 mW, 10 Hz, 5 ms per pulse) significantly improved contralateral detection (increased hit rates, $F_{laser}(1,8) = 18.07$, $P_{laser} = 0.003$; decreased FA rates, $F_{laser}(1,8) = 14.37$, $P_{laser} = 0.005$; two-way repeated ANOVA) and impaired ipsilateral detection (decreased hit rates, $F_{laser}(1,8) = 15.49$, $P_{laser} = 0.004$; increased FA rates, $F_{laser}(1,13) = 14.34$, $P_{laser} = 0.005$; two-way repeated ANOVA). **f** CPN activation significantly decreased contralateral reaction time and increased ipsilateral reaction time at the 12.5% contrast level (contralateral reaction time, $F_{laser}(1,8) = 10.46$, $P_{laser} = 0.01$; ipsilateral reaction time, $F_{laser}(1,8) = 5.71$, $P_{laser} = 0.04$; one-way repeated ANOVA). Data are presented as the mean ± SEM. See Supplementary Data 2 for ANOVA parameters. Source data are provided as a Source Data file. Created with BioRender.com.

optogenetic inactivation targeting the callosal-projection axon terminals of CPNs in the opposite ACA using the inhibitory opsin eOPN3, which can effectively suppress synaptic transmission through the $G_{i/o}$ signaling pathway[49]. To examine the temporal dynamics of eOPN3-induced inactivation of CPN callosal-projection axons, we made whole-cell recordings in ACA slices while optogenetically activating callosal projection axons both during and after eOPN3 activation. To express ChrimsonR (for activation of axons using red light) and eOPN3 (for inactivation of axons using green light) in ACA CPNs, we injected a retrograde AAV expressing Cre (Retro-AAV-hSyn-Cre) into the ACA of one hemisphere and mixed Cre-inducible AAVs expressing ChrimsonR and eOPN3 (AAV-hSyn-DIO-ChrimsonR-EYFP and AAV-hSyn-SIO-eOPN3-mScarlet) into the opposite ACA in wild-type mice (Fig. 5a and Supplementary Fig. 8). During the green light stimulation, which lasted for 4 s, eOPN3 efficiently inhibited the transmitter release from callosal-projection axons, and the transmitter release fully recovered at 8 s after the green light stimulation (Fig. 5b).

Based on the temporal dynamics of eOPN3-induced axon inactivation, we examined the behavioral bias induced by optogenetic inactivation of CPN callosal-projection axons using

interleaved laser-on and laser-off trials with a 10-s inter-trial interval (Fig. 5c, d). Compared to laser-off trials, inactivation of CPN callosal-projection axon terminals in the opposite ACA resulted in significant improvements in contralateral performance (i.e., increased hit rates, decreased FA rates, and decreased reaction time) while simultaneously causing impairment in ipsilateral performance (i.e., decreased hit rates, increased FA rates, and increased reaction time, Fig. 5e, f). The magnitude of the behavioral change resulting from the inactivation of callosal-projection axon terminals from the opposite ACA is comparable to that caused by the inactivation of CPNs in the opposite ACA (Supplementary Data 1), indicating that callosal projection is a major pathway mediating the behavioral bias induced by CPN inactivation. Further, the behavioral effects induced by the inactivation of callosal-projection axon terminals from the opposite ACA are similar to those caused by the activation of CPNs in the same hemisphere as the inactivated callosal-projection axon terminals (Supplementary Data 1), indicating a net effect of disinhibition of CPNs following inactivation of callosal projections. Together, these results demonstrated that ACA CPNs mediate interhemispheric inhibition in visuospatial processing via their callosal projections.

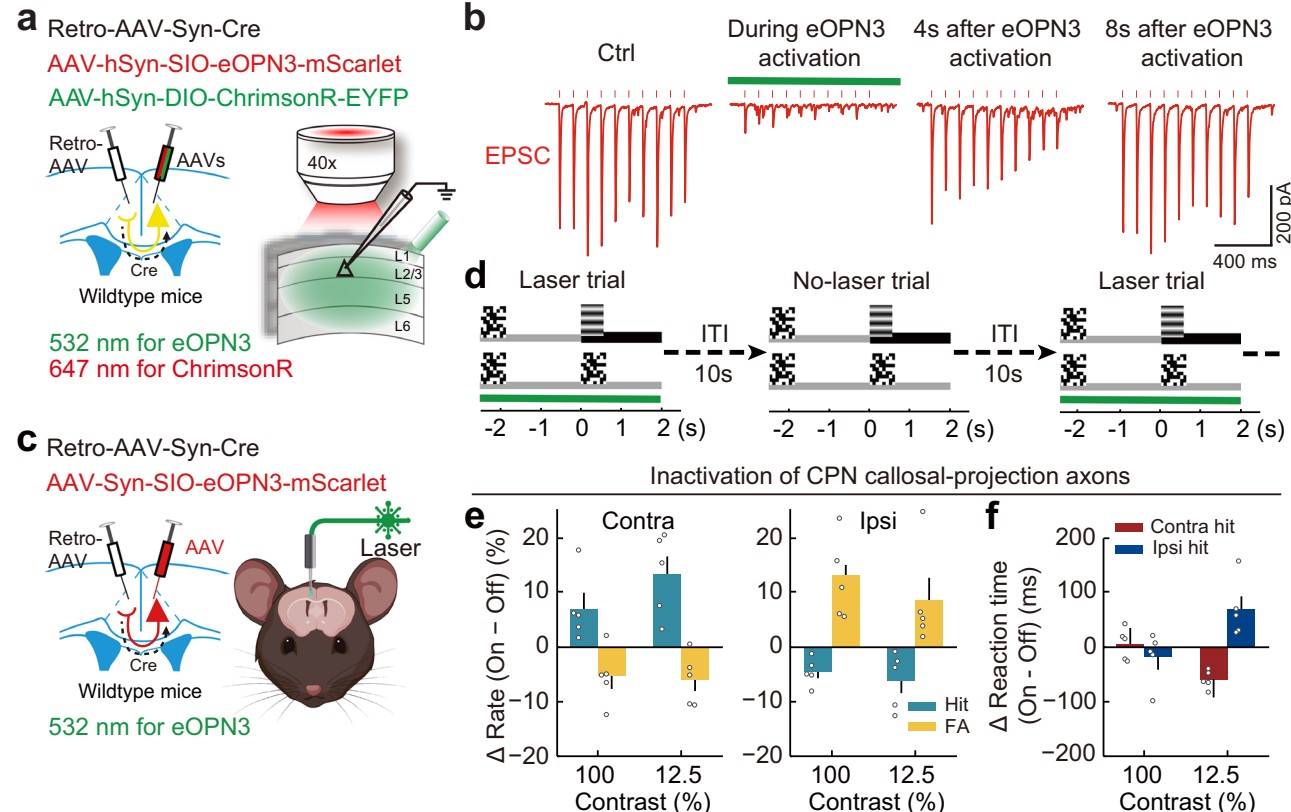

**Fig. 5 | Inhibiting CPN callosal-projection axons induced similar behavioral bias to inhibiting contralateral CPNs. a** Schematic of the slice experiment for examining the temporal dynamics of eOPN3-induced inactivation of callosal-projection axons. **b** EPSCs induced by activation of ChrimsonR at the axon terminals of contralateral CPNs in an example layer 2/3 pyramidal neuron. Red dots, 5-ms red light pulses (647 nm, 10 Hz, 3 mW). Control EPSCs were recorded before eOPN3 activation. eOPN3 activation lasted for 4 s (green laser, 532 nm, constant light, 5 mW), and the red light pulses were delivered 2 s after the green light start. Red light pulses were delivered again at 4 s and 8 s after the end of green light stimulation. **c** Schematic diagrams of the viral strategy (left) and optogenetic inactivation of callosal-projection axons (right) in behaving mice. **d** Schematic of the behavior paradigm for optogenetic inactivation of callosal-projection axons. The laser-on trials and laser-off trials were interleaved with a 10-s inter-trial interval. **e** Effect of CPN callosal-projection axon inactivation on visual-change detection (*n* = 5 mice). Left, changes in contralateral performance. Right, changes in ipsilateral

performance. Inhibiting CPN callosal-projection axons (green laser, 532 nm, 5 mW, constant light) significantly improved contralateral detection (increased hit rates, green bars, $F_{laser}(1,4) = 50.82$, $P_{laser} = 0.002$; decreased FA rates, yellow bars, $F_{laser}(1,4) = 9.05$, $P_{laser} = 0.04$; two-way repeated ANOVA) and impaired ipsilateral detection (decreased hit rates, green bars, $F_{laser}(1,4) = 41.96$, $P_{laser} = 0.003$; increased FA rates, yellow bars, $F_{laser}(1,4) = 8.14$, $P_{laser} = 0.046$; two-way repeated ANOVA). **f** Effect of CPN callosal-projection axon inactivation on reaction times in hit trials (*n* = 5 mice). Inhibiting CPN callosal-projection axons significantly decreased the reaction time in contralateral-hit trials (red bar) and increased the reaction time in ipsilateral-hit trials (blue bar) at 12.5% contrast (contralateral reaction time, $F_{laser}(1,4) = 62.53$, $P_{laser} = 0.001$; ipsilateral reaction time, $F_{laser}(1,4) = 8.39$, $P_{laser} = 0.04$; one-way repeated ANOVA). Data are presented as the mean ± SEM. See Supplementary Data 2 for ANOVA parameters. Source data are provided as a Source Data file. Created with BioRender.com.

## Four major types of ACA neurons are directly innervated by the contralateral ACA

Previous studies have proposed that lesion-induced visuospatial bias can be caused by imbalanced transcallosal inhibition[6,27,28,50]. Given that transcallosal connections are formed by axons of CPNs—which are known to release the excitatory neurotransmitter glutamate[51,52]—it seems plausible that transcallosal inhibition between the ACA in each hemisphere could be mediated by local inhibitory neurons activated by callosal projections from contralateral CPNs. Parvalbumin-positive (PV+), somatostatin-positive (SST+), and vasoactive intestinal peptide-positive (VIP+) neurons are three major non-overlapping classes of cortical GABAergic interneurons[53,54]. To dissect the synaptic circuits mediating transcallosal inhibition, we conducted rabies-virus-mediated retrograde trans-monosynaptic tracing (RV-tracing)[55] on both excitatory pyramidal neurons (CaMKIIα+) and inhibitory GABAergic interneurons (PV+, SST+, and VIP+ neurons) in the ACA. Avian-specific retroviral receptor (TVA), red fluorescent protein (mCherry), and rabies glycoprotein (RG) were expressed in ACA neurons by injecting Cre-inducible AAVs into the ACA of CaMKIIα-, PV-, SST-, and VIP-Cre mice. Two weeks later, we injected a modified RV-

expressing enhanced green fluorescent protein (EGFP) (RV-ΔG-EGFP +EnvA), which only infects cells expressing TVA and requires RG to spread retrogradely to presynaptic cells (Fig. 6a).

After histological sectioning and fluorescence imaging, we used our previously reported pipeline[30,56] to analyze the digitized brain images. Briefly, the rotation module was used to rotate the 3D Allen Mouse Brain Atlas[57] to mimic the aberrant sectioning angle of the experimental brain. We next used the registration module to align the digitized brain images to the rotated 3D reference altas. In each digitized brain image, the position of manually identified RV-labeled neurons was recorded using the detection module. After detection and registration, the quantification/visualization module was used to quantify the RV-labeled neurons across the whole brain and to project these neurons onto the 3D reference atlas for visualization.

Across brain samples, the starter cells (expressing both mCherry and EGFP) were distributed throughout the cortical layers over large portions of the ACA (Fig. 6b). Around 80% of the starter cells were located in the ACA, with a few cells spreading into the adjacent secondary motor cortex (MOs, Fig. 6c). Reconstructing

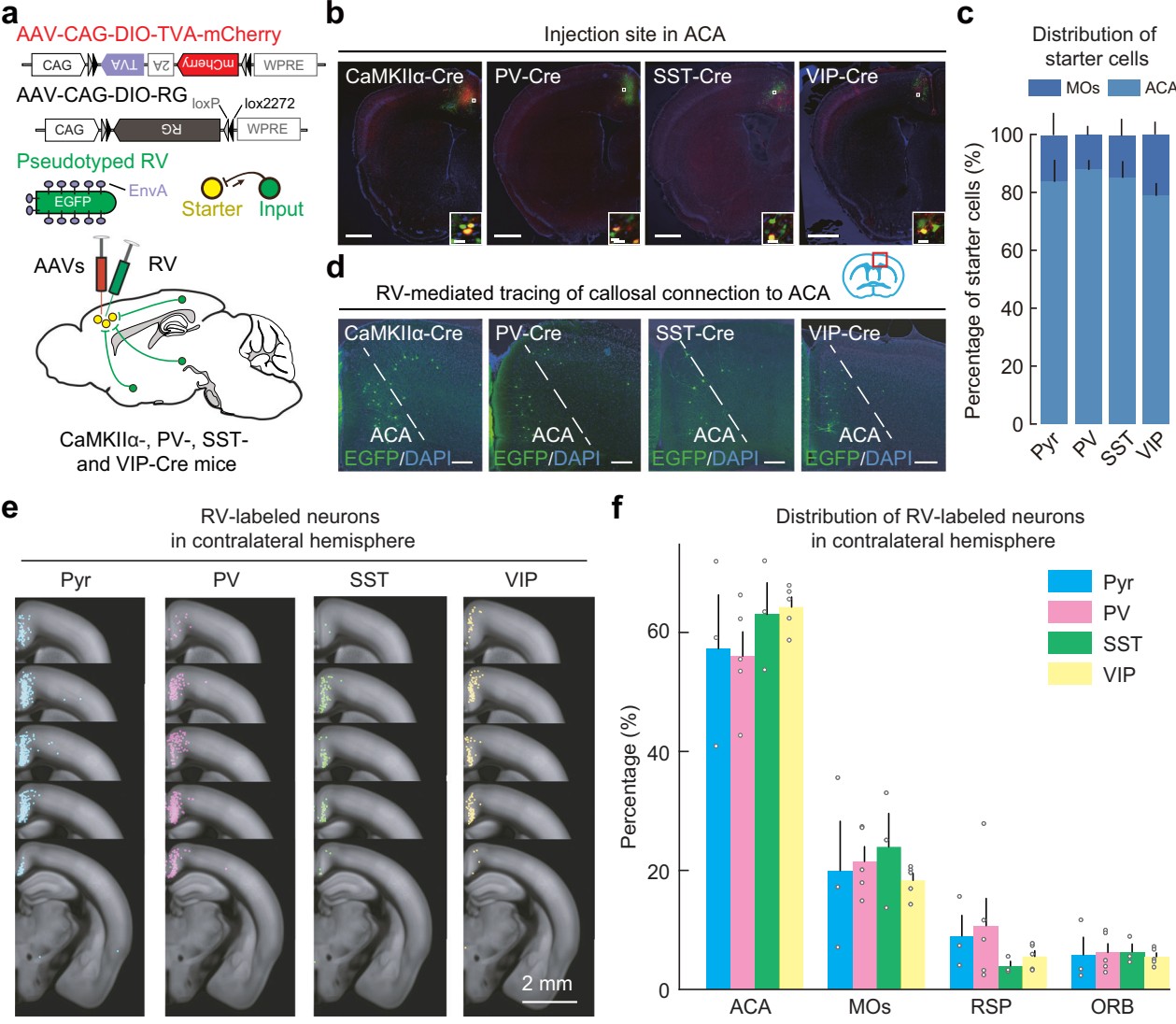

**Fig. 6 | Direct interhemispheric connections to different types of ACA neurons detected by RV-mediated trans-synaptic retrograde tracing. a** Viral vectors and injection procedure for RV-mediated trans-synaptic retrograde tracing from the indicated types of ACA neurons. **b** Viral injection sites in the ACA of CaMKIIα-, PV-, SST-, and VIP-Cre mice (scale bar, 1 mm). Inset, enlarged view of the region in the white box showing starter cells for RV tracing (yellow, scale bar, 20 μm). Green, EGFP; red, mCherry; blue, DAPI. Pyr, *n* = 3 mice; PV, *n* = 5; SST, *n* = 3; VIP, *n* = 5. **c** Distribution of starter cells in each group (Pyr, *n* = 3 mice; PV, *n* = 5; SST, *n* = 3; VIP,

*n* = 5). **d** RV-labeled neurons in the contralateral ACA (red box in coronal diagram) of CaMKIIα-, PV-, SST-, and VIP-Cre mice (scale bar, 200 μm). Pyr, *n* = 3 mice; PV, *n* = 5; SST, *n* = 3; VIP, *n* = 5. **e** Detected RV-labeled neurons in the contralateral hemisphere in each group. Dots, detected neurons. **f** Percentages of RV-labeled interhemispheric presynaptic neurons in each region among all the RV-labeled neurons in the contralateral hemisphere (Pyr, *n* = 3 mice; PV, *n* = 5; SST, *n* = 3; VIP, *n* = 5). Included are areas with >2% labeling. Data are presented as the mean ± SEM. Source data are provided as a Source Data file.

and superimposing the locations of RV-labeled presynaptic neurons (expressing EGFP only) in the contralateral hemisphere in CaMKIIα-, PV-, SST-, and VIP-Cre mice revealed extensive overlap in their spatial distributions (Fig. 6d,e). Given that the number of labeled neurons varied across brains, and aiming to assess an equally weighted population average for each brain, we quantified the input from each contralateral region by dividing the number of labeled neurons found in that region by the total number of labeled neurons detected in the entire contralateral hemisphere. More than 97% of the RV-labeled presynaptic neurons in the contralateral hemisphere were found in the cortical areas, in which ~60% of the presynaptic neurons were found in the contralateral ACA (Fig. 6f), indicating the most extensive callosal inputs to ACA neurons were from the contralateral ACA. These results demonstrate that both excitatory neurons and three major types of inhibitory neurons in the ACA are directly innervated by callosal inputs from the contralateral ACA.

## CPNs and PV+_cal neurons form a transcallosal inhibition loop that mediates interhemispheric balance

RV tracing is a powerful tool for visualizing the distribution of inputs to specific neuronal populations but lacks the information about the synaptic properties of labeled connections. To further investigate the neuronal mechanisms of transcallosal inhibition, we made whole-cell recordings in ACA slices while optogenetically activating callosal inputs. We initially compared the responses of CPNs to those of other pyramidal neurons. CPNs were identified by injecting a retrograde AAV expressing Cre (Retro-AAV-hSyn-Cre) in the opposite ACA of loxP-flanked tdTomato reporter mice (Fig. 7a). We detected both excitatory and inhibitory postsynaptic currents (EPSCs and IPSCs) in pyramidal neurons. The EPSCs in pyramidal neurons showed short onset latencies ($3.2 \pm 0.1$ ms, mean ± SEM), suggesting monosynaptic excitatory inputs from callosal projections. However, the IPSCs showed significantly longer latencies ($5.3 \pm 0.1$ ms, $P = 2 \times 10^{-27}$, Wilcoxon signed-rank test; mean ± SEM), suggesting disynaptic inhibition from local

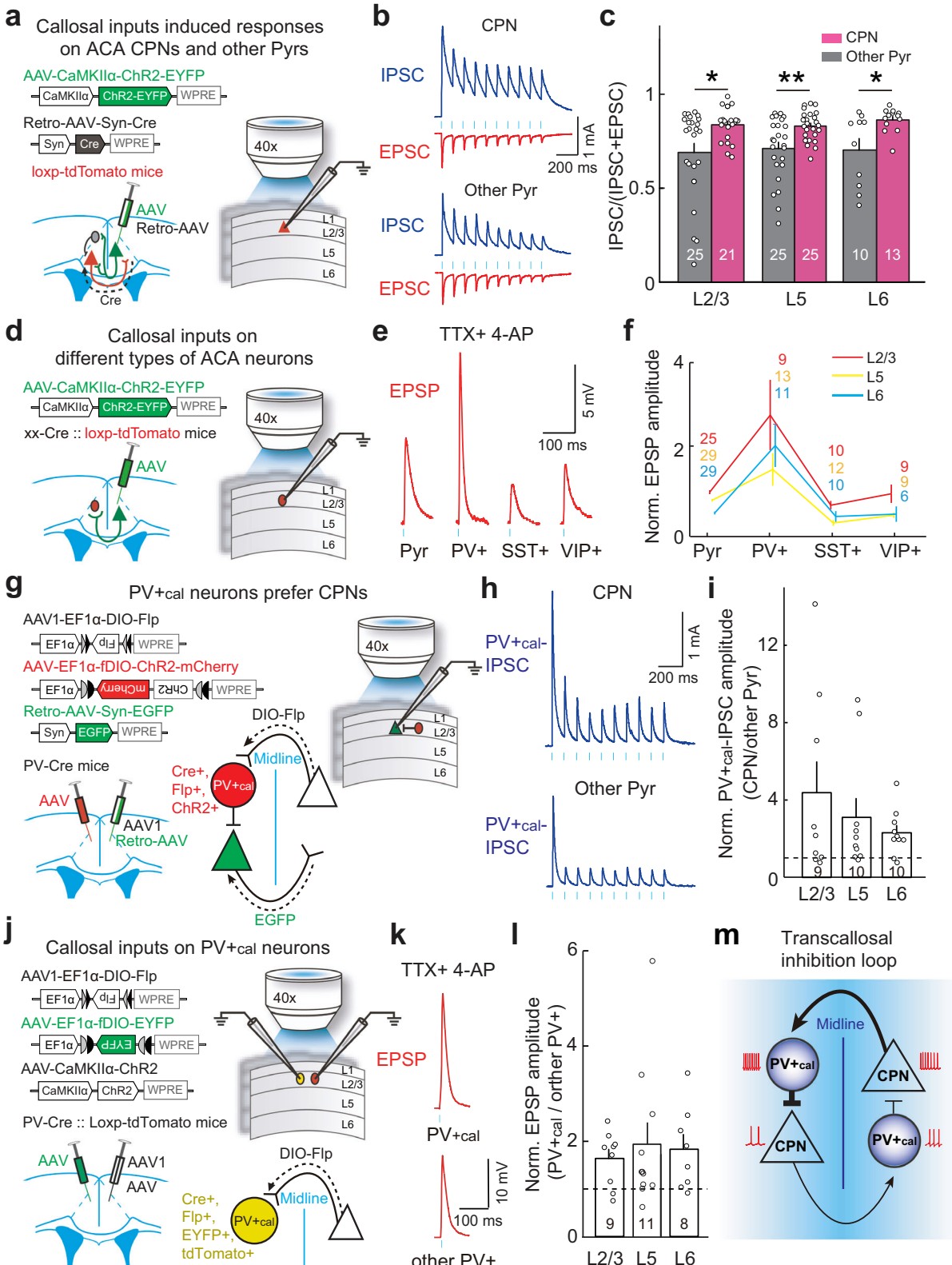

GABAergic interneurons. Notably, callosal inputs induced significantly stronger inhibition of CPNs than of the other pyramidal neurons examined across different layers (Fig. 7b, c), supporting the notion that CPNs received more intense transcallosal inhibition.

We next investigated the potential contribution of local GABAergic interneurons to the transcallosal inhibition of CPNs. First, we measured the input patterns of callosal projections on each class of

cortical interneurons, which were identified by breeding PV-, SST-, or VIP-Cre mice with loxP-flanked tdTomato reporter mice (Fig. 7d). After local neuron spiking was blocked with TTX[58], activation of callosal inputs evoked monosynaptic excitatory postsynaptic potentials (EPSPs) in all three examined interneuron classes (Fig. 7e), findings consistent with our RV-tracing results. Notably, the strength of callosal inputs was found to be comparable between SST+ and VIP+ neurons

**Fig. 7 | A transcallosal inhibition loop formed by CPNs and PV+$_{cal}$ neurons.**
**a** Schematic of the slice experiment, with whole-cell recording of ACA Pyrs and optogenetic activation of callosal inputs. **b** EPSCs and IPSCs in an example L2/3 CPN and a nearby Pyr. Blue dots, 5-ms blue-light pulses (10 Hz). **c** Contribution of inhibitory inputs to total synaptic inputs, measured as IPSC$_{charge}$/(IPSC$_{charge}$ + EPSC$_{charge}$). Callosal inputs drive stronger inhibition in CPNs than in other Pyrs. *$P < 0.05$; **$P < 0.01$; two-sided Wilcoxon rank-sum test. **d** Schematic of the slice experiment to measure callosal inputs-induced EPSPs in ACA neurons. **e** Example monosynaptic EPSPs recorded from Pyr, PV+, SST+, and VIP+ neurons in L2/3 with TTX and 4-AP treatment to block local neuron spiking. **f** Normalized EPSP amplitudes in different cell types (normalized to L2/3 Pyrs). Most neuron types in L2/3 received stronger callosal inputs as compared to the corresponding neuron types in L5 and L6 ($P < 0.05$, two-sided Wilcoxon rank-sum test), excluding the similar input strength for L2/3 PV+ and L6 PV+ neurons ($P = 0.49$). For each examined layer, the EPSP amplitude in PV+

neurons was larger than those in any other group of ACA neurons ($P < 0.005$, two-sided Wilcoxon rank-sum test). **g** Schematic of the slice experiment to measure PV+$_{cal}$-neuron-driven IPSCs in CPNs. **h** Example IPSCs recorded from a L2/3 CPN and a nearby Pyr. **i** Normalized PV+$_{cal}$-IPSC amplitudes in CPNs were larger than in other Pyrs across different layers ($P < 0.04$, two-sided Wilcoxon signed-rank test). **j** Schematic of the slice experiment to measure callosal inputs in PV+$_{cal}$ and other nearby PV+ neurons. **k** Example EPSPs recorded from L2/3 PV+$_{cal}$ and other PV + neurons. **l** Normalized EPSP amplitudes in PV+$_{cal}$ neurons were larger than in other PV+ neurons ($P < 0.04$, two-sided Wilcoxon rank-sum test). **m** Conceptual diagram of the transcallosal inhibition loop. Line width represents the amplitude of synaptic inputs. Data are presented as the mean ± SEM. The number of neurons in each group is indicated by the numbers displayed in the figure. Source data are provided as a Source Data file.

across different layers, but in each layer, PV+ neurons exhibited much stronger callosal inputs (greater than 2.8-fold) compared to SST+ and VIP+ neurons. (Fig. 7f). These results indicated that callosal inputs weakly activate SST+ and VIP+ neurons and strongly activate PV+ neurons. Interestingly, we found that callosal inputs were stronger in each type of L2/3 neurons examined here as compared to the same type of neurons in other cortical layers, excluding the similarly input strength for L2/3 PV+ and L6 PV+ neurons (Fig. 7f). These findings are consistent with previous studies reporting that L2/3 are the major callosal recipient layers in the V1, RSP and primary somatosensory cortex (S1)[59–61].

Given that the ipsilateral CPNs may also provide excitatory inputs to local interneurons and exhibit an opposite activity bias to contralateral CPNs in the visual-change-detection task, we investigated the input strengths from ipsilateral CPNs to ACA interneurons. To identify different types of interneurons, we co-injected Flp-dependent AAV expressing EYFP (AAV-EF1α-fDIO-EYFP) and AAV expressing tdTomato driven by a PV-specific enhancer (AAV-S5E2-tdTomato) into the ACA of one hemisphere in SST-Flp and PV-Flp mice (Supplementary Fig. 9a). Consequently, SST+ or VIP+ neurons expressed EYFP, while PV+ neurons expressed tdTomato (Supplementary Fig. 9b). To target ipsilateral CPNs for optogenetic activation, Cre-inducible AAV expressing ChR2 (AAV-EF1α-DIO-ChR2) was injected into the ACA together with labeling vectors for identifying distinct interneuron types; a retrograde AAV for Cre expression (Retro-AAV-hSyn-Cre) was injected into the opposite ACA. After confirming that optogenetic activation of ipsilateral CPNs induced large polysynaptic EPSPs in all three examined interneuron types without blocking local neuron spiking (Supplementary Fig. 9c), we applied TTX to suppress local neuronal spiking. Upon blocking, we found that optogenetic activation of ipsilateral CPNs induced very weak monosynaptic EPSPs (mean EPSP amplitude <1.2 mV) in all three interneuron types (Supplementary Fig. 9c). Notably, the amplitudes of both polysynaptic and monosynaptic EPSPs induced by the activation of ipsilateral CPNs were similar across all three interneuron types ($P > 0.07$, Wilcoxon rank-sum test). Thus, ipsilateral CPNs provide weak and non-selective inputs to all three examined interneuron types, whereas ACA PV+ neurons receive specific, strong inputs from contralateral CPNs; these findings reinforce the conclusion that inputs from contralateral CPNs preferentially activate PV+ neurons to provide transcallosal inhibition in visuospatial processing.

We subsequently examined the potential contribution of callosal-input-driven PV+ (PV+$_{cal}$) neurons to the transcallosal inhibition of CPNs. To activate PV+$_{cal}$ neurons, we injected anterograde Cre-inducible AAV1 expressing Flp (AAV1-EF1α-DIO-Flp) into the ACA of one hemisphere and Flp-dependent AAV expressing ChR2 (AAV-EF1α-fDIO-ChR2-mCherry) into the opposite ACA in PV-Cre mice (Fig. 7g). AAV1 anterogradely labeled ~50% of PV+ neurons (Supplementary Fig. 10). CPNs were identified by retrograde AAV expressing EGFP (Retro-AAV-hSyn-EGFP), which was injected together with AAV1.

Optogenetic activation of PV+$_{cal}$ neurons induced significantly stronger IPSCs in CPNs than in other pyramidal neurons examined across different layers (~4-fold in L2/3, >2-fold in L5 and L6, Fig. 7h, i). Notably, activation of all PV+ neurons evoked similar IPSCs in CPNs and other pyramidal neurons (Supplementary Fig. 11), indicating that PV+$_{cal}$ neurons likely mediate the observed stronger transcallosal inhibition in CPNs.

To investigate the contralateral input specificity of PV+$_{cal}$ neurons, we measured the strength of callosal inputs to PV+$_{cal}$ neurons compared to other PV+ neuron types. To differentiate PV+$_{cal}$ neurons from other PV+ neurons, we injected anterograde Cre-inducible AAV1 expressing Flp (AAV1-EF1α-DIO-Flp) into the ACA of one hemisphere and Flp-dependent AAV expressing EYFP (AAV-EF1α-fDIO-EYFP) into the opposite ACA in PV-tdTomato mice (Fig. 7j). AAV1 anterogradely labeled PV+$_{cal}$ neurons express both EYFP and tdTomato, and other PV+ neurons only express tdTomato. To optogenetically activate callosal inputs onto PV+$_{cal}$ neurons, AAV-CaMKIIα-ChR2 was co-injected with AAV1. We found that PV+$_{cal}$ neurons received much stronger callosal inputs (~1.8-fold) than other PV+ neurons (Fig. 7k, l), supporting a major role for PV+$_{cal}$ neurons in providing callosal-input-activation induced transcallosal inhibition of ipsilateral CPNs. Together, our results demonstrate that ACA CPNs and PV+$_{cal}$ neurons form a transcallosal loop providing transcallosal inhibition to maintain interhemispheric balance (Fig. 7m). Equal CPN activity in each hemisphere enable balanced transcallosal inhibition through activating contralateral PV+$_{cal}$ neurons. However, if one hemisphere has higher CPN activity, it will provide stronger transcallosal inhibition to the other hemisphere, resulting in a bias in the interhemispheric balance.

## ACA PV+$_{cal}$ neurons exhibit selective visual-related activity with an ipsilateral bias

To further investigate the contribution of ACA PV+$_{cal}$ neurons in visuospatial processing, we measured their activity during the 2AUC change-detection task. The calcium indicator GCaMP6s was expressed in PV+$_{cal}$ neurons by injecting anterograde Cre-inducible AAV1 expressing Flp (AAV1-EF1α-DIO-Flp) into the ACA of one hemisphere and injecting Flp-dependent AAV expressing GCaMP6s (AAV-EF1α-fDIO-GCaMP6s) in the opposite ACA in PV-Cre mice (Fig. 8a). Calcium imaging was performed during the 2AUC change-detection task (Fig. 8b–d, and Supplementary Figs. 2 and 12). During the visuomotor period, 26% of PV+$_{cal}$ neurons responded differentially in contralateral-hit vs. ipsilateral-hit trials. Notably, there were significantly more ipsilateral-preferring PV+$_{cal}$ neurons than contralateral-preferring PV+$_{cal}$ neurons (35 vs. 4, Fig. 8e). Measurement of the change-detection-task-related activity of PV+ neurons also revealed more ipsilateral-preferring PV+ neurons than contralateral-preferring PV+ neurons (Supplementary Fig. 13); however, in this case, there was a significantly larger proportion of contralateral-preferring PV+ neurons than contralateral-preferring PV+$_{cal}$ neurons ($P = 2 \times 10^{-6}$, chi-square test), indicating enrichment of ipsilateral-preferring neurons in the

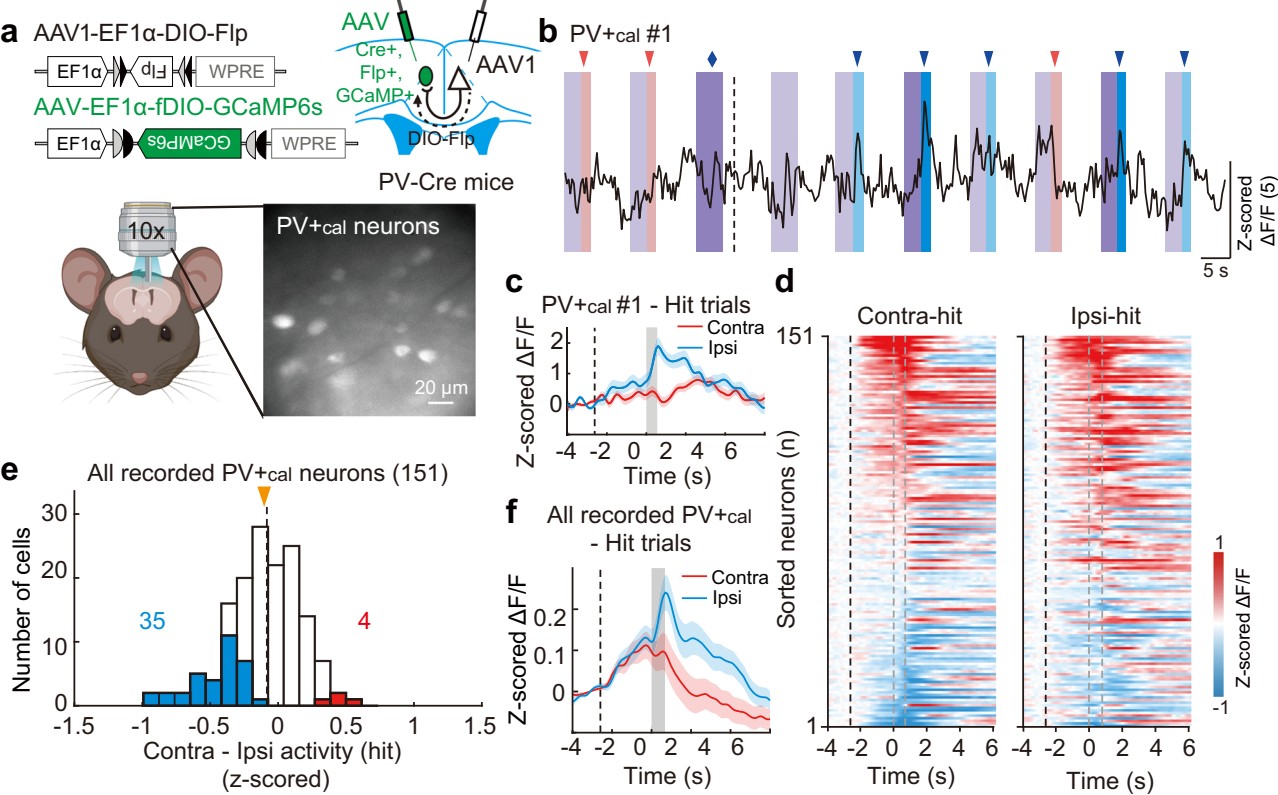

**Fig. 8 | ACA PV+$_{cal}$ neurons show an ipsilateral bias in the 2AUC change-detection task. a** Schematic diagrams of the viral strategy (top) and calcium imaging of PV+$_{cal}$ neurons (bottom). **b** Raw fluorescence trace of an example PV+$_{cal}$ neuron. Same behavior paradigm as the test phase shown in Fig. 1h, containing the trials at 100% and 12.5% contrast levels. Dark color, high contrast; light color, low contrast. Purple shading, visual white noise; red shadings, contralateral gratings; blue shading, ipsilateral gratings. Red and blue arrowheads, end of wheel rotation in the contralateral and ipsilateral-hit trials, respectively. Blue diamond, end of ipsilateral wheel rotation in the FA trial. Dashed line, end of timeout in the FA trial. **c** Averaged ΔF/F (z-scored) of an example PV+$_{cal}$ neuron in hit trials. Colored shading, ± SEM. Black dashed line, the start of visual white noise. Gray shading, window of the visuomotor period (from the start of drifting grating to the end of wheel rotation). **d** Color-coded averaged ΔF/F (z-scored) of all recorded PV+$_{cal}$ neurons in contralateral-hit (left) and ipsilateral-hit (right) trials (n = 8 mice,

8 sessions, 151 neurons). Each row shows the average activity of one neuron, sorted by peak response during the visuomotor period. Black dashed line, the start of visual white noise. The left and right gray dashed lines present the start of drifting grating and the end of wheel rotation, respectively. **e** Distribution of the difference in peak responses of z-scored ΔF/F during the visuomotor period between the contralateral-hit and ipsilateral-hit trials for the imaged PV+$_{cal}$ neurons. Blue, ipsilateral-preferring cells; red, contralateral-preferring cells. Orange arrowhead, median. There were significantly more ipsilateral-preferring PV+$_{cal}$ neurons than contralateral-preferring PV+$_{cal}$ neurons ($P = 2 \times 10^{-5}$, two-sided chi-square test). **f** Similar to **c**, but for all recorded PV+$_{cal}$ neurons. The averaged responses of PV+$_{cal}$ neurons were significantly stronger in the ipsilateral-hit trials than in the contralateral-hit trials during the visuomotor period ($P = 3 \times 10^{-5}$, two-sided paired t-test). See also Supplementary Fig. 12 and "Methods". Source data are provided as a Source Data file. Created with BioRender.com.

PV+$_{cal}$ neuron population. In addition, the averaged responses of ACA PV+$_{cal}$ neurons were significantly stronger in the ipsilateral-hit trials than in the contralateral-hit trials during the visuomotor period, showing the inverse scenario of the aforementioned CPNs (Fig. 8f).

Given the relative dearth of contralateral-preferring PV+$_{cal}$ neurons, we focused the following analysis on ipsilateral-preferring PV+$_{cal}$ neurons. At the population level, ipsilateral-preferring PV+$_{cal}$ neurons showed selectivity for visual stimuli in miss trials, indicating that they encoded the properties of visual stimulus (Fig. 9 and Supplementary Fig. 14). Similar to CPNs, the differences in the responses of PV+$_{cal}$ neurons to contralateral versus ipsilateral visual change were significantly greater in hit trials compared to miss trials (ipsilateral-preferring CPNs, $P = 1.2 \times 10^{-4}$, paired t-test), likely owing to differences in behavioral states. In addition, ipsilateral-preferring PV+$_{cal}$ neurons did not show selectivity for the direction of motor actions in FA trials, indicating that they did not encode the properties of motor action. Together, these results demonstrate that the ACA PV+$_{cal}$ neurons show selective visual-related activity that is biased towards the ipsilateral side.

**Activation of PV+$_{cal}$ neurons results in similar behavioral bias to contralateral CPN activation and ipsilateral CPN inactivation**

We next optogenetically activated ACA PV+$_{cal}$ neurons to directly examine their functions in the 2AUC change-detection task. ChR2 was expressed in PV+$_{cal}$ neurons by injecting anterograde Cre-inducible AAV1 expressing Flp (AAV1-EF1α-DIO-Flp) into the ACA of one hemisphere and injecting Flp-dependent AAV expressing ChR2 (AAV-EF1α-fDIO-ChR2-EYFP) in the opposite ACA in PV-Cre mice (Supplementary Fig. 15a, b). ChR2-mediated unilateral activation of ACA PV+$_{cal}$ neurons resulted in impaired change detection of the contralateral visual stimulus and in a simultaneous improvement in ipsilateral performance (Supplementary Fig. 15c–g). The behavioral bias induced by PV+$_{cal}$ neuron activation was similar to that of contralateral CPN activation and ipsilateral CPN inactivation (Supplementary Data 1), supporting the notion that contralateral CPNs activate PV+$_{cal}$ neurons via callosal projections to generate transcallosal inhibition and in turn inhibit ipsilateral CPNs, thus mediate interhemispheric balance in visuospatial processing.

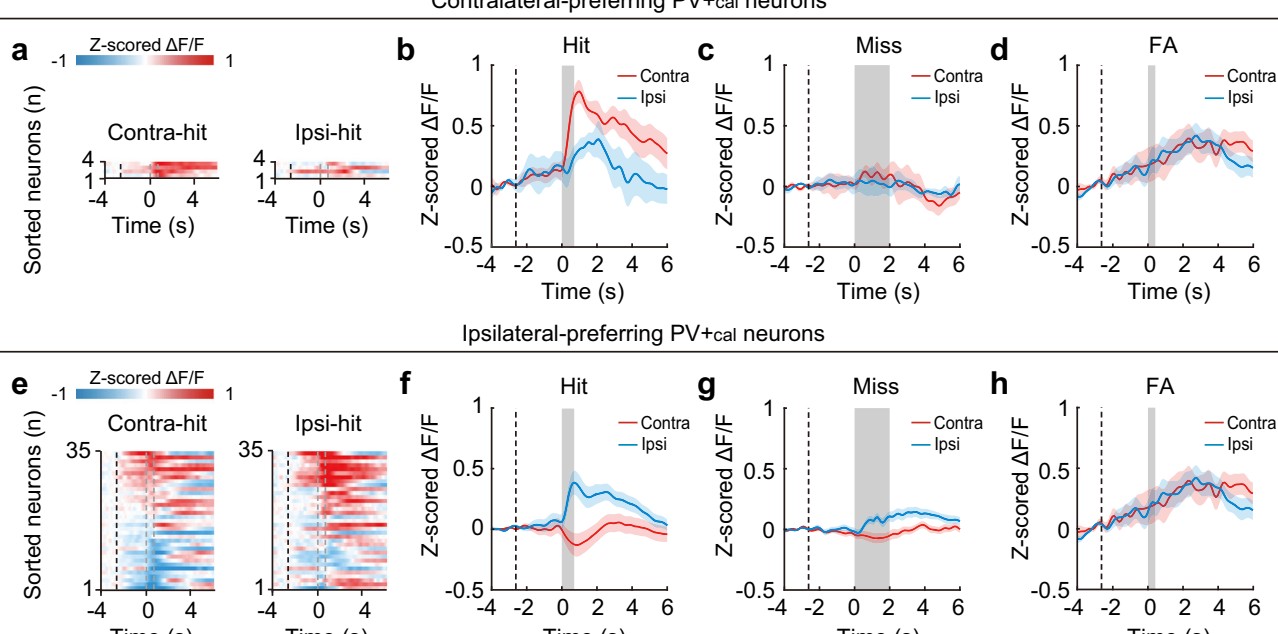

**Fig. 9 | ACA PV+$_{cal}$ neurons show selective visual-related activity in the 2AUC change-detection task. a** Color-coded averaged ΔF/F (z-scored) of the contralateral-preferring PV+$_{cal}$ neurons in contralateral-hit (left) and ipsilateral-hit (right) trials ($n = 8$ mice, 8 sessions, 4 neurons). Each row shows the average activity of one neuron, sorted by peak response during the visuomotor period in the contralateral-hit trials. Black dashed line, the start of visual white noise. The left and right gray dashed lines present the start of drifting grating and the end of wheel rotation, respectively. **b–d** Averaged ΔF/F (z-scored) of contralateral-preferring PV+$_{cal}$ neurons in different types of trials. Black dashed line, the start of visual white noise. Colored shading, ± SEM. **b** Activity in hit trials. Gray shading, window of the visuomotor period. **c** Activity in miss trials. Gray shading, window during the presentation of drifting grating. **d** Activity in FA trials. Gray shading, window during the wheel rotation. **e–h** Similar to **a–d**, but for ipsilateral-preferring PV+$_{cal}$ neurons. The activity of ipsilateral-preferring PV+$_{cal}$ neurons was significantly lower in the contralateral-hit trials than in the ipsilateral-hit trials during the visuomotor period ($P = 1.8 \times 10^{-8}$, two-sided paired $t$-test). Similarly, the activity of ipsilateral-preferring PV+$_{cal}$ neurons was significantly lower in the contralateral-miss trials than in the ipsilateral-miss trials during the presentation of drifting grating ($P = 0.003$, two-sided paired $t$-test). However, there was no significant activity difference between the contralateral-FA trials and the ipsilateral-FA trials during the wheel rotation ($P = 0.8$, two-sided paired $t$-test). See also Supplementary Fig. 14 and "Methods". Source data are provided as a Source Data file.

## Activation of contralesional PV+ neurons reverses lesion-induced visuospatial bias

Given our findings that PV+ neurons mediate transcallosal inhibition between the ACA in the two hemispheres, we tested whether the behavioral bias induced by unilateral inactivation of the ACA can be reversed by activation of contralateral ACA PV+ neurons. To optogenetically activate ACA PV+ neurons, we crossed PV-Cre mice with loxP-flanked ChR2 reporter mice. Unilateral activation of ACA PV+ neurons, which has been shown to suppress pyramidal neuron spiking[62–66], resulted in impaired change detection to contralateral visual stimulus, with simultaneous improvement in ipsilateral change detection (Fig. 10a–c, and Supplementary Fig. 16), findings similar to the behavioral bias induced upon unilateral inactivation of ACA CPNs. Notably, bilateral (i.e., simultaneous) activation of ACA PV+ neurons in the two hemispheres eliminated this behavioral bias (Fig. 10d, e, and Supplementary Fig. 16), indicating that rebalanced interhemispheric interactions reverse visuospatial bias following unilateral inactivation of the ACA.

We then examined whether activating contralesional ACA PV+ neurons in ACA-lesioned mice conferred any benefits for contralesional change detection. AAV expressing hM3D driven by a PV-specific enhancer (AAV-S5E2-hM3D-tdTomato)[67] (for chemogenetic activation) was injected into the contralesional ACA of CaMKIIα-Cre mice (Fig. 10f, and Supplementary Fig. 17). These animals were then trained to perform the 2AUC change-detection task with activation of PV+ neurons (expressing hM3D) induced by daily injection of Clozapine N-oxide (CNO) into the contralesional ACA. Daily CNO injection during learning (for 12 days) resulted in prolonged improvement in contralesional performance, showing a

significantly increased contralesional hit rate, both during and after learning (Fig. 10g). These results demonstrated that lesion-induced visuospatial bias can be reversed by enhancing transcallosal inhibition via short-term activation of PV+ neurons in the contralesional ACA (conceptual diagram shown in Fig. 10h).

## Discussion

In this study, we found that a transcallosal inhibition loop in the mouse ACA—a frontal cortical area providing top-down modulation to visual cortices[33,35–39]—mediates interhemispheric balance in visuospatial processing. This transcallosal loop comprises CPNs and PV+$_{cal}$ neurons in the left and right ACA. In this loop, left-CPNs preferentially activate right-PV+$_{cal}$ neurons through the corpus callosum to provide specific transcallosal inhibition of right-CPNs, in turn reducing the excitatory inputs from right-CPNs to left-PV+$_{cal}$ neurons and thereby reducing the transcallosal inhibition of left-CPNs, leading to further increase of left-CPN activity. In the 2AUC change-detection task, increasing the activity of left-CPNs facilitates change detection in the right visual field, and simultaneously impairs change detection in the left visual field, ostensibly owing to the enhanced activity of right-PV+$_{cal}$ neurons and the reduced activity of right-CPNs.

In mammals, both sensory and motor systems are highly lateralized. Unilateral lesions reduce ipsilesional network activity, disrupt the interaction balance between the two hemispheres, and bias the sensory and motor processing towards the ipsilesional side[1,6,7,68]. Our results showed that unilateral inactivation of the ACA in mice biased their performance toward the ipsilateral side in a visual-change detection task, consistent with previous studies on unilateral inactivation of sensory and motor cortices in various sensory-motor

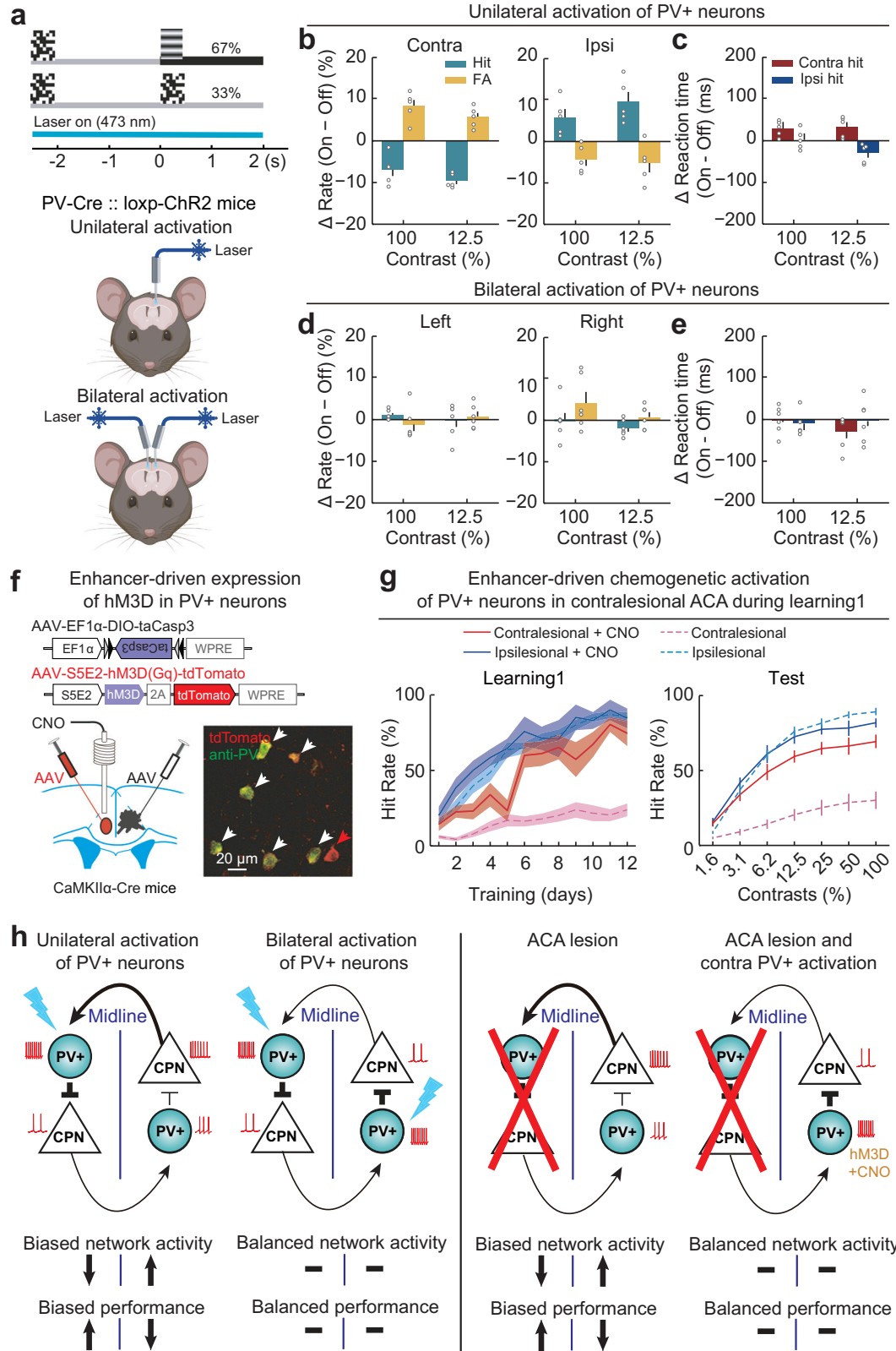

tasks[44,62,66,69], thus supporting consistent functional consequences from imbalanced interhemispheric interaction.

Previous studies have shown that activation of callosal projections can drive excitation and/or inhibition in cortical circuits[51,70–75]. While the callosal projections between the primary visual cortices provide interhemispheric facilitation of visual response gain[5,12–15,18], the callosal connections between the frontal cortices allow

interhemispheric inhibition in visuospatial processing[3,6]. Consistent with the callosal projections between the visual, somatosensory, auditory, retrosplenial, and prelimbic cortices[19,70,71,73,75,76], our results support the notion that the callosal projections between the frontal cortices directly innervate both excitatory and inhibitory neurons. Since the behavioral effects induced by inhibition of callosal-projection axon terminals from the opposite ACA are similar to

**Fig. 10 | Activation of contralesional PV+ neurons improved contralesional change detection. a** Schematic of optogenetic manipulation. **b** Effect of unilateral PV+-neuron activation on visual-change detection ($n = 5$ mice). Left, changes in contralateral performance. Right, changes in ipsilateral performance. Unilateral PV+-neuron activation (473 nm, 3–8 mW, 10 Hz, 5 ms per pulse) significantly impaired contralateral detection (decreased hit rates, green bars, $F_{laser}(1,4) = 115.94$, $P_{laser} = 4 \times 10^{-4}$; increased FA rates, yellow bars, $F_{laser}(1,4) = 34.77$, $P_{laser} = 0.004$; two-way repeated ANOVA) and improved ipsilateral detection (increased hit rates, green bars, $F_{laser}(1,4) = 19.70$, $P_{laser} = 0.01$; decreased FA rates, yellow bars, $F_{laser}(1,4) = 8.19$, $P_{laser} = 0.046$). **c** Effect of unilateral PV+-neuron activation on reaction times in hit trials ($n = 5$ mice), showing significantly increased contralateral and decreased ipsilateral reaction times at 12.5% contrast (contralateral, $F_{laser}(1,4) = 8.09$, $P_{laser} = 0.047$; ipsilateral, $F_{laser}(1,4) = 8.12$, $P_{laser} = 0.046$). **d, e** Similar to **b, c**, but for bilateral PV+-neuron activation ($n = 6$ mice). Bilateral PV+-neuron activation did not induce visual-detection or reaction-time changes on either side. **f** Schematic for chemogenetic activation of contralesional PV+ neurons. Lower right, fluorescence image showing neurons with enhancer-driven expression of hM3D/tdTomato (red) and anti-PV staining (green). Red arrowheads, hM3D/tdTomato-only neurons; white arrowheads, double labeled neurons. **g** Performance of ACA-lesioned mice in learning phase1 and the test phase with/without contralesional PV+-neuron activation during learning phase1. Contralesional hit rates were significantly higher in CNO-treated ACA-lesioned mice ($n = 6$ mice) than in ACA-lesioned mice that did not receive CNO (Learning1, ACA-lesioned no-CNO, $n = 21$ mice, $F_{CNO}(1,300) = 187.40$, $P_{CNO} = 2 \times 10^{-33}$; test, ACA-lesioned no-CNO, $n = 11$ mice, $F_{CNO}(1,105) = 152.90$, $P_{CNO} = 3 \times 10^{-22}$, two-way ANOVA). Data shown with dashed lines, same datasets as in Fig. 1e, j. Colored shading and error bar represent the ± SEM. **h** Diagram for the transcallosal loop mediating interhemispheric inhibition in visuospatial processing. Line width: amplitude of synaptic inputs. Data are presented as the mean ± SEM. See Supplementary Data 2 for ANOVA parameters. Source data are provided as a Source Data file. Created with BioRender.com.

those caused by activating CPNs in the same hemisphere, the net impact of ACA callosal projections is inhibitory. Future studies delineating the area-specific callosal-input strengths on different types of postsynaptic neurons across multiple cortical areas should bring clarity to the ongoing debate about whether callosal projections are excitatory or inhibitory in cortical circuits, and will likely resolve how they exert distinct functions.

Cortical excitatory neurons have been divided into three main classes based on their axonal projection patterns: intratelencephalic (IT), pyramidal tract (PT), and corticothalamic (CT) neurons[77]. IT neurons, which are distributed across multiple cortical layers (L2–L6) and project to diverse cortical areas and the striatum, are the only cortical excitatory neurons that project to the contralateral hemisphere; thus, CPNs are conceptualized as IT neurons[77]. PT and CT neurons are distributed in deeper layers (L5 and L6), and mainly project to ipsilateral subcortical structures[29,77]. We found that frontal callosal inputs were stronger in almost every type of L2/3 neurons as compared to the same type of neurons in other cortical layers (L5 and L6). Considering that L2/3 IT neurons mostly distribute axons within the cerebral cortex[29], the frontal callosal projections may first affect the activity of cortical networks, and then propagate to subcortical structures, e.g., the thalamus, striatum, and superior colliculus[29,78–81]. A previous study of the mouse anterior lateral motor cortex (ALM) showed that the proportion of contralateral-lick preferring neurons was higher among PT neurons than among IT neurons[64]. Although PT and CT neurons do not directly project to the contralateral hemisphere, they may affect interhemispheric balance through indirect pathways, e.g., corticothalamic-cortical pathways.

In addition to PV+ neurons, other local interneurons may also contribute to interhemispheric balance in visuospatial processing. For example, layer 1 GABAergic neurons activated by callosal inputs have been shown to suppress the apical dendrites of pyramidal neurons in rat somatosensory cortex[73]. The frontal transcallosal loop comprising CPNs and PV+$_{cal}$ neurons identified in the current study can enhance the activity difference between the two hemispheres. Thus, it seems that healthy brains must engage some additional regulatory mechanism(s) to restore the interhemispheric balance. SST+ neurons receive strongly facilitating local cortical excitatory inputs[82]; perhaps these may be recruited in the active hemisphere to inhibit both inhibitory and excitatory neurons[83,84] and thereby rebalance interhemispheric interactions in healthy brains.

Previous lesion studies in felines and clinical observations on stroke patients have shown that a contralateral brain lesion on a previously spared hemisphere can, paradoxically, cancel the deficits in visuospatial functions generated by a first lesion (e.g., can enable recovery from visuospatial neglect, an attentional deficit)[6,85,86]. This is often referred to as the "Sprague Effect," named after J.M. Sprague who first reported it in the 1960s[87]. Consistent with previous studies, we found that bilateral activation of PV+ neurons reversed the unilateral

inactivation/lesion-induced visuospatial bias in rodents in a visual-change-detection task, suggesting that the ACA may contribute to the attentional components of decision-making in this task. These results indicate that unilateral lesion-induced visuospatial bias is mediated by imbalanced network activity between two hemispheres, rather than by the activity level of each brain structure in the network. Previous studies have shown that multiple cortical and subcortical brain structures exhibit task-related activity in visually guided decision-making[45,66,81]. In the current study, we also observed some extent of bias upon lesion of the RSP and PTLp. It is plausible that some compensatory activity (i.e., some form of rebalancing, as mediated by different structures or some combination of structures) could in theory be engaged upon bilateral ACA lesion that could explain the apparent canceling (*NOTA BENE*: assessed only as the final behavioral readout) of the bias that was uniquely evident upon unilateral lesion. Further, we found that short-term (12 days) activation of PV+ neurons in the contralesional ACA resulted in a benefit to contralesional detection, both during and after learning. Accordingly, our results support the notion that reestablishing interhemispheric balance can promote recovery from visuospatial bias after lesion.

The mouse ACA is anatomically well-positioned to provide top-down control signals for visual information processing. It is the only frontal cortical area that receives direct inputs from the primary visual cortex and provided a substantial amount of direct axon innervation to the primary visual cortex[29,30,56]. The ACA has been shown to modulate different aspects of visually guided goal-directed behavior in mice[33,35–39]. Activation of the ACA improved visual feature detection—a hallmark feature of visual attention[33,35,39,88]. Our results show that activation of ACA CPNs improves the ability to detect contralateral visual changes and decreases the reaction time to contralateral changes, findings consistent with the effects of attention modulation in primates[89–91]. The functional characterization of callosal projections between frontal areas is still in its infancy[3], and our study paves the way for subsequent detailed analyses of specific functional impacts of frontal callosal projections on higher cognitive functions, potentially with cell-type-specific resolution.

Mouse models have been used to examine the roles of specific cell types and circuits in visual perception and behavior. While the visual systems of mice and humans share certain aspects, significant differences should be noted: humans have better visual acuity and color vision, while mice have a wider field of view and better low-light vision[7,92,93]. Owing to the lateral eye placement in mice, they have larger monocular zones yet a smaller binocular zone compared to humans[7,93]. Moreover, mice have fewer specialized brain areas for parallel visual information processing[92], and multisensory integration is known to occur at an earlier stage in mice[94]. Although results obtained from mice cannot be directly extrapolated to humans or non-human primates, the use of enhancer-driven viral tools for PV+ neurons allows for selective targeting and manipulation of specific

interneuron types across species[67], thereby providing the possibility to examine whether the cell-type-specific mechanisms of transcallosal inhibition in visuospatial processing revealed here in mice also function in primates. Our successful demonstration of reversing lesion-induced deficits based on short-term activation of contralesional PV+ neurons to restore interhemispheric balance may inspire the development of innovative treatments for lesion-induced visuospatial bias based on enhancing transcallosal inhibition.

# Methods

## Animals
Animal care and the experimental protocols were approved by the Animal Committee of Shanghai Jiao Tong University School of Medicine and the Animal Committee of the Institute of Neuroscience, Chinese Academy of Sciences. Experiments were performed on wild-type (C57) and transgenic mice. The transgenic mice used were PV-Cre (Jackson lab stock #017320), SST-Cre (#013044), VIP-Cre (#010908), CaMKIIα-Cre (#005359), loxP-flanked-tdTomato (#007909), loxP-flanked-ChR2-EYFP (#024109), SST-Flp (#031629), and VIP-Flp (#028578) mice. To visualize the interneurons, PV-, SST- or VIP-Cre mice were crossed with loxP-flanked-tdTomato mice. To activate the PV+ neurons, PV-Cre mice were crossed with loxP-flanked-ChR2-EYFP mice. Male and female mice aged 1.5–6 months were used. Mice were group housed (5–6/cage) in the standard laboratory condition (temperature: $23 \pm 1\,°C$; humidity: 50–70%) under a 12 h light/dark cycle (light on from 7 a.m. to 7 p.m.).

## Viral constructs
The adeno-associated viruses AAV-DJ-EF1α-DIO-taCasp3 (genomic titer, $2 \times 10^{13}$ gc/mL), AAV2-CAG-DIO-TVA-mCherry ($7 \times 10^{12}$ gc/mL), AAV2-CAG-DIO-RG ($8 \times 10^{12}$ gc/mL), Retro-AAV-hSyn-Cre ($7 \times 10^{13}$ gc/mL), AAV2-CaMKIIα-EYFP ($8 \times 10^{12}$ gc/mL), AAV-DJ-CaMKIIα-ChR2-EYFP ($8 \times 10^{12}$ gc/mL), AAV9-S5E2-Gq-tdTomato ($1.8 \times 10^{12}$ gc/mL), AAV9-S5E2-tdTomato ($5 \times 10^{12}$ gc/mL), and AAV9-EF1a-DIO-hChR2(H134R) ($3 \times 10^{12}$ gc/mL) were acquired from the WZ Biosciences. AAV9-EF1α-DIO-GCaMP6s ($5 \times 10^{12}$ gc/mL), AAV9-EF1a-DIO-hChR2(H134R)-EYFP ($5 \times 10^{12}$ gc/mL), AAV9-EF1α-DIO-eNpHR3.0-EYFP ($2 \times 10^{12}$ gc/mL), Retro-AAV-hSyn-EYFP ($5 \times 10^{12}$ gc/mL) and Retro-AAV-hSyn-mCherry ($5 \times 10^{12}$ gc/mL) were acquired from BrainVTA. AAV1-EF1α-DIO-Flp ($1 \times 10^{13}$ gc/mL), AAV9-EF1a-fDIO-EYFP ($2 \times 10^{12}$ gc/mL), and AAV9-hSyn-SIO-eOPN3-mScarlet ($1 \times 10^{12}$ gc/mL) were acquired from BrainCase. AAV9-EF1α-fDIO-GCaMP6s ($5 \times 10^{12}$ gc/mL) and AAV9-hSyn-DIO-ChrimsonR-EYFP ($3 \times 10^{12}$ gc/mL) was acquired from Taitool Bioscience. Glycoprotein-deleted ($\Delta G$) and EnvA-pseudotyped rabies virus (RV-ΔG-EGFP+EnvA, $2 \times 10^8$ IU/mL) was acquired from BrainVTA.

## Surgery
**Stereotaxic surgeries.** Adult mice were anesthetized with isoflurane (5% induction and 1.5% maintenance) and placed on a stereotaxic frame (Ruiwode Life Science). The temperature was kept at 37 °C throughout the procedure using a heating pad. Eye ointment was applied to protect the animal's eyes. After asepsis, the skin was incised to expose the skull and the overlying connective tissue was removed. A craniotomy (~0.5 mm diameter) was made above the injection site. Viruses were loaded in a sharp micropipette mounted on a Nanoject II attached to a micromanipulator and then injected at a speed of 60 nL per min. Coordinates used were as following: ACA (Bregma, +0.3 mm; lateral, 0.3 mm; depth, 0.9 mm), RSP (Bregma, −1.8 mm; lateral, 0.3 mm; depth, 0.5 mm), or PTLp (Bregma, −2.2 mm; lateral, 1.5 mm; depth, 0.5 mm). For behavioral experiments, a stainless steel headplate was affixed to the skull using machine screws and dental cement for head fixation after 1-2 weeks of virus injection. Behavioral experiments were performed >3 weeks after injection.

**Lesion.** For lesion experiments, AAV9-EF1α-DIO-taCasp3 (500–1000 nL) was injected into the ACA, RSP or PTLp of CaMKIIα-Cre mice in one hemisphere to induce the death of pyramidal neurons.

**Rabies-virus-mediated retrograde monosynaptic tracing.** For retrograde monosynaptic tracing from cortical interneurons and pyramidal neurons, TVA receptor and rabies glycoprotein, which are required for virus infection and trans-synaptic spread, respectively, were expressed in Cre-positive neurons by co-injection of AAV2-CAG-DIO-TVA-mCherry and AAV2-CAG-DIO-Glycoprotein (1:2, 200–500 nL) into the ACA of PV-, SST-, VIP- and CaMKIIα-Cre mice. Two to three weeks later, RV-ΔG-EGFP+EnvA (200–500 nL) was injected into the same site as AAV injection. The histology experiments were performed 7 days after rabies virus injection.

**Calcium imaging through GRIN lens.** For imaging experiments on CPNs, the calcium indicator GCaMP6s was expressed in ACA CPNs by injecting retrograde AAV expressing Cre (Retro-AAV-hSyn-Cre, 500 nL) into the ACA of one hemisphere and injecting Cre-inducible AAV expressing GCaMP6s (AAV9-EF1α-DIO-GCaMP6s, 500 nL) into the opposite ACA in wild-type mice (C57). For imaging experiments on PV+cal neurons, we injected anterograde Cre-inducible AAV1 expressing Flp (AAV1-EF1α-DIO-Flp, 500 nL) into the ACA of one hemisphere and Flp-dependent AAV expressing GCaMP6s (AAV9-EF1α-fDIO-GCaMP6s, 500 nL) into the opposite ACA in PV-Cre mice. For imaging experiments on PV+ neurons, Cre-inducible AAV expressing GCaMP6s (AAV9-EF1α-DIO-GCaMP6s, 400 nL) was injected into the ACA of one hemisphere in PV-Cre mice.

After 1-2 weeks of virus injection, the GRIN lens (Go!Foton #: GFK-000786-PO; diameter: 0.5 mm; length: 3.692 mm; pitch: 0.433) were implanted 0.3–0.4 mm above the injection site of GCaMP6s-expressing virus. First, we expanded the virus-injection craniotomy to ~0.8 mm and slowly ruptured the dura using a stainless needle, avoiding blood clotting by constantly irrigating the tissue with sterile saline solution. After ensuring there was no active bleeding, the GRIN lens was gently placed upon the tissue and the space between the lens and the skull was sealed with silicone sealant (Kwik-Cast, WPI). The lens was then cemented to the rest of the implant with black dental cement and covered with a detachable plastic cap.

**Optogenetics.** For optogenetic activation of ACA CPNs, we injected retrograde AAV expressing Cre (Retro-AAV2-hSyn-Cre, 500 nL) into the ACA of one hemipshere and Cre-inducible AAV expressing ChR2 (AAV9-EF1α-DIO-ChR2-EYFP, 500 nL) into the opposite ACA in wild-type mice. For optogenetic inactivation of ACA CPNs, we injected retrograde AAV expressing Cre (Retro-AAV2-hSyn-Cre, 500 nL) in the ACA of one hemisphere and Cre-inducible AAV expressing NpHR (AAV9-EF1α-DIO-eNpHR3.0-EYFP, 400 nL) into the opposite ACA in wild-type mice. For optogenetic inactivation of CPN axon terminals, we injected retrograde AAV expressing Cre (Retro-AAV2-hSyn-Cre, 500 nL) in the ACA of one hemisphere and Cre-inducible AAV expressing eOPN3 (AAV9-hSyn-SIO-eOPN3-mScarlet, 400 nL) into the opposite ACA in wild-type mice. For control, AAV expressing EYFP (AAV9-CaMKIIα-EYFP, 500 nL) was unilaterally injected in the ACA of wild-type mice. After 1-2 weeks of recovery, optic fiber (Thorlabs, FT200UMT) attached to the implanted ferrule via a ceramic sleeve was implanted 0.3–0.4 mm above the injection site of AAVs expressing ChR2, NpHR or EYFP to deliver light for optogenetic manipulation of CPNs and control experiments. For optogenetic inactivation of CPN axon terminals, optic fibers were implanted to the opposite ACA of the injection site of AAV expressing eOPN3. For optogenetic activation of PV+ neurons, optic fibers were unilaterally or bilaterally implanted in the ACA of PV-ChR2 mice (PV-Cre mice crossed with loxP-flanked ChR2 reporter mice). Moreover, a layer of black dental cement was added to

additionally secure the optic fibers and reduce light emission from the skull.

**In vitro electrophysiology.** To examine the temporal dynamics of eOPN3-induced inactivation of callosal projection axons, we injected retrograde AAV expressing Cre (Retro-AAV-hSyn-Cre, 500 nL) in the ACA of one hemisphere, and mixed Cre-inducible AAVs expressing ChrimsonR and eOPN3 (AAV9-hSyn-DIO-ChrimsonR-EYFP, 200 nL; AAV9-hSyn-SIO-eOPN3-mScarlet, 200 nL) into the opposite ACA in wild-type mice. To compare the callosal inputs induced response on CPNs and other pyramidal neurons, we injected retrograde AAV expressing Cre (Retro-AAV-hSyn-Cre, 500 nL) and AAV-DJ-CaMKIIα-ChR2-EYFP (500 nL) in the ACA of one hemisphere in loxP-flanked tdTomato reporter mice. To measure the inputs from CPNs to each class of interneurons in the contralateral hemisphere (callosal inputs), we injected AAV-DJ-CaMKIIα-ChR2-EYFP (500 nL) in the ACA of one hemisphere in PV-, SST- and VIP-tdTomato mice. To measure the inputs from CPNs to each class of interneurons in the ipsilateral hemisphere, we injected retrograde AAV expressing Cre (Retro-AAV-hSyn-Cre, 500 nL) in the ACA of one hemisphere, and mixed Cre-inducible AAVs expressing ChR2 (AAV9-EF1α-DIO-ChR2, 300 nL), Flp-dependent AAV expressing EYFP (AAV9-EF1α-fDIO-EYFP, 300 nL) and PV-specific enhancer-driven AAV expressing of tdTomato (AAV9-S5E2-tdTomato, 300 nL) into the opposite ACA in SST-Flp and VIP-Flp mice. To compare the inhibition induced by PV+$_{cal}$ neurons on CPNs and other pyramidal neurons, we injected anterograde Cre-inducible AAV1 expressing Flp (AAV1-EF1α-DIO-Flp, 400 nL) and retrograde AAV expressing EGFP (Retro-AAV-hSyn-EGFP, 300 nL) into the ACA of one hemisphere, and Flp-dependent AAV expressing ChR2 (AAV9-EF1α-fDIO-ChR2-mCherry, 400 nL) into the opposite ACA in PV-Cre mice. To compare the inhibition induced by PV+ neurons on CPNs and other pyramidal neurons, we injected retrograde AAV expressing mCherry (Retro-AAV-hSyn-mCherry, 500 nL) into the ACA of one hemisphere and Cre-dependent AAV expressing ChR2 (AAV9-EF1α-DIO-ChR2-EYFP, 500 nL) into the opposite ACA in PV-Cre mice. To compare the callosal inputs induced excitatory response on PV+$_{cal}$ neurons and other PV+ neurons, we injected anterograde Cre-inducible AAV1 expressing Flp (AAV1-EF1α-DIO-Flp, 600 nL) and AAV-DJ-CaMKIIα-ChR2-EYFP (300 nL) into the ACA of one hemisphere, and Flp-dependent AAV expressing EYFP (AAV9-EF1α-fDIO-EYFP, 500 nL) into the opposite ACA in PV-tdTomato mice.

**Chemogenetics.** To investigate the effect of chemogenetic activation of contralesional PV+ neurons on contralesional visual-change detection, Cre-inducible AAV expressing taCasp3 (AAV9-EF1α-DIO-taCasp3, 1 µL) was injected into the ACA of one hemisphere in CaMKIIα-Cre mice to induce the death of pyramidal neurons. After six weeks, PV-specific enhancer-driven AAV expressing of hM3D (AAV9-S5E2-hM3D-tdTomato, 500 nL) for chemogenetic activation of PV+ neurons was injected in the opposite ACA of the first injection. After another week of recovery, a stainless steel guide cannula (Plastics One) was implanted in ACA 0.2 mm above the injection site of AAV9-S5E2-hM3D-tdTomato to deliver CNO. The cannula was then cemented with the head plate.

**Histology and anatomical data analysis**
Mice were deeply anesthetized with isoflurane and immediately perfused with chilled 0.1 M phosphate-buffered saline (PBS) followed by 4% paraformaldehyde (w/v) in PBS. The brain was removed and postfixed overnight at 4 °C. After fixation, the brain was placed in 30% sucrose (w/v) in PBS solution for 1–2 days at 4 °C. After embedding and freezing, the brain was sectioned into 50-µm coronal slices using a cryostat.

For fluorescent Nissl staining, brain slices were rehydrated with PBS for 0.5 h and permeabilized using PBST (0.3% Triton X-100 in PBS)

for 10 min and washed twice in PBS for 5 min each. The slices were then incubated with the NeuroTrace (1:200; N-21480, Thermo Fisher Scientific) for 20 min, washed with PBS for 2 hrs, and mounted with VECTASHIELD mounting medium (Vectorlabs).

For fluorescence images without staining, brain slices were rehydrated and mounted with VECTASHIELD mounting medium with 4′,6-diamidino-2-phenylindole (DAPI).

For immunohistochemistry for PV and mCherry/tdTomato, brain slices were rehydrated, permeabilized, and treated with 0.01 M sodium citrate for 10 min at 95–100 °C for antigen retrieval. The slices were then incubated with blocking solution (5% normal goat serum in PBST) for 1.5 h, followed by primary antibody incubation overnight at 4 °C using anti-PV guinea pig polyclonal antibody (1:500; 195004, Synaptic Systems) and anti-DsRed rabbit polyclonal antibody (1:500; 632496, TAKARA). The next day, slices were washed three times with PBST and then incubated with secondary antibodies (1:1000, Alexa Fluor 647 goat anti-guinea pig IgG, A-21450, Thermo Fisher Scientific; 1:750, Alexa Fluor 594 donkey anti-rabbit IgG, A-21207, Thermo Fisher Scientific) for 2 h at room temperature. The slices were washed three times with PBST again and then mounted with VECTASHIELD mounting medium with DAPI.

The brain slices were imaged in the high-throughput slide scanners (VS120, Olympus) for further processing. We also imaged selected example slices under a confocal microscope (Olympus FV-3000).

To analyze the distribution of RV-labeled neurons in the contralateral hemisphere, one of every three brain slices of the whole brain was imaged in the high-throughput slide scanners, and a custom-written software package was used to process the digitized brain images. The analysis software consists of four modules: atlas rotation, image registration, signal detection, and quantification/visualization. The detailed method has been described previously[38,56]. Briefly, we rotated the 3D Allen Mouse Brain Atlas to mimic the aberrant sectioning angle of the experimental brain. The digitized brain images were then aligned to the rotated 3D reference altas. In each digitized brain image, the position of manually identified RV-labeled neurons was recorded using the detection module. After detection and registration, signals were quantified across the whole brain and projected to the 3D reference atlas for better visualization. Since the number of labeled neurons varied across brains, the input from each contralateral region was quantified by dividing the number of labeled neurons found in that region by the total number of labeled neurons detected in the entire contralateral hemisphere.

**Behavior**
We trained head-fixed mice on a 2AUC change-detection task adapted from Burgess et al.[34]. Head-fixed mice were placed on a plastic apparatus with forepaws on a rotating wheel surrounded by three gamma-corrected 7-inch LCD screens (Viltrox, DC-70II, 60 Hz refresh rate), covering 230 × 42 degrees of visual angle (d.v.a.). Each screen was roughly 11 cm from the mouse's eyes at its nearest point. The rotating wheel is a ridged rubber wheel affixed to a rotary encoder (RotaryEncoder, GTS06-LD-RA360A-2M). A reward delivery spout was positioned near the snout of the mouse. Licks were detected by a capacitive-sensing circuit board connected to the spout. Experiments were controlled by custom-written code in MATLAB (Mathworks, Inc.).

Before the initial training, the mouse was implanted with head plates and allowed for at least 1-week recovery with free access to food and water. Subsequently, water restriction was applied that lasted till the training ended. The weight of the mouse was measured daily during water restriction. Mice had access to water only during training, but additional water was given if necessary to ensure that their body weight did not drop below 80% of the starting value. The training session for each mouse is about 1 h per day.

Mice were trained with the following protocol. Behavioral training started with habituation (2–3 d), during which there was no visual

stimulus, and mice were head-fixed in the rig with their forepaws resting on the wheel. RSP- and PTLp-lesion groups were then trained with learning phase1. Other mice were trained with learning phase1, 2 and test phase. High contrast visual stimuli (at 100% contrast level) were presented in learning phases. Test phase included different contrast levels. Trials of varying contrast conditions were randomly interleaved.

**Learning phase1 (10–12 days).** Mice were trained with alternative left-go and right-go blocks, in which the Go cue (drifting grating) was presented only on the left and right screen, respectively. Each block contained 30 trials. A trial was initiated after the mouse had held the wheel still for a short interval (0.2 s). At trial initiation, dynamic visual white noise was presented at the center of the left and right screens, located in the monocular zones of the mouse's visual field (62–95 deg lateral to vertical meridian, 7 deg below and 35 deg above horizontal meridian). After a random delay interval of 2.3–2.8 s, the white noise was replaced by the Go cue (sinusoidal drifting grating, orientation 0°, direction 90°, 2 Hz, 0.04 cycles/°) on either the left or right screen. As soon as the Go cue presentation, the position of the drifting grating is coupled to the wheel's movements. Mice were trained to turn the wheel to bring the drifting grating to the center of the middle screen for water reward in an 8-s response window (left-go block, clockwise turn 40°; right-go block, counterclockwise turn 40°; approximately 24 mm of movement of the surface of the wheel). If correct (hit), the water reward (2–3 μL) was delivered together with an auditory tone cue (6 kHz pure tone for 0.1 s), and the visual stimuli remained on the screen for 1 s. If incorrect (miss), the gray screens were presented after the response window. After a subsequent inter-trial interval of 3–4 s, the mouse could initiate another trial by holding the wheel still for the prescribed duration.

**Learning phase2 (7–10 days).** After mice mastered the task in the learning phase1 (ACA-lesioned group, ipsilesional hit rate >80%; other groups, hit rate on both sides >80%), we shortened the response window to 2 s and introduced high contrast no-go trials. Mice were trained with alternative left and right blocks. Each block contained 30 trials. In left block, left-go and no-go trials were randomly interleaved (2/3 left-go and 1/3 no-go). In right block, right-go and no-go trials were randomly interleaved (2/3 right-go and 1/3 no-go). In no-go trials, dynamic visual white noise was presented at the center of the left and right screens from the trial start to the end of the response window. If mice hold the wheel until the end of the response window (correct rejection, CR), the subsequent trial could be initiated after a 3–4 s inter-trial interval. If mice turned the wheel during the response window (false alarm, FA), feedback was a punishment sound cue (white noise sound) played for 1 s and a timeout for 1–2 s, after which the subsequent trial could be initiated after a 3–4 s inter-trial interval.

**Test phase (1–2 weeks).** After mice reached hit rate on both sides >80% and CR rate >60% for 3 days, we introduced low-contrast stimuli. Seven contrast levels were included for obtaining a psychometric function (1.6%, 3.1%, 6.2%, 12.5%, 25%, 50%, 100%). Two contrast levels were included for calcium imaging and optogenetic manipulation (12.5% and 100%). In each go trial, the visual white noise and the drifting grating had the same contrast levels. For calcium imaging, we increased the inter-trial interval to 6–7 s to avoid the calcium signal induced by trial outcomes superimposing on the calcium signal induced by the initiation of the next trial. For examining the effects of eOPN3-induced inactivation of CPN callosal-projection axons on visual-change detection, we increased the inter-trial interval to 10 s to allow the full recovery of synaptic transmission of callosal-projection axons after 4-s eOPN3 activation.

## In vivo calcium imaging and data analysis

We performed calcium imaging using a custom-built wide-field one-photon fluorescence microscope equipped with an objective lens (×10, NA 0.3, RMS10X-PF, Olympus), an LED fluorescent light source (470 nm), a LED driver, and a GFP filter set. The implanted GRIN lens relaid the excitation light to reach neurons of interest. The emission fluorescence signals coming back through the same GRIN lens were collected by the objective and recorded by a CCD camera (Qimaging, Retiga R1), which was controlled by Micro-manager software with an acquisition rate of 4 Hz.

The acquired images were first spatially downsampled by a factor of 4. Image stacks were then corrected for lateral motion using the rigid-affine algorithm in ANTs (http://stnava.github.io/ANTs/). We then used the CNMF-E approach to extract the average fluorescence for each ROI ($F_{corrected}$) (https://github.com/zhoupc/CNMF_E). $\Delta F/F$ was calculated as $\Delta F/F(t) = (F_{corrected}(t) - <F>)/<F>$, where $<F>$ is the average fluorescence across the entire recording. $\Delta F/F$ traces were Z-scored for further analysis.

To assess whether a neuron was significantly modulated by the task, we performed a four-way repeated ANOVA with factors trial type (go versus no-go trial), action (clockwise turn, counterclockwise turn, and no turn), contrast level (100% versus 12.5%) and epoch [baseline (1 s before visual cue start), visual stimulus1 (for go trials, from the start to end of visual white noise; for no-go trials, 2.5 s following the start of visual white noise), visual stimulus2 (for go trial, from the start to end of drifting grating; for no-go trial, from 2.5 s after the start of visual white noise to the end of visual white noise), and outcome (2 s following the water or punishment sound cue delivery)]. In each cell, the peak response of z-scored $\Delta F/F$ in each epoch was used for ANOVA analysis. A neuron was deemed significantly modulated if $P < 0.01$ for at least one of the factors or interaction terms.

To assess whether a neuron responded differently to high and low contrasts in hit trials, we performed a two-way repeated ANOVA with factors trial type (left-hit versus right-hit trial) and contrast level (100% versus 12.5%). In each cell, the peak response of z-scored $\Delta F/F$ during the visuomotor period was used for ANOVA analysis. A neuron was deemed significantly modulated by contrast levels if $P < 0.01$ for the factor contrast or the interaction term.

To determine whether a neuron was differentially modulated in contralateral-hit and ipsilateral-hit trials, we performed Student's $t$-test for the peak response of z-scored $\Delta F/F$ during the visuomotor period for each trial type. A neuron was considered 'contralateral-preferring' if its activity was significantly higher in contralateral-hit trials than in ipsilateral-hit trials ($P < 0.05$), or 'ipsilateral-preferring' if its activity was significantly higher in ipsilateral-hit trials ($P < 0.05$).

To assess whether the averaged population responses of contralateral-preferring and ipsilateral-preferring neurons were differentially modulated in contralateral-hit vs. ipsilateral-hit, contralateral-miss vs. ipsilateral-miss, and contralateral-FA vs. ipsilateral-FA trials, we conducted Student's $t$-tests on the peak response of z-scored $\Delta F/F$ during the visuomotor period in hit trials, during the presentation of drifting gratings in miss trials, and during the wheel rotation in FA trials.

## In vivo optogenetic manipulation

Optical activation of ChR2 was induced by blue light. A blue laser (473 nm, Shanghai Laser) was connected to an optic fiber (200 μm in diameter, ThorLabs) and controlled by an Arduino board. To activate ACA CPNs, we used pulse trains (10 Hz, 5 ms per pulse) of laser at a power of 1–3 mW at the fiber tip. To activate PV+cal neurons, we used pulse trains (10 Hz, 5 ms per pulse) of laser at a power of 1–5 mW at the fiber tip. To activate PV+ neurons, we used pulse trains (10 Hz, 5 ms per pulse) of laser at a power of 3–8 mW at the fiber tip. For the behavioral experiments, blue laser stimulation was delivered in odd trials of each

session. The light started with visual white noise onset and continued to the end of the response window in each trial.

Optical activation of NpHR was induced by yellow light. A yellow laser (589 nm, Shanghai Laser) was connected to an optic fiber (200 μm in diameter). To inactivate ACA CPNs, we used a constant laser at a power of 3–8 mW at the fiber tip. For the behavioral experiments, yellow laser stimulation was delivered in odd trials of each session. The light started with visual white noise onset and continued to the end of the response window in each trial.

Optical activation of eOPN3 was induced by green light. A green laser (532 nm, Shanghai Laser) was connected to an optic fiber (200 μm in diameter). To inactivate the callosal-projection axon terminals from ACA CPNs, we used a constant laser at a power of 5 mW at the fiber tip. For the behavioral experiments, green laser stimulation was delivered in odd trials of each session. The light started with visual white noise onset and continued to the end of the response window in each trial. Based on the temporal dynamics of eOPN3-induced inactivation of callosal-projection axons, the inter-trial interval was increased to 10 s.

### In vivo chemogenetic manipulation

For chemogenetic activation of PV+ neurons in the ACA, the DREADD agonist clozapine N-oxide (CNO, Sigma-Aldrich, 1 mg/mL, 300 nL) was intracranially microinjected into the ACA through implanted cannula at least 30 min before the behavioral test. Animals were anesthetized (isoflurane, 1.5–2%) during the injection and then returned to home cages until the behavioral test.

### Slice preparation and recording

Mice were anesthetized with 5% isoflurane. After decapitation, the brain was dissected rapidly and placed in ice-cold oxygenated NMDG-HEPES solution (in mM: NMDG 93, KCl 2.5, $NaH_2PO_4$ 1.2, $NaHCO_3$ 30, HEPES 20, glucose 25, sodium ascorbate 5, thiourea 2, sodium pyruvate 3, $MgSO_4 \cdot 7H_2O$ 10, $CaCl_2 \cdot 2H_2O$ 0.5 and NAC 12, at pH 7.4, adjusted with HCl), and coronal sections of brain slices were made with a vibratome. Slices (300 μm thick) were recovered in oxygenated NMDG-HEPES solution at 32 °C for 10 min and then maintained in an incubation chamber with oxygenated standard ACSF (in mM: NaCl 125, KCl 3, $CaCl_2$ 2, $MgCl_2$ 1, $NaH_2PO_4$ 1.25, sodium ascorbate 1.3, $NaHCO_3$ 26, glucose 10) at 30 °C for 1–4 h before recording.

Whole-cell recordings were made at 30 °C in oxygenated standard ACSF. EPSCs and IPSCs were recorded using a cesium-based internal solution (in mM: $CsMeSO_4$ 125, CsCl 2, HEPES 10, EGTA 0.5, MgATP 4, $Na_2GTP$ 0.3, $Na_2$-phosphocreatine 10, TEACl 5, QX-314 3.5, at pH 7.3, adjusted with CsOH, 290–300 mOsm) and isolated by clamping the membrane potential of the recorded neuron at the reversal potential of inhibitory and excitatory synaptic currents, respectively. EPSPs were recorded using a potassium-based internal solution (in mM: K-gluconate 135, KCl 5, HEPES 10, EGTA 0.3, MgATP 4, Na2GTP 0.3, and Na2-phosphocreatine 10, at pH 7.3, adjusted with KOH, 290–300 mOsm). For measuring the monosynaptic callosal inputs to ACA neurons, TTX (1 μM) and 4-aminopyridine (100 μM) were bath applied to block action potentials and permit direct depolarization of axon terminals by ChR2 activation with 5-ms pulses of blue light[58]. To compare the response among different classes of interneurons, the EPSP amplitude recorded from interneurons was normalized by the EPSP amplitude from Pyr neurons in the same layer (Fig. 7F).

To activate ChrimsonR, we used X-cite LED (Lumen Dynamics Group), which was controlled by a stimulator (Master8) and bandpass filtered at 610–650 nm (Semrock). Pulse trains of red light (10 Hz, 5 ms) were delivered through a 40× 0.8 NA water immersion lens at a power of 3 mW. To activate eOPN3, we used a green laser (532 nm, Shanghai Laser), which was controlled by a stimulator (Master8). Constant green light was delivered through an optic fiber (200 μm in diameter) at a power of 5 mW at the fiber tip. To activate ChR2, we used X-cite LED

(Lumen Dynamics Group), which was controlled by a stimulator (Master8) and bandpass filtered at 419–465 nm (Semrock). Pulse trains of blue light (10 Hz, 5 ms) were delivered through a 40× 0.8 NA water immersion lens at a power of 1–2 mW. The resistance of the patch pipette was 3–5 MΩ. The cells were excluded if the series resistance exceeded 40 MΩ or varied by more than 20% during the recording period. Data were recorded with a Multiclamp 700B amplifier (Axon Instruments), filtered at 2 kHz, and digitized with a Digidata 1322 (Axon Instruments) at 10 kHz. Recordings were analyzed using custom software.

### Behavioral data analysis

For left-go and right-go blocks in learning phase1, hit rate was quantified as follows:

left-hit rate = number of left-hits/(number of left-hits + number of left-misses)

left-hit + left-miss = 100%

right-hit rate = number of right-hits/(number of right-hits + number of right-misses)

right-hit + right-miss = 100%

For test phase, hit, FA, and CR rates at each contrast level were quantified as follows:

left-hit rate = number of left-hits/(number of left-hits + number of left-misses)

left-hit + left-miss = 100%

right-hit rate = number of right-hits/(number of right-hits + number of right-misses)

right-hit + right-miss = 100%

left-FA rate = number of left-FAs/number of No-go trials

right-FA rate = number of right-FAs/number of No-go trials

CR = number of CRs/number of No-go trials

left-FA + right-FA + CR = 100%

### Quantification and statistical analysis

ANOVA was performed using SPSS, and other statistical analyses were performed using MATLAB. The selection of statistical tests was based on previous studies. All statistical tests were two-sided. The exact number of mice and recorded cells were described in figure legends. Data were reported as the mean ± SEM in all figures. Statistical method, statistics, and corresponding *P* values were reported in the figure legends. Source data are provided with this paper.

### Reporting summary

Further information on research design is available in the Nature Portfolio Reporting Summary linked to this article.

## Data availability

All data needed to evaluate the conclusions in the paper are present in the paper and the Supplementary Materials. Data related to this paper may be requested from the authors. The Allen 3D mouse brain atlas, which is used for anatomical data analysis, can be accessed at mouse. brain-map.org and atlas.brain-map.org. Source data are provided with this paper.

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

## Acknowledgements

We thank L. Wang, T. Yang, N. Xu, S. Kuai and Z. Liang for critical reading of our manuscript, and Z. Xiao for the help with data analysis. This work was supported by grants from STI2030-Major Projects (2021ZD0202804 and 2021ZD0203704 to S.Z.), National Natural Science Foundation of China (32170993 to S.Z. and 32200811 to Y.L.), the Science and Technology Commission of Shanghai Municipality (21ZR1436400 to S.Z.), China Postdoctoral Science Foundation (2022M722135 to Y.L.) the innovative research team of high-level local universities in Shanghai, and the Shanghai Frontiers Science Center of Cellular Homeostasis Regulation and Human Diseases.

## Author contributions

Yanjie Wang, M.X., and S.Z. designed the experiments. Yanjie Wang and Z.C. performed behavioral experiments with assistance by M.Q. G.M. performed electrophysiological experiments on brain slices. L.W. performed some behavioral experiments. L.W. and Y.L. performed RV-tracing experiments. X.F. and Yifan Wu assisted in data analysis. Yanjie Wang, C.Z., G.M., M.X., and S.Z. wrote the manuscript.

## Competing interests

The authors declare no competing interests.
