## [Peer Review File · Nature Communications]

A Frontal Transcallosal Inhibition Loop Mediates Interhemispheric Balance in Visuospatial ProcessingREVIEWER COMMENTS

Reviewer #1 (Remarks to the Author):

This paper explores the role of cortical neurons that project across the callosum and argues that they mediate a form of competition between the two hemispheres. The study argues for a transcallosal competitive network, where “callosal projecting neurons” (CPNs) in one hemisphere preferentially contact inhibitory neurons of the PV type in the other hemisphere, thus suppressing excitatory neurons in that hemisphere. The study argues that enhancing transcallosal inhibition can even help in the case of unilateral lesions such as those experienced by stroke patients.

Many aspects of the study are praiseworthy. For instance, the study uses a variety of techniques, from controlled behavior to circuit mapping to in vitro assays. The behavior is elegant, as it allows one to distinguish errors of two types: moving / not moving the wheel at the wrong times (false alarms and misses) and moving the wheel in the wrong direction (incorrect choices, which can provide very useful information if analyzed). The transcallosal viral strategies are also elegant.

MAJOR COMMENTS

The main weak point is that the chain of reasoning is strained. In various cases the effects are more complex than they are described, but in the interest of making the story work, they are summarized as being “preferentially” in one direction. For instance, in Figure 2, the ACA neurons seem to be as modulated by contralateral stimuli as by ipsilateral stimuli, but they are described as being mostly contralateral. Similarly, in Figure 5, all inhibitory neurons (not just PV cells) get contralateral excitatory input. This happens multiple other times in the paper: often a categorical conclusion is implied by describing an effect as “preferential” when in fact the results are much more nuanced. And a result that seems problematic is not discussed: shutting down both sides of ACA does not affect behavior (Figure 8 d,e).

Another main criticism concerns the statistics. It is not clear if the statistics in the paper consider the independent measurement to be the session or the mouse. It should be the mouse. For instance, in the caption to fig 1j it says there are 11 mice and 69 sessions. If the statistics assume $n = 69$, that’s a mistake: the unit of independent measurement here is the mouse, so $n = 11$. Sessions in the same mouse are not independent measurements. Similarly, Figure 3 shows one bar per session (29 sessions). But the unit of independent measurement is the mouse (14 mice), not the session. Statistics should be done across mice, not sessions. If desired, one can use a nested model.

Furthermore, the paper does not seem to show responses of neurons when the mouse makes the incorrect decision, moving the wheel in the wrong direction. And yet this would be a useful thing to see, because it would help show if the responses are related to the stimulus or to the movement or to both. There should be many such trials at low contrast.

Finally, the paper should show more results of anatomical reconstructions (histology), to show us locations of lesions, placements of GRIN rods, etc.

DETAILED COMMENTS

Introduction. The introduction is a bit verbose. It devotes almost 2 pages (the entire page 5 and half of page 6) to summarizing the results of the paper. This is usually done in a brief paragraph. (On the other hand, the first paragraph of Results should really be in Introduction.)

Figure 1. It would help to see a picture of the cortex from the top, indicating the exact locations of the 3 areas that were lesioned. This would be particularly useful for ACA, which is a vast region. The figure could indicate the part of ACA that is the subject of the experiments in this figure and in subsequent figures.

Figure 2. The GRIN rod was presumably used to go deeper and reach ACA. But how deep? (apologies if this is already in Methods – I could not find it). Please show the histology in a supplementary figure.

Crucially, there seem to be almost as many ipsi-preferring neurons as contra-preferring. So why does the paper place so much emphasis on the former?

In the pie charts, not sure what “visuomotor association” stands for. Does it mean that the neuron encodes both visual stimuli and movements?

Figure 3. Here and in other figures that show the effects of laser stimulation, it would be good to have a consistent format. For instance, one could adopt the single, simplified format like the one in Figure 4e,f. The detailed breakdown session by session should definitely be in the paper, but perhaps it can be in Supplementary Figures.

Figure 5. This seems to be another case of overreach in terms of conclusions: the figure shows that all inhibitory neurons (not just PV cells) get contralateral excitatory input .

Figure 6. This figure shows that PV cells get particularly strong EPSCs from contralateral CPNs. But perhaps they get strong EPSCs from anyone, including ipsilateral CPNs? If so, it's another doubtful use of “preferential”.

Figure 7. Panel jkl : isn't it expected that PVcal neurons would receive more input from contralateral side than other PV neurons? They are called PVcal because they receive contralateral input. It would be good to clarify this matter.

Figure 7. This figure shows the activity of PVcal neurons and demonstrates convincingly that they are more ipsi-preferring. However, it does not show us what those neurons do in the incorrect trials. Showing this would help us distinguish visual responses from motor responses.

Figure 8. This figure could open with the effects of activation of PV cells during the task, which are currently in Supp Fig 11. They could be brought to the main text if the graphics are simplified (using the format of Figure 4e,f, for figures in main text).

It is surprising that there is no effect of bilateral PV+ activation (Figure 8 d,e). And yet, it must shut down both sides of ACA. Please explain in the paper how this can be the case (if there is an explanation).

OTHER COMMENTS

Style. Some statistics seem to appear both in main text and figure legends, sometimes with subtle variations. This is confusing. It would be best to choose one place and be consistent about it.

page 8: "indicating a role for the ACA in mediating interhemispheric visuospatial competition." This sentence is premature because at this point we don't know that there is competition between the two stimuli, and that the competition between stimuli involves a competition between hemispheres.

Page 21 "which has been shown to suppress pyramidal neuron spiking". The method of shutting down cortex by activating PV cells was demonstrated in Supplementary Figure 8 of this paper: Olsen, S. R., D. S. Bortone, H. Adesnik and M. Scanziani (2012). "Gain control by layer six in cortical circuits of vision." *Nature* 483(7387): 47-52.

The literature contains studies where frontal cortex does not appear to be lateralized (work by the Svoboda laboratory) and studies where it does seem to be lateralized (e.g. this paper, and Zátka-Haas et al, *eLife* 2021). It would be good to clarify where results agree and disagree with these and other previous studies.

Typos: in page 12, Fig 6a and Fig 6b should be Fig 4a and Fig 4b

Reviewer #2 (Remarks to the Author):

NCOMMS-23-04432-T

A Frontal Transcallosal Inhibition Loop Mediates Interhemispheric Visuospatial Competition

Wang et al.

General Comments

While I can appreciate that brevity is often welcomed, this should not be at the expense of clarity. The authors often employ forms of expression, particularly in describing results, in which terms such as

“ipsilateral” and “contralateral” are used in a “shorthand” manner. Although the intended interpretation may be apparent to the authors, this will not be the case for many readers. It will therefore be necessary for the authors to expand several descriptions, to reduce the burden on the reader, and remove the scope for ambiguity.

There is a good deal of repetition in the Introduction and Results sections. The former contains material that I would have expected to see in the Results section. The latter contains material that would be more appropriately located in the Introduction section.

In many ways, the reporting of the results of the statistical tests does not accord with the indications provided in the Reporting Summary. For example, neither the test statistics nor the associated degrees of freedom are provided. Effect sizes are not given. The Reporting Summary also appears to stipulate that the standard deviations, rather than the standard error) should be reported. This has not been done.

Specific Comments

Abstract

1) The authors state that in the context of a change-detection task “ACA callosal projection neurons (CPNs) were more active with contralateral changes than with ipsilateral changes” Without prior knowledge of this specific paradigm, it will not be possible for the reader to appreciate the dimension along which “contralateral changes” are differentiated from “ipsilateral changes”. The text should be revised to make transparent the nature of the behavioural phenomenon that is manifest in a change-detection task and, in particular, how “contralateral” and “ipsilateral” are to be understood in this context.

2) There is scope to enhance the precision of the writing, particularly with respect to statements such as “improved ipsilateral detection via callosal projections”, which imply a lack of conceptual clarity. In this instance, “ipsilateral detection” is a behavioural phenomenon, whereas “via callosal projections” is a statement concerning anatomy. That which is omitted is reference to the process that relates the two. Although treatment of process follows, the authors should give further consideration to the sequence in which information is provided, and make a greater effort to provide explanation, rather than a set of observations, with the reader being left to expend too much effort in order to draw the intended inferences. In this context, the concluding statement cannot be seen to follow in an obvious way from the preceding material.

Introduction (text commencing on page 3)

3) The opening statement, “Animals receive a large amount of environmental information at any given moment”, implies a particular conception of the relationship between perception, action (and

cognition), in which the organism is a passive recipient (rather than there being an active reciprocal relationship between perception and action). While it may be the case that the authors indeed adhere to the view that animals “receive a large amount of ... information” is the appropriate way in which to conceive of perception, they should perhaps consider that many readers will not share this view, and that the opening statement will thus have a rather jarring effect.

4) (page 3, para 1). Some further work is required to explain the meaning of “along an anterior-posterior gradient” in the context of “callosal fibers mostly project to homotopic regions of the hemispheres”. Are the authors implying that the density (or some other property) of the callosal fibres varies along an anterior-posterior axis? If so, the property in question, and the polarity of the variation, should be stated.

5) (page 3, para 1). The expression “interhemispheric facilitation from redundant information” is cryptic and fails to convey useful information.

6) (page 3, para 1). In a similar vein, the expression “interhemispheric visuospatial competition” could be interpreted in very many ways. Greater precision is required.

7) (page 3, para 2). Rather than using terms such as “right-sided sensory inputs” and “left-sided sensory inputs”, perhaps the authors might refer to visual fields.

8) (page 4, para 2). Since it is central to the entire endeavour, it is essential that the authors provided a clear description of the defining features of interhemispheric visuospatial competition”, upon first use of this term. It cannot be assumed that the characteristics of this phenomenon will be known to most readers.

This is a particular requirement, since the reader is then informed that a consequences of “unilateral damage of the parietal and frontal cortices” is to “bias visuospatial competition”.

9) (page 4, para 2). The description of the behavioural consequences is fair enough (i.e., “... even leading to neglect of visual stimuli on the contralesional side ...”, however, appended to this is an assertion with respect to process (“due to the interrupted top-down control of contralesional visual information processing”), which is not argued, and cannot be seen to follow from the characteristics of the behavioural phenomenon.

Indeed, it is not clear why this is necessary, given that follows is a more clearly specified proposition that relates to, “impaired mutual transcallosal inhibition” (which is much more closely related to the study that is then described”.

10) (page 4, para 3). The authors then go on to state that, “we examined the transcallosal synaptic circuits mediating interhemispheric visuospatial competition in the mouse brain”. How does this relate to the behavioural consequences of “unilateral damage of the parietal and frontal cortices” just described? The key problem is that the authors have not defined the nature of interhemispheric visuospatial competition, nor have they made clear the hypothesised relationship between unilateral

damage of the parietal and frontal cortices and interhemispheric visuospatial competition. The logic of the implied association may be apparent to the authors, however, it is not explained to the reader.

11) (page 5, para 1). It does make sense to assert that the cingulate cortex (ACA) is, within the mouse frontal cortex, the “the region with the most severe detected visuospatial bias following unilateral lesion”. Visuospatial bias is a behavioural phenomenon that is defined in terms of choices made by an animal. It is not a property of brain regions. The authors should give greater consideration to the manner in which they conceive of and describe the mechanisms that mediate the behavioural phenomena under consideration.

12) Even though a larger number of words will be required to make the same point, I would urge the authors to rephrase/expand upon statements such as “were more active with contralateral changes”. For example, it might be stated that “ACA CPNs were more active when the stimulus that changed was located in the contralateral visual field”. This may seem a pedantic point, however, the aim should be to eliminate ambiguity. This point applies to all statements of a similar nature made throughout the manuscript.

13) (page 5, para 2). Please expand the statement, “callosal-input-driven PV+ (PV+cal) neurons preferentially inhibit ipsilateral CPNs”, to make the meaning of “inhibit ipsilateral” entirely unambiguous.

15) (page 5, para 2). As per previous comments, while I can appreciate that the authors are seeking economy of expression, this should not be at the expense of clarity. Please rephrase, “were more active with ipsilateral changes than with contralateral changes”. In this case, how is the reader to understand the meaning of “ipsi-preferring neurons”? This is not a statement with physiological meaning.

Results

16) (page 6). Please “speak” more plainly. The precise intended meaning of statements such as “disrupt the balance of interaction” and “biased the visuospatial competition” cannot be determined.

17) (page 8). The meaning of the “±” and the figures that follow is nor made apparent. Do the values represent the range, the standard deviation, the standard error, the confidence interval)?

18) (page 8). Please provide the test statistic (e.g., F ratio), the associated degrees of freedom, and the exact associated p value. Ideally, an effect size statistic should be provided along with the corresponding confidence interval. This information was not provided in the Supplementary Materials. In the absence of such, it is not possible to determine the veracity of the statistical analyses. It is simply not sufficient to provide only probability values.

I note that the reporting of the results of the statistical tests does not accord with the indications provided in the Reporting Summary (which also appears to stipulate that the standard deviations, rather than the standard error) should be reported.

19) (Figure 1 and supplementary figures). It is recommended that the error bars represent standard deviations, rather than standard error of the means. The Reporting Summary appears to indicate that this is a requirement.

20) In some cases, the number of animals included was extremely small. For example, in the legend to Fig 3, it is stated that, “n = 14 mice” and “n = 8 mice”. In Fig 4, it is “n= 5 mice”. In Fig 5, “(Pyr, n = 3 mice; PV, n = 5; SST, n = 3; VIP, n = 5)”. In Extended Data Fig. 2, it is stated that “n = 8 mice”, in Extended Data Fig. 5, it is “n = 6 mice”, in Extended Data Fig. 6, it is “n = 3 mice”, in Extended Data Fig. 10, it is “n = 7 mice”, in Extended Data Fig. 11, it is “n = 6 mice”. While it can be appreciated that there is a general desire to minimise the number of animals used in laboratory research, the challenges to replicability that arise from small sample sizes, have been well documented.

Reviewer #3 (Remarks to the Author):

Review of manuscript NCOMMS-23-04432-T

The present manuscript describes experiments manipulating callosal projecting neurons (CPN) and their targets in the anterior cingulate area (ACA) of awake mice through optogenetic excitation and inhibition while performing a two-alternative unforced-change detection task involving a motor response (turning a wheel) to indicate the discrimination of gratings versus noise stimuli placed in either of the two monocular visual fields of the mouse.

Besides the behavior, the activity of the neurons of the awake animal was monitored by calcium-imaging to demonstrate that ACA-CPNs are indeed associated with the visuomotor task. Unilateral ACA lesion or inhibition of ACA-CPN impaired contralateral visual change detection (independent of the contrast, i.e. visual neglect) and while improving ipsilateral change detection indicative of release of transcallosally mediated inhibition during this manipulation.

Using transmonosynaptic viral tracing, the ACA-CPN were found to project preferentially (60 % of the contralateral targets) to the contralateral ACA innervating both excitatory and inhibitory targets, i.e. pyramidal neurons, parvalbumin-positive (PV+), somatostatin-positive (SST+) and vasoactive intestinal peptide-positive (VIP+) neurons.

Subsequently, a CPN driven loop activating contralateral PV+ neurons which in turn inhibit ipsilateral excitatory CPNs was revealed by cell-type specific dissection of synaptic circuits in coronal brain slices. Conformingly, unilateral activation of PV+ neurons in the ACA had similar behavioral effects as the unilateral inhibition of the excitatory CPNs, namely the contralateral neglect-like visuospatial impairment, a bias which was eliminated during bilateral PV+ neuron activation. Finally, a short-term activation of the contralateral PV+ neurons during learning of the change detection task improved the

performance of the ACA-lesioned animals both during and after learning.

The authors conclude that they challenged and identified with their experiments a transcallosal loop including ipsilateral excitatory neurons in the anterior cingulate cortex impinging directly and indirectly on contralateral PV+ interneurons which modulate in turn the excitatory ipsilateral tonus and thus again the contralateral inhibition. They assume that the mouse ACA circuit provides top-down modulation on visual cortices and mediates interhemispheric visuospatial “competition” via the demonstrated transcallosal inhibitory loop.

The study is methodologically sound, comprehensive and elegant, the results are convincing and well documented as far as they concern the involvement of the mouse anterior cingulate cortex in visuomotor integration or visuospatial attention. The intended link to interhemispheric competition in humans might not be appropriate.

Major critiques

1. Title, abstract and following text. “...visuospatial competition”. I would not use the word competition, neither in the title nor in the abstract or discussion.

The authors claim to identify a loop mediating visuospatial competition but their task is actually not suited to characterize competition between the two hemifields. In my view, that would be a task demonstrating that two visual stimuli of equal salience (i.e. grating of the same SF and contrast) (such as in binocular rivalry, which does not exist here) compete for visual attention, or because of being represented in different hemispheres (such as in the Stroop-match-to sample task where the hemispheres are supposed to be in a “competitive” situation).

Being trained to detect a grating and rotate a wheel, I rather assume that the mouse deals with top-down visuomotor integration and visuospatial attention, and the loop which is identified is a reciprocal callosal excitatory-inhibitory loop.

The idea of visuospatial competition is driven from the human point of view who has access to the same central visual information in both hemispheres. The mouse might not even have this, and moreover, the task actively avoids the tiny rodent binocular visual field. It is thus possibly better to speak of interhemispheric cooperation (involving transcallosal inhibition) or excitatory-inhibitory balance instructing downstream motor, extrastriate and subcortical areas.

Most of the references in the introduction about an assumed visuospatial competition are rather general and human-specific and do not introduce what is known about the role of the anterior cingulate cortex in mice.

Citations 10-12 do not fit for “visuospatial competition” either. Note that the almost identical argument is repeated in the beginning of the discussion on page 22.

2. Page 3, last paragraph: I suggest to reformulate the sentence “... enhance visual information processing ...” to “Recent studies have shown that also in rodents V1 callosal projections contribute to visual information processing....”.

First of all functions, connections between the primary visual cortices in lateral-eyed mammals such as mice most likely and inevitably link the two halves of the visual field, because - in contrast to humans - more than 90% of the retinal fibres, i.e. the primary visual input, cross.

Second, noteworthy, many of the visual callosal functions listed here as specific for rodents, and more, have been detailed in frontal-eyed mammals such as cats and monkeys previously. Strangely, sometimes these studies are not put in the right context or not even cited in the more recent rodent studies, although they are much more similar to the situation in humans.

In this manuscript, it is important to identify and discuss the possible differences between the visual system of rodents and humans, and their implications for the present findings. Visual input from monocular and binocular fields in rodents and humans might not be of similar behavioral significance. It is likely that mice might not use the same visual strategies as primates. For one, rodent V1 probably receives already much more multisensory input.

There exists the possibility that the slight performance deficit in the monocular field of a mouse induced by manipulating mouse cingulate cortex does not have a human homologue. Maybe it is not accurate to compare it to the human visual neglect situation following stroke.

Minor critiques

- Line numbers would be useful

- In the first introduction of the abbreviation, you might want to state explicitly ACA for “anterior cingulate area”.

- Page 3, last paragraph: “...enhance the gain of the visual response”^{18,19,20}. For clarification, complete the sentence with “...enhance the gain of the visual response in function of the retino-geniculo-cortical input”.²¹. Cite here also Rochefort et al., 2007.

- From page 8 onward, the word “p-value” is written in different styles (italic) and sometimes P in capital letters, be consistent throughout.

- Figure 3b-d: Some of the effects look tiny. The traces of the off-on effects refer to a single hit rate for a single session of a single mouse? In the legend of this figure and the others where this applies, you might state on how many hit rates in total the ANOVAs are based on, and the effect size.

- Page 4, second line: “... to facilitate stereoscopic vision...” Do not confound binocular with stereoscopic vision. The cited paper 22 is not about the latter.

- Page 22: “..comprising..” reads “comprises”

- Page 24 and else: "...ameliorate the behavioral bias..." might be contextually wrong. "...reverse the inactivation/lesion-induced behavioral bias?"

- Page 24: "...support that reestablishing balanced interhemispheric interaction...after lesion". Cite Lomber and Payne, 1996, for a similar restoration of a behavioral unilateral visual neglect by experimental bilateral inactivation of extrastriate or subcortical areas in an awake animal.

Methods

- Page 11: state the direction of movement of the vertical ? grating. This might be relevant as rodents have directional biases in the visual system.

- Page 12: "... expert.."?

- Page 15: Arduino board

- Page 16: "through an implanted canular.."

Reviewer #1 (reviewer's comments in italic):

This paper explores the role of cortical neurons that project across the callosum and argues that they mediate a form of competition between the two hemispheres. The study argues for a transcallosal competitive network, where “callosal projecting neurons” (CPNs) in one hemisphere preferentially contact inhibitory neurons of the PV type in the other hemisphere, thus suppressing excitatory neurons in that hemisphere. The study argues that enhancing transcallosal inhibition can even help in the case of unilateral lesions such as those experienced by stroke patients.

Many aspects of the study are praiseworthy. For instance, the study uses a variety of techniques, from controlled behavior to circuit mapping to in vitro assays. The behavior is elegant, as it allows one to distinguish errors of two types: moving / not moving the wheel at the wrong times (false alarms and misses) and moving the wheel in the wrong direction (incorrect choices, which can provide very useful information if analyzed). The transcallosal viral strategies are also elegant.

MAJOR COMMENTS

1) The main weak point is that the chain of reasoning is strained. In various cases the effects are more complex than they are described, but in the interest of making the story work, they are summarized as being “preferentially” in one direction. For instance, in Figure 2, the ACA neurons seem to be as modulated by contralateral stimuli as by ipsilateral stimuli, but they are described as being mostly contralateral.

Similarly, in Figure 5, all inhibitory neurons (not just PV cells) get contralateral excitatory input. This happens multiple other times in the paper: often a categorical conclusion is implied by describing an effect as “preferential” when in fact the results are much more nuanced.

RESPONSE:

We now understand that we overly simplified the interpretation in the originally submitted manuscript and have changed the revised manuscript to present a fuller, more nuanced argument.

In Figure 2, we now add the analysis for the neuronal encoding at the population level. Although the single neuron activity was heterogeneous, the averaged responses of ACA CPNs were significantly stronger in the contralateral-hit trials than in the ipsilateral-hit trials during the visuomotor period, indicating that the overall strength of callosal inputs are stronger with contralateral visual change than with ipsilateral visual

change (Fig. 2f). We have revised the manuscript accordingly (P. 8, Lines 9-16):

“During the visuomotor period, 50% of ACA CPNs responded differentially in the contra-hit (visual change on the side opposite to recorded CPNs) vs. the ipsi-hit trials, with significantly more contralateral-preferring CPNs than ipsilateral-preferring CPNs ($P = 8 \times 10^{-5}$, chi-square test, Fig. 2e). Although the single neuron activity was heterogeneous, the averaged responses of ACA CPNs were significantly stronger in the contralateral-hit trials than in the ipsilateral-hit trials during the visuomotor period, indicating that the overall strength of callosal inputs is stronger with contralateral visual change than with ipsilateral visual change (Fig. 2f).”

“Fig. 2 | ACA CPNs show selective visual-related activity with a contralateral bias in the 2AUC change-detection task. a, Schematic diagrams of the viral strategy (top) and calcium imaging of CPNs (bottom). **b,** Raw fluorescence trace of an example CPN. Same behavior paradigm as the test phase shown in Fig. 1h, containing the trials at 100% and 12.5% contrast levels. Dark color, high contrast; light color, low contrast. Purple shading, visual white noise; red shadings, contralateral gratings; blue shading, ipsilateral gratings. Red and blue arrowheads, end of wheel rotation in the contralateral and ipsilateral hit trials, respectively. Red diamond, end of contralateral wheel rotation in the FA trial. Dashed line, end of timeout in the FA trial. **c,** Averaged $\Delta F/F$ (z-scored) of an example

CPN in hit trials. Colored shading, \pm SEM. Black dashed line, the start of visual white noise. Gray shading, window of the visuomotor period (from the start of drifting grating to the end of wheel rotation). **d**, Color-coded averaged $\Delta F/F$ (z-scored) of all recorded CPNs in contralateral-hit (left) and ipsilateral-hit (right) trials ($n = 8$ mice, 8 sessions, 396 neurons). Each row shows the average activity of one neuron, sorted by peak response during the visuomotor period. Black dashed line, the start of visual white noise. The left and right gray dashed lines present the start of drifting grating and the end of wheel rotation, respectively. **e**, Distribution of the difference in peak responses of z-scored $\Delta F/F$ during the visuomotor period between the contralateral-hit and ipsilateral-hit trials for the imaged CPNs. Blue, ipsilateral-preferring cells; red, contralateral-preferring cells. Orange arrowhead, median. There were significantly more contralateral-preferring CPNs than ipsilateral-preferring CPNs ($P = 8 \times 10^{-5}$, chi-square test). **f**, Similar to c, but for all recorded CPNs. The averaged responses of CPNs were significantly stronger in the contralateral-hit trials than in the ipsilateral-hit trials during the visuomotor period ($P = 0.001$, paired t-test test). **g**, Similar to d, but for contralateral-preferring CPNs. **h**, Averaged $\Delta F/F$ (z-scored) of contralateral-preferring CPN in different types of trials. Left, activity in hit trials. Black dashed line, the start of visual white noise. Gray shading, window of the visuomotor period (from the start of drifting grating to the end of wheel rotation). The activity of contralateral-preferring CPNs was significantly higher in the contralateral-hit trials than in the ipsilateral-hit trials during the visuomotor period ($P = 1 \times 10^{-17}$, paired t-test test). Middle, activity in miss trials. Dashed line, the start of visual white noise. Gray shading, window during the presentation of drifting grating. The activity was significantly higher in the contralateral-miss trials than in the ipsilateral-miss trials during the presentation of drifting grating ($P = 3 \times 10^{-8}$, paired t-test test). Right, activity in FA trials. Black dashed line, the start of visual white noise. Gray shading, window during the wheel rotation. The activity did not differ between the contralateral-FA trials and ipsilateral-FA trials during the wheel rotation ($P = 0.14$, paired t-test test). Colored shading, \pm SEM. **i,j** Similar to g,h, but for ipsilateral-preferring CPNs. The activity of ipsilateral-preferring CPNs was significantly lower in the contralateral-hit trials than in the ipsilateral-hit trials during the visuomotor period ($P = 8 \times 10^{-17}$, paired t-test test). Similarly, the activity of ipsilateral-preferring CPNs was significantly lower in the contralateral-miss trials than in the ipsilateral-miss trials during the presentation of drifting grating ($P = 6 \times 10^{-8}$, paired t-test test). However, there was no significant activity difference between the contralateral-FA trials and the ipsilateral-FA trials during the wheel rotation ($P = 0.10$, paired t-test test). See also Extended Data Fig. 4,5 and Methods.”

Related to Fig.5, RV tracing is a powerful tool for visualizing the distribution of inputs to specific neuronal populations but lacks information about the synaptic properties of labeled connections. Based on the RV-tracing results in Fig.5, we concluded that “both excitatory neurons and three major types of inhibitory neurons in the ACA are directly innervated by callosal inputs from the contralateral ACA” (P. 13, Lines 15-16). Next, we examined the ACA callosal input strength on the inhibitory neurons and found that the PV+ neurons received the strongest callosal input among examined excitatory and inhibitory neurons (Fig. 6f). Notably, the callosal input strength on PV+ neurons was more than 2.8 fold larger than on SST+ and VIP+ neurons at each layer (**Reviewer Fig. 1**), indicating that callosal inputs weakly activate SST+ and VIP+ neurons and strongly activate PV+ neurons. To improve the precision of the conclusion, we dropped “preferentially activate PV+ neurons” and replaced with “weakly activate SST+ and VIP+ neurons and strongly activate PV+ neurons” (P. 14, Lines 19-20).

I would like to take this opportunity to elaborate a bit more about this issue. We have shown the results of RV-tracing experiments from various types of ACA neurons in different Cre lines, including three major types of cortical interneurons, PV+, SST+, and VIP+ neurons. Consistent with other studies (Ma et al., 2021; Yao et al., 2023), we found that the distribution patterns of RV-labeled presynaptic neurons to different types of cortical neurons were similar between Cre lines (CaMKII α -Cre, PV-Cre, SST-Cre, and VIP-Cre). Based on these results, we concluded that “both excitatory neurons and three major types of inhibitory neurons in the ACA are directly innervated by callosal inputs from the contralateral ACA” (P. 13, Lines 15-16).

“Fig. 5 | Direct interhemispheric connections to different types of ACA neurons detected by RV-mediated trans-synaptic retrograde tracing. a, Viral vectors and injection procedure for RV-mediated trans-synaptic retrograde tracing from the indicated types of ACA neurons. **b,** Viral injection sites in the ACA of CaMKII α -, PV-, SST-, and VIP-Cre mice (scale bar, 1 mm). Inset, enlarged view of the region in the white box showing starter cells for RV tracing (yellow, scale bar, 20 μ m). Green, EGFP; red, mCherry; blue, DAPI. **c,** Distribution of starter cells in each group (Pyr, $n = 3$ mice; PV, $n = 5$; SST, $n = 3$; VIP, $n = 5$). **d,** RV-labeled neurons in the contralateral ACA (red box in coronal diagram) of CaMKII α -, PV-, SST-, and VIP-Cre mice (scale bar, 200 μ m). **e,** Detected RV-labeled neurons in the contralateral hemisphere in each group. Dots, detected neurons. **f,** Percentages of RV-labeled interhemispheric presynaptic neurons in each region among all the RV-labeled neurons in the contralateral hemisphere (Pyr, $n = 3$ mice; PV, $n = 5$; SST, $n = 3$; VIP, $n = 5$). Included are areas with $>2\%$ labeling. All data are presented as the mean \pm SEM.”

RV tracing is a powerful tool for visualizing the distribution of inputs to specific neuronal populations but lacks information about the synaptic properties of labeled

connections. Thus, we further examined the ACA callosal input strength on these inhibitory neurons. We found that the SST+ and VIP+ neurons received weak callosal input, and the PV+ neurons received the strongest callosal input among all three types of inhibitory neurons (Fig. 6f). The revised text is as follows (P. 14, Lines 17-20):

“Notably, the callosal inputs were much stronger in PV+ neurons (> 2.8 folds) than in SST+ and VIP+ neurons examined across different layers (Fig. 6f). These results indicated that callosal inputs weakly activate SST+ and VIP+ neurons and strongly activate PV+ neurons.”

“Fig. 6 | A transcallosal inhibition loop formed by CPNs and PV+cal neurons. **a**, Schematic of the slice experiment, with whole-cell recording of ACA pyramidal neurons and optogenetic activation of callosal inputs. **b**, EPSCs and IPSCs in an example layer 2/3 CPN and a nearby Pyr neuron. Blue dots, 5-ms blue light pulses (10 Hz). **c**, Contribution of inhibitory inputs to total synaptic inputs, measured as $IPSC_{charge}/(IPSC_{charge}+EPSC_{charge})$. Callosal inputs drive stronger inhibition in CPNs than in other Pyrs. *, $P < 0.05$; **, $P < 0.01$; Wilcoxon

rank-sum test. **d**, Schematic of the slice experiment to measure callosal-inputs-induced EPSPs in ACA neurons. **e**, Example monosynaptic EPSPs recorded from Pyr, PV+, SST+, and VIP+ neurons in layer 2/3 with TTX and 4-AP treatment to block local neuron spiking. **f**, Normalized EPSP amplitudes in different cell types (normalized to L2/3 pramidal neurons). Most of the neuron types in L2/3 received stronger callosal inputs as compared to the corresponding neuron types in L5 and L6 ($P < 0.05$, Wilcoxon rank-sum test), excluding the similar input strength for L2/3 PV+ and L6 PV+ neurons ($P = 0.49$). For each examined layer, the EPSP amplitude in PV+ neurons was larger than those in any other group of ACA neurons ($P < 0.005$, Wilcoxon rank-sum test). **g**, Schematic of the slice experiment to measure PV_{+cal}-neuron-driven IPSCs in CPNs. **h**, Example IPSCs recorded from a layer 2/3 CPN and a nearby pyramidal neuron. **i**, Normalized PV_{+cal}-IPSC amplitudes in CPNs were larger than in other pyramidal neurons across different layers ($P < 0.04$, Wilcoxon signed-rank test). **j**, Schematic of the slice experiment to measure callosal inputs in PV_{+cal} and other nearby PV+ neurons. **k**, Example EPSPs recorded from layer 2/3 PV_{+cal} and other PV+ neurons. **l**, Normalized EPSP amplitudes in PV_{+cal} neurons were larger than in other PV+ neurons ($P < 0.04$, Wilcoxon rank-sum test). **m**, Conceptual diagram of the positive feedback transcallosal inhibition loop. Line width represents the amplitude of synaptic inputs. All data are presented as the mean \pm SEM.”

Reviewer Fig. 1| The input strengths of the callosal inputs on different types of inhibitory neurons. Normalized EPSP amplitudes in different cell types (normalized to VIP+ neurons in each layer). The synaptic strength of callosal inputs was comparable

on SST+ and VIP+ neurons in each examined layer. The callosal inputs were much stronger in PV+ neurons (> 2.8 folds) than in SST+ and VIP+ neurons examined across different layers. The dataset is the same as shown in Fig. 6f, but instead of normalized EPSP to L2/3 Pyrs, the EPSP amplitudes are here normalized to the responses of VIP+ neurons in each layer. All data are presented as the mean \pm SEM.

2) *And a result that seems problematic is not discussed: shutting down both sides of ACA does not affect behavior (Figure 8 d,e).*

RESPONSE:

Thank you for focusing our attention here. We now understand the need for further discussion of our result showing that shutting down both sides of the ACA does not affect behavior.

Lesion studies in felines and clinical observations on stroke patients have shown that a contralateral brain lesion on a previously spared hemisphere could paradoxically cancel the neurological deficits in visuospatial function generated by a first lesion, *e.g.*, recovery from visuospatial neglect (Corbetta and Shulman, 2011; Lomber and Payne, 1996; Valero-Cabre et al., 2020). For example, it was observed initially in the cat where severe and permanent contralateral visually guided attentional deficits generated by the ablation of large areas of the visual cortex were reversed by the subsequent removal of the superior colliculus (SC) opposite to the cortical lesion (Sprague, 1966); this was termed the “Sprague Effect”. Later studies have shown that the sequential lesions in the posterior parietal, frontal, or midbrain locations in the two hemispheres can also reverse the visuospatial bias induced by the first lesion (Dambeck et al., 2006; Hilgetag et al., 2001; Lomber and Payne, 1996; Lomber et al., 2002; Payne et al., 1996; Rushmore et al., 2013; Valero-Cabre et al., 2017; Vuilleumier et al., 1996; Weddell, 2004).

Consistent with these previous studies, we found that bilateral activation of ACA PV+ neurons (shutting down both sides of ACA) reversed the behavioral bias induced by unilateral activation of ACA PV+ neurons (shutting down one side of ACA) in the visual change-detection task in rodents. These results supported that unilateral lesion-induced visuospatial bias is better explained by imbalanced network activity between two hemispheres, rather than by structural damage of specific brain regions. Interestingly, we also found that short-term (12 days) activation of PV+ neurons in the contralesional ACA resulted in a benefit to contralesional detection, both during and after learning. Accordingly, our results support the notion that reestablishing a balanced interhemispheric interaction can promote recovery from visuospatial bias after lesion in one hemisphere. We have added related content in the Discussion section (P. 23, Lines 6-15):

“Previous lesion studies in felines and clinical observations on stroke patients have shown that a contralateral brain lesion on a previously spared hemisphere can, paradoxically, cancel the deficits in visuospatial functions generated by a first lesion, *e.g.*, recovery from visuospatial neglect-an attentional deficit (Corbetta and Shulman, 2011; Lomber and Payne, 1996; Valero-Cabre et al., 2020). This is often referred to as the "Sprague Effect," named after J.M. Sprague who first reported it in the 1960s (Sprague, 1966). Consistent with previous studies, we found that bilateral activation of PV+ neurons reversed the unilateral inactivation/lesion-induced visuospatial bias in rodents in a visual change-detection task, suggesting that the ACA may contribute to the attentional components of decision-making in this task. Together, these results supported that unilateral lesion-induced visuospatial bias is mediated by imbalanced network activity between two hemispheres, rather than by the activity level of each brain structure in the network.”

3) Another main criticism concerns the statistics. It is not clear if the statistics in the paper consider the independent measurement to be the session or the mouse. It should be the mouse. For instance, in the caption to fig 1j it says there are 11 mice and 69 sessions. If the statistics assume $n = 69$, that's a mistake: the unit of independent measurement here is the mouse, so $n = 11$. Sessions in the same mouse are not independent measurements. Similarly, Figure 3 shows one bar per session (29 sessions). But the unit of independent measurement is the mouse (14 mice), not the session. Statistics should be done across mice, not sessions. If desired, one can use a nested model.

RESPONSE:

We have re-examined the statistical analysis methods and corrected previous errors. And all the detailed information for ANOVA, including the F ratio, the associated degrees of freedom, the exact associated P value, and partial η^2 for ANOVA are now provided in Extended Data Table 2.

In Fig. 1 and Fig. 8g, the statistical analysis was done across mice, and we have now removed the misleading information about the number of sessions in the figure legend. As the parameters being compared were obtained through repeated measurements of the same object, we have replaced the previously used one-way ANOVA with a two-way repeated ANOVA and have corrected the figure legends accordingly. The results of corrected statistics support our conclusions in the originally submitted version.

In addition, the statistics related to optogenetic manipulations in Fig. 3, 4, and 8b-e, and Extended Data Fig. 5c,d and 13c,d were done across sessions. We have now

replaced all statistics done across sessions with statistics done across mice:

1. A three-way repeated ANOVA was used to assess single and crossover effects for side (contralateral and ipsilateral), contrast (100% and 12.5%), and laser (on and off).
2. A two-way repeated ANOVA was used to assess single and crossover effects for contrast and laser on each side.
3. A one-way repeated ANOVA was used to assess the laser effects on each side at each contrast level.

The ANOVA conducted across mice yielded similar laser effects on hit rates, false alarm (FA) rates, and reaction time as the one conducted across sessions. Briefly, in the unilateral neural activity manipulation experiments, three-way repeated ANOVAs revealed that laser had similar effects on hit rates and FA rates at the 100% and 12.5% contrast levels, but had significantly different effects on contralateral vs. ipsilateral hit rates and FA rates. Two-way repeated ANOVAs revealed significant laser effects on hit rates and FA rates on each side. One-way repeated ANOVA results are more susceptible to the sample size: we observed significant laser effects on hit rates and FA rates in some cases, but not in every case. For reaction time, three-way repeated ANOVAs revealed significantly different effects on contralateral vs. ipsilateral reaction time. One-way repeated ANOVAs revealed significant laser effects on the contralateral and ipsilateral sides at the 12.5% contrast level, but not at the 100% contrast level. One exception is that activation of PV^{+cal} neurons did not result in a significant change in reaction time; this was consistently revealed by the ANOVAs conducted across mice and across sessions.

These corrections to the statistical analyses in the major figures and supplementary figures are shown as follows:

“Fig. 1 | Performance of ACA-lesion mice in the 2AUC change-detection task. a, Schematic for the behavioral tasks. FA, false alarm; CR, correct rejection. **b**, Schematic of the behavioral paradigms for learning phase 1. **c**, Performance of the control group in learning phase 1. The performance was not significantly different between the left-go (red) and right-go (blue) trials ($n = 9$ mice, $F_{\text{side}(1,8)} = 0.07$, $P_{\text{side}} = 0.8$, $F_{\text{side} \times \text{day}(9,72)} = 1$, $P_{\text{side} \times \text{day}} = 0.45$, two-way repeated ANOVA; day 10, left-hit, $94.4 \pm 2.2\%$; right-hit, $92.6 \pm 1.5\%$; $P = 0.52$, paired t-test). Colored shading, \pm SEM. **d**, Schematic diagram of the viral strategy used to induce ACA lesion (left) and fluorescence image of a coronal section showing an ACA lesion (right). Green, fluorescent Nissl staining. **e**, Performance of the ACA-lesion group in learning phase 1. ACA-lesion mice failed to learn the

contralesional detection task. The hit rates in contralesional trials were significantly lower than in ipsilesional trials during learning1 ($n = 21$ mice, $F_{\text{side}}(1,20) = 161.66$, $P_{\text{side}} = 5 \times 10^{-11}$, $F_{\text{side*day}}(11,220) = 19.85$, $P_{\text{side*day}} = 2 \times 10^{-27}$, two-way repeated ANOVA). At the end of learning phase1, the contralesional hit rate is significantly lower than the ipsilesional hit rate (Day 12, $P = 1 \times 10^{-9}$, paired t-test). **f,g**, Similar to **e**, but for performance of the RSP-lesion and PTLp-lesion groups. RSP and PTLp lesions slowed down the learning of the contralesional change-detection task (RSP-lesion, $n = 7$ mice, $F_{\text{side*day}}(12,72) = 1.92$, $P_{\text{side*day}} = 0.046$; PTLp-lesion, $n = 6$ mice, $F_{\text{side*day}}(12,60) = 5.44$, $P_{\text{side*day}} = 4 \times 10^{-6}$, two-way repeated ANOVA). However, at the end of learning phase1 (Day 12), the contralesional hit rate caught up with the ipsilesional hit rate (RSP-lesion, $P = 0.96$; PTLp-lesion, $P = 0.36$, paired t-test). **h**, Schematic of the behavioral paradigms for the test phase. **i**, Performance of the control group in the test phase. The hit rates and FA rates were not significantly different between the left-go (red) and right-go (blue) trials across different contrast levels ($n = 8$ mice, hit rates, $F_{\text{side}}(1,7) = 0.08$, $P_{\text{side}} = 0.79$; FA rates, $F_{\text{side}}(1,7) = 0.009$, $P_{\text{side}} = 0.93$, two-way repeated ANOVA). At each contrast level, left-hit + left-miss = 100%; right-hit + right-miss = 100%; left-FA + right-FA + CR = 100%. **j**, Similar to **i**, but for performance of the ACA-lesion group. The contralesional (red) hit rates and FA rates were significantly lower than the ipsilesional (blue) hit rates and FA rates ($n = 11$ mice, hit rates, $F_{\text{side}}(1,10) = 124$, $P_{\text{side}} = 6 \times 10^{-7}$; FA rates, $F_{\text{side}}(1,10) = 8.94$, $P_{\text{side}} = 0.01$, two-way repeated ANOVA). All data are presented as the mean \pm SEM. Also see Extended Data Table 2 for detailed parameters of ANOVA.”

“Fig. 3 | Manipulation of ACA CPN activity has opposite effects on contralateral and ipsilateral performance. a, Schematic diagram of the viral strategy for optogenetic manipulation of CPNs. **b**, Schematic diagram of optogenetic inhibition and activation of CPNs with yellow and blue laser, respectively. **c**, Effect of unilateral CPN inhibition on visual change detection ($n = 14$ mice). Left, changes in contralateral performance. Right, changes in ipsilateral performance. Inhibition of CPNs (yellow laser, 589 nm, 3-8 mW, constant light) had similar effects on hit rates and FA rates at the 100% and 12.5% contrast levels (hit rates, $F_{\text{contrast}*\text{laser}}(1,13) = 0.02$, $P_{\text{contrast}*\text{laser}} = 0.90$; FA rates, $F_{\text{contrast}*\text{laser}}(1,13) = 0.03$, $P_{\text{contrast}*\text{laser}} = 0.87$; three-way repeated ANOVA), while having significantly different effects on contralateral vs. ipsilateral hit rates and FA rates (hit rates, $F_{\text{side}*\text{laser}}(1,13) = 53.96$, $P_{\text{side}*\text{laser}} = 6 \times 10^{-6}$; FA rates, $F_{\text{side}*\text{laser}}(1,13) = 22.24$, $P_{\text{side}*\text{laser}} = 4 \times 10^{-4}$; three-way repeated ANOVA). Inhibition of CPNs significantly impaired contralateral detection (decreased hit rates, green bars, $F_{\text{laser}}(1,13) = 30.04$, $P_{\text{laser}} = 1 \times 10^{-4}$; increased FA rates, yellow bars, $F_{\text{laser}}(1,13) = 9.85$, $P_{\text{laser}} = 0.008$; two-way repeated ANOVA) and improved ipsilateral detection (increased hit rates, green bars, $F_{\text{laser}}(1,13) = 40.11$, $P_{\text{laser}} = 3 \times 10^{-5}$; decreased FA rates, yellow bars, $F_{\text{laser}}(1,13) = 8.27$, $P_{\text{laser}} = 0.01$; two-way repeated ANOVA). **d**, Effect of unilateral CPN inhibition on reaction times in hit trials. Inhibition of CPNs induced significantly different effects on contralateral and ipsilateral reaction time ($F_{\text{side}*\text{laser}}(1,13) = 25.60$, $P_{\text{side}*\text{laser}} = 2 \times 10^{-4}$, three-way repeated ANOVA), showing a significant increase in the contralateral reaction time and a significant decrease in the ipsilateral reaction time at the 12.5% contrast level (contralateral reaction time, $F_{\text{laser}}(1,13) = 7.17$,

$P_{\text{laser}} = 0.02$; ipsilateral reaction time, $F_{\text{laser}}(1,13) = 17.255$, $P_{\text{laser}} = 0.001$; one-way repeated ANOVA). **e,f**, Similar to **c,d**, but for optogenetic activation of CPNs ($n = 9$ mice). **e**, Activation of CPNs (blue laser, 473 nm, 1-3 mW, 10 Hz, 5 ms per pulse) had similar effects on hit rates and FA rates at the 100% and 12.5% contrast levels (hit rates, $F_{\text{contrast}*\text{laser}}(1,8) = 2.38$, $P_{\text{contrast}*\text{laser}} = 0.16$; FA rates, $F_{\text{contrast}*\text{laser}}(1,8) = 2.32$, $P_{\text{contrast}*\text{laser}} = 0.17$; three-way repeated ANOVA), while having significantly different effects on contralateral vs. ipsilateral hit rates and FA rates (hit rates, $F_{\text{side}*\text{laser}}(1,8) = 27.88$, $P_{\text{side}*\text{laser}} = 0.001$; FA rates, $F_{\text{side}*\text{laser}}(1,8) = 18.31$, $P_{\text{side}*\text{laser}} = 0.003$; three-way repeated ANOVA). Activation of CPNs significantly improved contralateral detection (increased hit rates, $F_{\text{laser}}(1,8) = 18.07$, $P_{\text{laser}} = 0.003$; decreased FA rates, $F_{\text{laser}}(1,8) = 14.37$, $P_{\text{laser}} = 0.005$; two-way repeated ANOVA) and impaired ipsilateral detection (decreased hit rates, $F_{\text{laser}}(1,8) = 15.49$, $P_{\text{laser}} = 0.004$; increased FA rates, $F_{\text{laser}}(1,13) = 14.34$, $P_{\text{laser}} = 0.005$; two-way repeated ANOVA). **f**, Activation of CPNs induced significantly different effects on contralateral and ipsilateral reaction time ($F_{\text{side}*\text{laser}}(1,8) = 5.90$, $P_{\text{side}*\text{laser}} = 0.04$, three-way repeated ANOVA), showing a significant decrease in the contralateral reaction time and a significant increase in the ipsilateral reaction time at the 12.5% contrast level (contralateral reaction time, $F_{\text{laser}}(1,8) = 10.46$, $P_{\text{laser}} = 0.01$; ipsilateral reaction time, $F_{\text{laser}}(1,8) = 5.71$, $P_{\text{laser}} = 0.04$; one-way repeated ANOVA). All data are presented as the mean \pm SEM. Also see Extended Data Table 2 for detailed parameters of ANOVA.”

“Fig. 4 | Inhibiting CPN callosal-projection axons induced similar behavioral bias to inhibiting contralateral CPNs. a, Schematic of the slice experiment for examining the temporal dynamics of eOPN3-induced inhibition of callosal-projection axons. **b**, EPSCs induced by activation of ChrimsonR at the axon terminals of contralateral CPNs in an example layer 2/3 pyramidal neuron. Red dots, 5-ms red light pulses (647 nm, 10 Hz, 3 mW). Control EPSCs were recorded before eOPN3 activation. eOPN3 activation lasted for 4s (green laser, 532 nm, constant light, 5 mW), and the red light pulses were delivered 2s after the green light start. Red light pulses were delivered again at 4s and 8s after the end of green light stimulation. **c**, Schematic diagrams of the viral strategy (left) and optogenetic inhibition of callosal-projection axons (right) in behaving mice. **d**, Schematic of the behavior paradigm for optogenetic inhibition of callosal-projection axons. The laser-on trials and laser-off trials were interleaved with a 10-s inter-trial interval. **e**, Effect of CPN callosal-projection axon inhibition on visual change detection ($n = 5$ mice). Left, changes in contralateral performance. Right, changes in ipsilateral performance. Inhibiting CPN callosal-projection axons (green, laser, 532 nm, 5 mW, constant light) had similar effects on hit rates and FA rates at the 100% and 12.5% contrast levels (hit rates, $F_{\text{contrast}*\text{laser}}(1,4) = 0.78$, $P_{\text{contrast}*\text{laser}} = 0.43$; FA rates, $F_{\text{contrast}*\text{laser}}(1,4) = 3.72$, $P_{\text{contrast}*\text{laser}} = 0.13$; three-way repeated ANOVA), while having significantly different effects on contralateral vs. ipsilateral hit rates and FA rates (hit rates, $F_{\text{side}*\text{laser}}(1,4) = 87.25$, $P_{\text{side}*\text{laser}} = 0.001$; FA rates, $F_{\text{side}*\text{laser}}(1,4) = 9.70$, $P_{\text{side}*\text{laser}} = 0.036$; three-way repeated ANOVA). Inhibiting CPN callosal-projection axons significantly improved contralateral detection (increased hit rates, green bars, $F_{\text{laser}}(1,4) = 50.82$, $P_{\text{laser}} = 0.002$; decreased FA rates, yellow bars, $F_{\text{laser}}(1,4) = 9.05$, $P_{\text{laser}} = 0.04$; two-way repeated ANOVA) and impaired ipsilateral detection (decreased hit rates, green bars, $F_{\text{laser}}(1,4) = 41.96$, $P_{\text{laser}} = 0.003$; increased FA rates, yellow bars, $F_{\text{laser}}(1,4) = 8.14$, $P_{\text{laser}} = 0.046$; two-way repeated ANOVA). **f**, Effect of CPN callosal-projection axon inhibition on reaction times in hit trials. Inhibiting CPN callosal-projection axons significantly decreased the reaction time in contralateral-hit trials (red bar) and increased the reaction time in ipsilateral-hit trials (blue bar) at 12.5% contrast (contralateral reaction time, $F_{\text{laser}}(1,4) = 62.53$, $P_{\text{laser}} = 0.001$; ipsilateral reaction time, $F_{\text{laser}}(1,4) = 8.39$, $P_{\text{laser}} = 0.04$; one-way repeated ANOVA). All data are presented as the mean \pm SEM. Also see Extended Data Table 2 for detailed parameters of ANOVA.”

‘Fig. 8 | Activation of contralesional PV+ neurons improved contralesional change detection. a, Schematic of optogenetic manipulation. b, Effect of unilateral activation of PV+ neurons on visual

change detection ($n = 5$ mice). Left, changes in contralateral performance. Right, changes in ipsilateral performance. Unilateral activation of PV+ neurons (blue laser, 473 nm, 3-8 mW, 10 Hz, 5 ms per pulse) had similar effects on hit rates and FA rates at the 100% and 12.5% contrast levels (hit rates, $F_{\text{contrast*laser}(1,4)} = 0.15$, $P_{\text{contrast*laser}} = 0.72$; FA rates, $F_{\text{contrast*laser}(1,4)} = 3.99$, $P_{\text{contrast*laser}} = 0.12$; three-way repeated ANOVA), while having significantly different effects on contralateral vs. ipsilateral hit rates and FA rates (hit rates, $F_{\text{side*laser}(1,4)} = 51.37$, $P_{\text{side*laser}} = 0.002$; FA rates, $F_{\text{side*laser}(1,4)} = 36.01$, $P_{\text{side*laser}} = 0.004$; three-way repeated ANOVA). Unilateral activation of PV+ neurons significantly impaired contralateral detection (decreased hit rates, green bars, $F_{\text{laser}(1,4)} = 115.94$, $P_{\text{laser}} = 4 \times 10^{-4}$; increased FA rates, yellow bars, $F_{\text{laser}(1,4)} = 34.77$, $P_{\text{laser}} = 0.004$; two-way repeated ANOVA) and improved ipsilateral detection (increased hit rates, green bars, $F_{\text{laser}(1,4)} = 19.70$, $P_{\text{laser}} = 0.01$; decreased FA rates, yellow bars, $F_{\text{laser}(1,4)} = 8.19$, $P_{\text{laser}} = 0.046$; two-way repeated ANOVA).

c, Effect of unilateral activation of PV+ neurons on reaction times in hit trials. Unilateral activation of PV+ neurons induced significantly different effects on contralateral and ipsilateral reaction times ($F_{\text{side*laser}(1,4)} = 36.66$, $P_{\text{side*laser}} = 0.004$, three-way repeated ANOVA), showing a significant increase in the contralateral reaction time and a significant decrease in the ipsilateral reaction time at the 12.5% contrast level (contralateral reaction time, $F_{\text{laser}(1,4)} = 8.09$, $P_{\text{laser}} = 0.047$; ipsilateral reaction time, $F_{\text{laser}(1,4)} = 8.12$, $P_{\text{laser}} = 0.046$; two-way repeated ANOVA).

d,e, Similar to **b,c**, but for bilateral activation of PV+ neurons ($n = 6$ mice). Bilateral activation of PV+ neurons did not induce a change in visual detection or reaction time on either side (hit rates, $F_{\text{side*laser}(1,5)} = 0.78$, $P_{\text{side*laser}} = 0.42$, $F_{\text{contrast*laser}(1,5)} = 0.78$, $P_{\text{contrast*laser}} = 0.42$; FA rates, $F_{\text{side*laser}(1,5)} = 2.77$, $P_{\text{side*laser}} = 0.16$, $F_{\text{contrast*laser}(1,5)} = 0.16$, $P_{\text{contrast*laser}} = 0.70$; reaction time, $F_{\text{side*laser}(1,5)} = 0.30$, $P_{\text{side*laser}} = 0.61$, $F_{\text{contrast*laser}(1,5)} = 0.85$, $P_{\text{contrast*laser}} = 0.40$; three-way repeated ANOVA).

f, Viral vectors and injection procedure for chemogenetic activation of contralesional PV+ neurons. Lower right, fluorescence image showing neurons with enhancer-driven expression of hM3D/tdTomato (red) and anti-PV staining (green). Red arrowheads, hM3D/tdTomato-only neurons; white arrowheads, double labeled neurons.

g, Performance of ACA-lesion mice in learning phase1 and the test phase with or without contralesional PV+ neuron activation during learning phase1. Contralesional hit rates were significantly higher in CNO-treated ACA-lesion mice ($n = 6$ mice) than in ACA-lesion mice that did not receive CNO ($n = 21$ mice) during learning phase1 ($F_{\text{CNO}(1,300)} = 187.40$, $P_{\text{CNO}} = 2 \times 10^{-33}$, two-way ANOVA). In the test phase, contralesional hit rates were still significantly higher in CNO-treated ACA-lesion mice ($n = 6$ mice) than in ACA-lesion mice that did not receive CNO ($n = 11$ mice) ($F_{\text{CNO}(1,105)} = 152.90$, $P_{\text{CNO}} = 3 \times 10^{-22}$, two-way ANOVA). The data shown with dashed lines are the same

datasets as in Fig. 1e,j. Colored shading (left) and error bar (right) represent the \pm SEM. **h**, Conceptual diagram of the transcallosal loop that mediates interhemispheric inhibition in visuospatial processing. Line width represents the amplitude of synaptic inputs. All data are presented as the mean \pm SEM. Also see Extended Data Table 2 for detailed parameters of ANOVA.

“Extended Data Fig. 7. Effects of laser stimulation on the performance of control mice in the 2AUC change-detection task.

(a) Schematic diagrams of the virus injection and unilateral laser stimulation (473 nm, 8 mW, 10 Hz, 5 ms per pulse) in control mice. (b) Coronal section showing the locations of virus expression (green) and the implanted optical fiber. (c-d) Effect of unilateral laser stimulation in control mice ($n = 6$ mice) on visual change detection (c) and reaction time (d). Blue laser did not induce a change in visual detection or reaction time on either side (hit rates, $F_{\text{side}*\text{laser}}(1,5) = 1.40$, $P_{\text{side}*\text{laser}} = 0.29$,

$F_{\text{contrast}*\text{laser}}(1,5) = 0.56$, $P_{\text{contrast}*\text{laser}} = 0.49$; FA rates, $F_{\text{side}*\text{laser}}(1,5) = 1.08$, $P_{\text{side}*\text{laser}} = 0.35$, $F_{\text{contrast}*\text{laser}}(1,5) = 0.20$, $P_{\text{contrast}*\text{laser}} = 0.68$; reaction time, $F_{\text{side}*\text{laser}}(1,5) = 0.67$, $P_{\text{side}*\text{laser}} = 0.45$, $F_{\text{contrast}*\text{laser}}(1,5) = 0.04$, $P_{\text{contrast}*\text{laser}} = 0.85$; three-way repeated ANOVA). (e-g) Effects of unilateral laser stimulation on hit rates (e), FA rates (f), and reaction times in hit trials (g). The data shown in (e-g) are the same datasets as in (c-d), but show laser effects in each session ($n = 27$ sessions). All data are presented as the mean \pm SEM.”

“Extended Data Fig. 15. Effects of unilateral activation of PV+cal neurons on the performance in the 2AUC change-detection task.

(a) Schematic diagrams of the viral strategy (top) and optogenetic activation of PV+cal neurons (bottom). (b) Coronal section showing the

location of implanted optical fiber. **(c)** Effect of unilateral activation of PV_{+cal} neurons on visual change detection ($n = 6$ mice). Left, changes in contralateral performance. Right, changes in ipsilateral performance. Unilateral activation of PV_{+cal} neurons (blue laser, 473 nm, 1-5 mW, 10 Hz, 5 ms per pulse) had similar effects on hit rates and FA rates at the 100% and 12.5% contrast levels (hit rates, $F_{\text{contrast}*\text{laser}}(1,5) = 0.12$, $P_{\text{contrast}*\text{laser}} = 0.75$; FA rates, $F_{\text{contrast}*\text{laser}}(1,5) = 0.004$, $P_{\text{contrast}*\text{laser}} = 0.95$; three-way repeated ANOVA), while having significantly different effects on contralateral vs. ipsilateral hit rates and FA rates (hit rates, $F_{\text{side}*\text{laser}}(1,5) = 54.48$, $P_{\text{side}*\text{laser}} = 0.001$; FA rates, $F_{\text{side}*\text{laser}}(1,5) = 56.80$, $P_{\text{side}*\text{laser}} = 0.001$; three-way repeated ANOVA). Unilateral activation of PV_{+cal} neurons significantly impaired contralateral detection (decreased hit rates, green bars, $F_{\text{laser}}(1,5) = 17.22$, $P_{\text{laser}} = 0.009$; increased FA rates, yellow bars, $F_{\text{laser}}(1,5) = 7.55$, $P_{\text{laser}} = 0.04$; two-way repeated ANOVA) and improved ipsilateral detection (increased hit rates, green bars, $F_{\text{laser}}(1,5) = 14.65$, $P_{\text{laser}} = 0.01$; decreased FA rates, yellow bars, $F_{\text{laser}}(1,5) = 8.26$, $P_{\text{laser}} = 0.035$; two-way repeated ANOVA). **d**, Effect of unilateral activation of PV_{+cal} neurons on reaction times in hit trials. Unilateral activation of PV_{+cal} neurons did not cause significant change in the reaction time (reaction time, $F_{\text{side}*\text{laser}}(1,5) = 1.15$, $P_{\text{side}*\text{laser}} = 0.33$, $F_{\text{contrast}*\text{laser}}(1,5) = 2.29$, $P_{\text{contrast}*\text{laser}} = 0.19$; three-way repeated ANOVA). **(e-g)** Effects of unilateral laser stimulation on hit rates **(e)**, FA rates **(f)** and reaction times in hit trials **(g)**. The data shown in **(e-g)** are the same datasets as in **(c-d)**, but show laser effects in each session ($n = 24$ sessions). All data are presented as the mean \pm SEM. Also see Extended Data Table 2 for detailed parameters of ANOVA.”

4) Furthermore, the paper does not seem to show responses of neurons when the mouse makes the incorrect decision, moving the wheel in the wrong direction. And yet this would be a useful thing to see, because it would help show if the responses are related to the stimulus or to the movement or to both. There should be many such trials at low contrast.

RESPONSE:

Based on the reviewer’s guidance, we have now reanalyzed the behavioral data from previous miss trials (*i.e.*, where a visual change was present, but no correct wheel turn was made). We found few instances where the mouse made an incorrect decision (incorrect trials, where the wheel was turned 40 degrees in the wrong direction, the same criterion as in hit trials) at the 12.5% contrast level in each session (as shown in Reviewer Table 1 and Reviewer Fig. 2). And even fewer incorrect trials were observed at the 100% contrast level. In most cases, there was no wheel turn or only a small wheel turn (less than 10 degrees in either direction).

Previous miss trials at 12.5% contrast level						
Animal name	Contralateral miss			Ipsilateral miss		
	Total	Incorrect decision	Others	Total	Incorrect decision	Others
CPN #1	5	0	5	12	0	12
CPN #2	9	4	5	1	0	1
CPN #3	23	0	23	17	0	17
CPN #4	11	0	11	14	0	14
CPN #5	20	1	19	13	1	12
CPN #6	30	0	30	27	1	26
CPN #7	19	2	17	7	0	7
CPN #8	15	1	14	19	0	19
PV+cal #1	18	1	17	11	0	11
PV+cal #2	8	1	7	17	1	16
PV+cal #3	26	0	26	47	2	45
PV+cal #4	40	1	39	32	0	32
PV+cal #5	12	0	12	8	1	7
PV+cal #6	22	0	22	23	0	23
PV+cal #7	24	1	23	33	0	33
PV+cal #8	23	0	23	34	0	34

Reviewer Table 1. Re-analysis of previous “miss trials”.

Reviewer Fig. 2| Frequency plot of the distribution of trials with incorrect decisions (incorrect trials) at the 12.5% contrast level. a,b, The distribution of

incorrect-trial numbers in the contralateral-go trials (a) and ipsilateral-go trials (b), in the sessions with calcium imaging of CPNs. **c,d**, Similar to a,b, but for the sessions with calcium imaging of PV_{+cal} neurons.

Given this relative dearth of suitable incorrect-trial data, we instead reevaluated the encoding of visual stimulus and motor action from the neuronal responses in miss and FA trials (Fig. 2h,j, Fig. 7h,j, and Extended Data Fig. 5 and 14). Briefly, these analyses indicate that both ACA CPNs and PV_{+cal} neurons show selective visual-related activity, but did not show selective motor-related activity.

The more detailed revised text for CPNs is as follows (P. 8, Lines 17-22, and P. 9, Lines 1-6):

“We further investigated the neuronal encoding of visual stimulus and motor action from the CPN responses in miss and FA trials. At the population level, both contralateral-preferring and ipsilateral-preferring CPNs showed selectivity for visual stimuli (contralateral vs. ipsilateral drifting gratings) in miss trials (visual change without motor action), indicating that they encoded the properties of visual stimulus (Fig. 2g-j and Extended Data Fig. 5). The differences in neuronal responses to contralateral versus ipsilateral visual change were significantly greater in hit trials compared to miss trials (contralateral-preferring CPNs, $P = 4.5 \times 10^{-6}$; ipsilateral-preferring CPNs, $P = 1.2 \times 10^{-4}$, paired t-test), likely owing to differences in behavioral states such as task engagement and/or attention levels. Additionally, none of these neuron populations showed selectivity for the direction of motor actions (clockwise vs. counter-clockwise wheel rotation) in FA trials (motor action without visual change), indicating that they did not encode the properties of motor action. Together, these results demonstrate that ACA CPNs exhibit selective visual-related activity with a contralateral bias.”

Note that the full Fig. 2 (*i.e.*, all panels) is shown in our response to Reviewer #1’s first comment. Here, we present Fig. 2h,j, showing the averaged $\Delta F/F$ (z-scored) of contralateral-preferring and ipsilateral-preferring CPN in hit, miss, and FA trials are as follows:

“Fig. 2 | ACA CPNs show selective vision-related activity with a contralateral bias in the 2AUC change-detection task. h, Averaged $\Delta F/F$ (z-scored) of contralateral-preferring CPN in different types of trials. Left, activity in hit trials. Black dashed line, the start of visual white noise. Gray shading, window of the visuomotor period (from the start of drifting grating to the end of wheel rotation). The activity of contralateral-preferring CPNs was significantly higher in the contralateral-hit trials than in the ipsilateral-hit trials during the visuomotor period ($P = 1 \times 10^{-17}$, paired t-test test). Middle, activity in miss trials. Dashed line, the start of visual white noise. Gray shading, window during the presentation of drifting grating. The activity was significantly higher in the contralateral-miss trials than in the ipsilateral-miss trials during the presentation of drifting grating ($P = 3 \times 10^{-8}$, paired t-test test). Right, activity in FA trials. Black dashed line, the start of visual white noise. Gray shading, window during the wheel rotation. The activity did not differ between the contralateral-FA trials and ipsilateral-FA trials during the wheel rotation ($P = 0.14$, paired t-test test). Colored shading, \pm SEM. j, Similar to h, but for ipsilateral-preferring CPNs. The activity of ipsilateral-preferring CPNs was significantly lower in the contralateral-hit trials than in the ipsilateral-hit trials during the visuomotor period ($P = 8 \times 10^{-17}$, paired t-test test). Similarly, the activity of ipsilateral-preferring CPNs was significantly lower in the contralateral-miss trials than in the ipsilateral-miss trials during the presentation of drifting grating ($P = 6 \times 10^{-8}$, paired t-test test). However, there was no significant activity difference between the contralateral-FA trials and the ipsilateral-FA trials during the wheel rotation ($P = 0.10$, paired t-test test). See also Extended Data Fig. 4,5 and Methods.”

“Extended Data Fig. 5. The activity of contralateral-preferring and ipsilateral-preferring CPNs in the 2AUC change-detection task.

(a) Color-coded averaged $\Delta F/F$ (z-scored) of contralateral-preferring CPNs in miss trials. Each row shows the average activity of one neuron, sorted by peak response during the visuomotor period in hit trials (same as showing in Fig. 2g). Black dashed line, the start of visual white noise. The left and right gray dashed lines present the start and end of drifting grating, respectively. (b) Similar to (a), but for contralateral-preferring CPNs in FA trials. Black dashed line, the start of visual white noise. The left and right gray dashed lines present the start and end of wheel rotation, respectively. (c) Similar to (a), but for contralateral-preferring CPNs in CR trials. Black and gray dashed lines, the start and end of visual white noise, respectively. (d) Averaged $\Delta F/F$ (z-scored) of contralateral-preferring CPNs in CR trials. Colored shading, \pm SEM. Black dashed line, the start of visual white noise. Gray shading, window during the presentation of visual white noise. (e-h) Similar to (a-b), but for ipsilateral-preferring CPNs. Each row shows the average activity of one neuron, sorted by peak response during the visuomotor period in hit trials (same as showing in Fig. 2i).”

We have also reevaluated the encoding of visual stimulus and motor action from the neuronal responses in miss and FA trials of ACA PV_{+cal} neurons (Fig. 7h,j and Extended Data Fig. 14).

The revised text for PV_{+cal} neurons is as follows (P. 18, Lines 5-14):

“Given the relative dearth of contralateral-preferring PV_{+cal} neurons, we focused the following analysis on ipsilateral-preferring PV_{+cal} neurons. At the population level, ipsilateral-preferring PV_{+cal}

neurons showed selectivity for visual stimuli in miss trials, indicating that they encoded the properties of visual stimulus (Fig. 7g-j and Extended Data Fig. 14). Similar to CPNs, the differences in the responses of PV_{+cal} neurons to contralateral versus ipsilateral visual change were significantly greater in hit trials compared to miss trials (ipsilateral-preferring CPNs, $P = 1.2 \times 10^{-4}$, paired t-test), likely owing to differences in behavioral states. In addition, ipsilateral-preferring PV_{+cal} neurons did not show selectivity for the direction of motor actions in FA trials, indicating that they did not encode the properties of motor action. Together, these results demonstrate that the ACA PV_{+cal} neurons show selective visual-related activity that is biased towards the ipsilateral side.”

“Fig. 7 | ACA PV_{+cal} neurons show selective visual-related activity with an ipsilateral bias in the 2AUC change-detection task. a, Schematic diagrams of the viral strategy (top) and calcium imaging of PV_{+cal} neurons (bottom). **b,** Raw fluorescence trace of an example PV_{+cal} neuron. Same behavior paradigm as the test phase shown in Fig. 1h, containing the trials at 100% and 12.5% contrast levels. Dark color, high contrast; light color, low contrast. Purple shading, visual white noise; red shadings, contralateral gratings; blue shading, ipsilateral gratings. Red and blue arrowheads, end of wheel rotation in the contralateral and ipsilateral hit trials, respectively. Blue diamond, end of ipsilateral wheel rotation in the FA trial. Dashed line, end of timeout in the FA trial. **c,** Averaged $\Delta F/F$ (z-scored) of an example PV_{+cal} neuron in hit trials. Colored shading, \pm SEM. Black dashed line, the start of visual white noise. Gray shading, window of the visuomotor period (from the start of drifting grating to the end of wheel rotation). **d,** Color-coded averaged $\Delta F/F$ (z-scored) of all recorded PV_{+cal} neurons in contralateral-hit (left) and ipsilateral-hit (right) trials ($n = 8$ mice, 8 sessions, 151 neurons). Each row shows the average activity of one neuron, sorted by peak response during the visuomotor period. Black dashed line, the start of visual white noise. The left and right gray dashed lines present the start of drifting grating and the end of wheel rotation, respectively. **e,** Distribution of the difference in peak responses of z-scored $\Delta F/F$ during the visuomotor period between the contralateral-hit and ipsilateral-hit trials for the imaged PV_{+cal} neurons. Blue, ipsilateral-preferring cells; red, contralateral-preferring cells. Orange arrowhead, median. There were significantly more ipsilateral-preferring PV_{+cal} neurons than contralateral-preferring PV_{+cal} neurons ($P = 2 \times 10^{-5}$, chi-square test). **f,** Similar to c, but for all recorded PV_{+cal} neurons. The averaged responses of PV_{+cal} neurons were significantly stronger in the ipsilateral-hit trials than in the contralateral-hit trials during the visuomotor period ($P = 3 \times 10^{-5}$, paired t-test test). **g,** Similar to d, but for contralateral-preferring PV_{+cal} neurons. **h,** Averaged $\Delta F/F$ (z-scored) of contralateral-preferring PV_{+cal} neurons in different types of trials. Left, activity in hit trials. Black dashed line, the start of visual white noise. Gray shading, window of the visuomotor period (from the start of drifting grating to the end of wheel rotation). Middle, activity in miss trials. Dashed line, the start of visual white noise. Gray shading, window during the presentation of drifting grating. Right, activity in FA trials. Black dashed line, the start of visual white noise. Gray shading, window during the wheel rotation. Colored shading, \pm SEM. **i,j** Similar to g,h, but for ipsilateral-preferring PV_{+cal} neurons. The activity of ipsilateral-preferring PV_{+cal} neurons was significantly lower in the contralateral-hit trials than in the ipsilateral-hit trials during the visuomotor period ($P = 1.8 \times 10^{-8}$, paired t-test test). Similarly, the activity of ipsilateral-preferring PV_{+cal} neurons was significantly lower in the contralateral-miss trials than in the

ipsilateral-miss trials during the presentation of drifting grating ($P = 0.003$, paired t-test test). However, there is no difference in activity between the contralateral-FA trials and the ipsilateral-FA trials during the wheel rotation ($P = 0.8$, paired t-test test). See also Extended Data Fig. 12,14 and Methods.”

“Extended Data Fig. 14. The activity of contralateral-preferring and ipsilateral-preferring PV_{+cal} neurons in the 2AUC change-detection task.

(a) Color-coded averaged $\Delta F/F$ (z-scored) of contralateral-preferring PV_{+cal} neurons in miss trials. Each row shows the average activity of one neuron, sorted by peak response during the visuomotor period in hit trials (same as showing in Fig. 7g). Black dashed line, the start of visual white noise. The left and right gray dashed lines present the start and end of drifting grating, respectively. (b) Similar to (a), but for contralateral-preferring PV_{+cal} neurons in FA trials. Black dashed line, the start of visual white noise. The left and right gray dashed lines present the start and end of wheel rotation, respectively. (c) Similar to (a), but for contralateral-preferring PV_{+cal} neurons in CR trials. Black and gray dashed lines, the start and end of visual white noise, respectively. (d) Averaged $\Delta F/F$ (z-scored) of contralateral-preferring PV_{+cal} neurons in CR trials. Colored shading, \pm SEM. Black dashed line, the start of visual white noise. Gray shading, window during the presentation of visual white noise. (e-h) Similar to (a-b), but for ipsilateral-preferring PV_{+cal} neurons. Each row shows the average activity of one neuron, sorted by peak response during the visuomotor period in hit trials (same as showing in Fig. 7i).”

5) Finally, the paper should show more results of anatomical reconstructions (histology), to show us locations of lesions, placements of GRIN rods, etc.

RESPONSE:

Based on the reviewer’s guidance, we have added more results showing anatomical reconstructions, including for the locations of lesions and for the placements of GRIN rods, optical fibers, and cannulas in Extended Data Fig. 1, 2, 4, 6, 7, 8, 12, 13, 15, 16, and 17 (Extended Data Fig. 7 and 15 have been shown in the response to 3rd comment).

“Extended Data Fig. 1. Unilateral lesions of ACA, RSP, and PTLp.

(a) Schematic diagram for the viral strategy to induce unilateral RSP lesion (left) and fluorescence image of a coronal section showing a RSP lesion (right). Green, fluorescent Nissl staining. (b) Similar to (a), but for PTLp lesion. (c) Location of ACA lesion on a flat cortical map. (d) Similar to (c), but for RSP lesion. (e) Similar to (c), but for PTLp lesion.”

“Extended Data Fig. 2. Locations of GRIN lens and optical fibers in the ACA.

(a) Top-down view of a brain atlas illustrating the implantation sites of the GRIN lenses used in the calcium imaging experiments. Inset: A coronal section from the atlas, showing the anatomical locations of the ACA and the secondary motor cortex (MOs) at the level of the bregma. (b) Similar to (a), but for the implantation sites of the optical fibers used in the unilateral optogenetic manipulation experiments. (c) Similar to (a), but for the implantation sites of the optical fibers used in the bilateral optogenetic manipulation experiments.”

“Extended Data Fig. 4. The activity of example CPNs in the 2AUC change-detection task.

(a) Coronal section showing GCaMP6s fluorescence (green) and the location of implanted GRIN lens. (b-d) Averaged $\Delta F/F$ (z-scored) of CPN #1. Dashed line, the start of visual white noise. Colored shading, \pm SEM. (b) Activity in miss trials. Gray shading, window during the presentation of drifting grating. (c) Activity in CR trials. Gray shading, window during the presentation of visual white noise. (d) Activity in FA trials. Gray shading, window during the wheel rotation. (e) Activity in hit trials. Gray shading, window from the start of drifting grating to the end of wheel rotation. (f-h) Similar to (b-d), but for CPN #2. (i-l) Similar to (e-h), but for CPN #3. (m-p) Similar to (e-h), but for CPN #4.”

“Extended Data Fig. 6. Activation of ACA CPNs causes opposite behavioral changes to inhibition of ACA CPNs in the 2AUC visual change-detection task.

(a) Coronal section showing NpHR fluorescence (green) and the location of implanted optical fiber for unilateral inhibition of ACA CPNs. (b-d) Effects of unilateral inhibition of ACA CPNs on hit rates (b), FA rates (c) and reaction times in hit trials (d). The data shown in (b-d) are the same datasets as in Fig. 3c,d, but show laser effects in each session ($n = 29$ sessions). (e) Similar to (a), but for showing ChR2 fluorescence (green) and the location of implanted optical fiber for unilateral activation

of ACA CPNs. **(f-h)** Similar to **(b-d)**, but for the effects of unilateral activation of ACA CPNs ($n = 21$ sessions). The data shown in **(f-h)** are the same datasets as in Fig. 3e,f. All data are presented as the mean \pm SEM.”

“Extended Data Fig. 8. Inhibition of CPN callosal-projection axons causes similar behavioral changes to inhibiting their somas.

(a) Coronal section showing eOPN3 fluorescence (red) and the location of implanted optical fiber for inhibition of CPN callosal-projection axons. **(b-d)** Effects of inhibition of CPN callosal-projection axons on hit rates **(b)**, FA rates **(c)** and reaction times in hit trials **(e)**. The data shown in **(b-d)** are the same datasets as in Fig. 4e,f, but show laser effects in each session ($n = 25$ sessions). All data are presented as the mean \pm SEM.”

“Extended Data Fig. 12. The activity of example PV+cal neurons in the 2AUC change-detection task.

(a) Coronal section showing GCaMP6s fluorescence (green) and the location of implanted GRIN lens. (b-d) Averaged $\Delta F/F$ (z-scored) of PV+cal #1. Dashed line, the start of visual white noise. Colored shading, \pm SEM. (b) Activity in miss trials. Gray shading, window during the presentation of drifting grating. (c) Activity in CR trials. Gray shading, window during the presentation of visual white noise. (d) Activity in FA trials. Gray shading, window during the wheel rotation. (e-h) Averaged $\Delta F/F$ (z-scored) of PV+cal #2. (e) Activity in hit trials. Gray shading, window from the start of drifting grating to the end of wheel rotation. (f-h) Similar to (b-d), but for PV+cal #2. (i-l) Similar to (e-h), but for PV+cal #3.”

“Extended Data Fig. 13. The activity of PV+ neurons in the 2AUC change-detection task.

(a) Schematic diagrams for viral strategy (left) and calcium imaging of PV+ neurons (right). (b) Coronal section showing GCaMP6s fluorescence (green) and the location of implanted GRIN lens. (c) Color-coded averaged $\Delta F/F$ (z-scored) of all recorded PV+ neurons in contralateral-hit (left) and ipsilateral-hit (right) trials ($n = 7$ mice, 7 sessions, 209 neurons). Each row shows the average activity of one neuron, sorted by peak response during the visuomotor period. Black dashed line, the start of visual white noise. The left and right gray dashed lines present the start of drifting grating and the end of wheel rotation, respectively. (d)

Distribution of the difference in peak responses of z-scored $\Delta F/F$ during the visuomotor period between the contralateral-hit and ipsilateral-hit trials in the imaged PV+ neurons. Blue, ipsilateral-preferring cells; red, contralateral-preferring cells. Orange arrowhead, median. **(e)** Similar to **(c)**, but for contralateral-preferring PV+ neurons. **(f)** Averaged $\Delta F/F$ (z-scored) of contralateral-preferring PV+ neurons in hit trials. Colored shading, \pm SEM. Dashed line, the start of visual white noise. Gray shading, window of the visuomotor period. **(g-h)** Similar to **(e-f)** but for ipsilateral-preferring PV+ neurons in hit trials. **(i-j)** Similar to **(e-f)**, but for contralateral-preferring PV+ neurons in miss trials. **(i)** Black dashed line, the start of visual white noise. The left and right gray dashed lines present the start and end of drifting grating, respectively. **(j)** Black dashed line, the start of visual white noise. Gray shading, window during the presentation of drifting grating. **(k-l)** Similar to **(i-j)** but for ipsilateral-preferring PV+ neurons in miss trials. **(m-n)** Similar to **(e-f)**, but for contralateral-preferring PV+ neurons in FA trials. **(m)** Black dashed line, the start of visual white noise. The left and right gray dashed lines present the start and end of wheel rotation, respectively. **(n)** Black dashed line, the start of visual white noise. Gray shading, window during the wheel rotation. **(o-p)** Similar to **(m-n)**, but for ipsilateral-preferring PV+ neurons in FA trials. **(q-r)** Similar to **(e-f)**, but for contralateral-preferring PV+ neurons in CR trials. **(q)** Black and gray dashed lines, the start and end of visual white noise, respectively. **(r)** Black dashed line, the start of visual white noise. Gray shading, window during the presentation of visual white noise. **(s-t)** Similar to **(q-r)**, but for ipsilateral-preferring PV+ neurons in CR trials.”

“Extended Data Fig. 16. Bilateral activation of PV+ neurons reversed the behavioral bias induced by unilateral activation of PV+ neurons.

(a) Coronal section showing the location of implanted optical fiber for unilateral activation of PV+ neurons in PV-Ai32 mice. **(b-d)** Effects of unilateral activation of PV+ neurons on hit rates **(b)**, FA rates **(c)** and reaction times in hit trials **(d)**. The data shown in **(b-d)** are the same datasets as in Fig. 8b,c, but show laser effects in each session ($n = 25$ sessions). **(e)** Similar to **(a)**, but for showing the location of implanted optical fibers for bilateral activation of PV+ neurons in PV-Ai32 mice. **(f-**

h) Similar to **(b-d)**, but for the effects of bilateral activation of PV+ neurons ($n = 24$ sessions). The data shown in **(f-h)** are the same datasets as in Fig. 8d,e. All data are presented as the mean \pm SEM.”

“Extended Data Fig. 17. Enhancer-driven hM3D(Gq) expression in PV+ neurons in CaMKII α -Cre mice.

Schematic of the viral strategy for enhancer-driven expression of hM3D(Gq) in PV+ neurons in the contralesional ACA. **(b)** Coronal section showing an ACA lesion (right), contralesional hM3D fluorescence (red), and the location of the implanted cannula (left). We noted that the ipsilesional ACA and the area near the implanted cannula showed a relatively strong spontaneous fluorescence signal, which could represent dead cell debris. **(c)** Specificity and sensitivity of enhancer-driven expression of hM3D(Gq)-tdTomato in PV+ neurons. “Specificity” here refers to the percentage of cells expressing the tdTomato (red) that co-express PV (anti-PV, green, see Fig. 8f); “Sensitivity” refers to the percentage of cells expressing PV (anti-PV, green) that co-express tdTomato (red). **(d)** FA rates of ACA-lesion mice in test phase with or without contralesional PV+ neuron activation during learning phase1. In CNO-treated ACA-lesion mice, ipsilesional FA rates were significantly higher than contralesional FA rates ($n = 6$ mice, $F_{side(1,5)} = 31.32$, $P_{side} = 0.003$; two-way repeated ANOVA). The ipsilesional FA rates in CNO-treated ACA-lesion mice were significantly lower than in the control ACA-lesion group (*i.e.*, without treatment) ($F_{CNO(1,105)} = 113.45$, P_{CNO}

$= 2 \times 10^{-18}$, two-way ANOVA). The contralesional FA rates in CNO-treated ACA-lesion mice were similar to those detected for the control ACA-lesion group ($F_{\text{CNO}}(1,105) = 0.41$, $P_{\text{CNO}} = 0.52$, two-way ANOVA). The data shown with dashed lines are the same datasets as in Fig. 1j. All data are presented as the mean \pm SEM. Also see Extended Data Table 2 for detailed parameters of ANOVA.”

DETAILED COMMENTS

6) *Introduction. The introduction is a bit verbose. It devotes almost 2 pages (the entire page 5 and half of page 6) to summarizing the results of the paper. This is usually done in a brief paragraph. (On the other hand, the first paragraph of Results should really be in Introduction.)*

RESPONSE:

We thank the reviewer for this guidance and have made revisions accordingly. Please note that we have moved the first paragraph of the Results section of the originally submitted manuscript to the Introduction section and that we have shortened the summary of our results into a single, brief paragraph. The modified final paragraph of the Introduction is as follows (P. 5, Lines 1-12):

“Here, we examined the transcallosal synaptic circuits mediating the interhemispheric inhibition underlying unilateral cortical-damage-induced visuospatial bias in the mouse brain. Examining all of the medial and frontal cortices directly innervating the V1 revealed that unilateral lesion of the ACA induced the most severe detected visuospatial bias in a visual change-detection task. We then examined the roles of the ACA callosal-projection neurons (CPNs) and their downstream contralateral local circuits that enable mutual transcallosal inhibition between the two hemispheres in a healthy brain. Combining cell-type-specific calcium imaging, optogenetic manipulation and synaptic-circuit dissection, we found a transcallosal loop wherein ACA CPNs strongly activate contralateral PV+ neurons and callosal-input-driven PV+ (PV_{+cal}) neurons preferentially inhibit nearby CPNs within the same hemisphere, thus mediating transcallosal inhibition in visuospatial processing. Additionally, we show that lesion-induced visuospatial bias can be reversed by restoring interhemispheric balance through increased transcallosal inhibition in the contralesional hemisphere.”

7) *Figure 1. It would help to see a picture of the cortex from the top, indicating the exact locations of the 3 areas that were lesioned. This would be particularly useful for ACA, which is a vast region. The figure could indicate the part of ACA that is the subject of*

the experiments in this figure and in subsequent figures.

RESPONSE:

The ACA is situated beneath the secondary motor cortex (MOs), making it difficult to observe from the top. To better illustrate the location of the ACA lesion, we have plotted it on a flat cortical map, as well as the location of the RSP and PTLp lesions, which can be found in the Extended Data Fig. 1 (shown in our response to Reviewer #1's 5th comment).

8) Figure 2. The GRIN rod was presumably used to go deeper and reach ACA. But how deep? (apologies if this is already in Methods – I could not find it). Please show the histology in a supplementary figure.

RESPONSE:

Based on the reviewer's guidance, we have added pictures showing the location of GRIN lens in the Extended Data Figures (Extended Data Fig. 4, 12, and 13). Note that these figures are shown in our response to Reviewer #1's 5th comment. To clarify, we injected the virus expressing GCamp6s into the ACA at the following coordinates: Bregma, +0.3 mm; lateral, 0.3 mm; depth, 0.9 mm. After 1~2 weeks of virus injection, the GRIN lens was implanted 0.3-0.4 mm above the virus injection site.

9) Crucially, there seem to be almost as many ipsi-preferring neurons as contra-preferring. So why does the paper place so much emphasis on the former?

RESPONSE:

Kindly see our response to the initial comment regarding this topic, above (first comment).

10) In the pie charts, not sure what “visuomotor association” stands for. Does it mean that the neuron encodes both visual stimuli and movements?

RESPONSE:

Upon reanalyzing neuronal responses in miss and FA trials (see related response, above, 4th comment), we now know that ACA CPNs and PV^{+cal} neurons encode the properties of visual stimulus rather than the properties of motor action. Consequently, the greater differences in neuronal responses to contralateral versus ipsilateral visual

change in hit trials compared to miss trials are likely due to differences in behavioral states, rather than the combined encoding of visual stimuli and movements. Accordingly, we have removed the previous single-cell classification related to their activities in different types of trials and the corresponding pie charts.

11) *Figure 3. Here and in other figures that show the effects of laser stimulation, it would be good to have a consistent format. For instance, one could adopt the single, simplified format like the one in Figure 4e,f. The detailed breakdown session by session should definitely be in the paper, but perhaps it can be in Supplementary Figures.*

RESPONSE:

We thank the reviewer for this guidance; we have replaced the previous Fig. 3 with a new figure based on the format in Fig. 4e,f. And we have moved the detailed session-by-session breakdown data of previous Fig. 3 to revised Extended Data Fig. 6. Fig. 3 has been shown in our response to 3rd comment, and Extended Data Fig. 6 has been shown in the response to 4th comment.

12) *Figure 5. This seems to be another case of overreach in terms of conclusions: the figure shows that all inhibitory neurons (not just PV cells) get contralateral excitatory input.*

RESPONSE:

Kindly see our response to the initial comment regarding this topic, above (first comment).

13) *Figure 6. This figure shows that PV cells get particularly strong EPSCs from contralateral CPNs. But perhaps they get strong EPSCs from anyone, including ipsilateral CPNs? If so, it's another doubtful use of "preferential".*

RESPONSE:

We thank the reviewer for this excellent question. In brief, PV+, SST+ and VIP+ neurons receive distinct long-range inputs and serves different functions in modulating sensory processing (Fishell and Kepecs, 2020; Harris and Shepherd, 2015; Isaacson and Scanziani, 2011; Tremblay et al., 2016). To the best of our knowledge, the input strengths of ipsilateral CPNs to local interneurons have not been reported. We have added a new experiment to examine the input strengths of ipsilateral CPNs to these interneurons. Briefly, we found that the inputs from ipsilateral CPNs to three major

types of cortical interneurons are weak and non-selective (Extended Data Fig. 9). These results support the conclusion that PV+ neurons are specifically activated by callosal inputs.

In more detail, previous studies have shown that PV+, SST+, and VIP+ neurons form conserved cortical circuit motifs to enable rich and rapid modulations in local circuits (Jiang et al., 2015; Kim et al., 2017; Lee et al., 2013; Pfeffer et al., 2013; Pi et al., 2013). Each of these interneuron types has been shown to receive distinct long-range inputs and to exert distinct functions in modulating sensory processing (Fishell and Kepecs, 2020; Harris and Shepherd, 2015; Isaacson and Scanziani, 2011; Tremblay et al., 2016). For example, top-down corticocortical (CC) and thalamocortical (TC) inputs preferentially activate VIP+ neurons, whereas bottom-up CC and TC inputs preferentially activate PV+ neurons (Cruikshank et al., 2007; D'Souza et al., 2016; Lee et al., 2013; Ma et al., 2021; Wehr and Zador, 2003; Zhang et al., 2014).

To the best of our knowledge, the input strengths of ipsilateral CPNs to local interneurons have not been reported. In our visual change-detection task, we found that the activities of ipsilateral and contralateral CPNs are biased towards visual changes in the opposite visual fields. Understanding the input strengths of ipsilateral CPNs to specific populations of local interneurons can enable finer-resolution dissection of the functional contributions of local circuits in interhemispheric balance in visuospatial processing.

Please note that we have added a new experiment, described as follows (P. 15, Lines 3-21, and P. 16, Lines 1-2):

“Given that the ipsilateral CPNs may also provide excitatory inputs to local interneurons and exhibit an opposite activity bias to contralateral CPNs in the visual change-detection task, we investigated the input strengths from ipsilateral CPNs to ACA interneurons. To identify different types of interneurons, we co-injected Flp-dependent AAV expressing EYFP (AAV-EF1 α -fDIO-EYFP) and AAV expressing tdTomato driven by a PV-specific enhancer (AAV-S5E2-tdTomato) into the ACA of one hemisphere in SST-Flp and PV-Flp mice (Extended Data Fig. 9a). Consequently, SST+ or VIP+ neurons expressed EYFP, while PV+ neurons expressed tdTomato (Extended Data Fig. 9b). To target ipsilateral CPNs for optogenetic activation, Cre-inducible AAV expressing ChR2 (AAV-EF1 α -DIO-ChR2) was injected into the ACA together with labeling vectors for identifying distinct interneuron types; a retrograde AAV for Cre expression (Retro-AAV-hSyn-Cre) was injected into the opposite ACA. After confirming that optogenetic activation of ipsilateral CPNs induced large polysynaptic EPSPs in all three examined interneuron types without blocking local neuron spiking (Extended Data Fig. 9c), we applied TTX to suppress local neuronal spiking. Upon blocking, we found that optogenetic activation of

ipsilateral CPNs induced very weak monosynaptic EPSPs (mean EPSP amplitude < 1.2 mV) in all three interneuron types (Extended Data Fig. 9c). Notably, the amplitudes of both polysynaptic and monosynaptic EPSPs induced by the activation of ipsilateral CPNs were similar across all three interneuron types ($P > 0.07$, Wilcoxon rank-sum test). Thus, ipsilateral CPNs provide weak and non-selective inputs to all three examined interneuron types, whereas ACA PV+ neurons receive specific, strong inputs from contralateral CPNs; these findings reinforce the conclusion that inputs from contralateral CPNs preferentially activate PV+ neurons to provide transcallosal inhibition in visuospatial processing.”

“Extended Data Fig. 9. Ipsilateral CPNs provide non-selective and weak direct inputs to three major types of cortical interneurons in the ACA.

(a) Schematic of the slice experiment to measure ipsilateral CPN-induced EPSPs in ACA interneurons. Left, viral vectors and injection procedure for expressing ChR2 in ipsilateral CPNs and fluorescent proteins in cortical interneurons (PV+ neurons, tdTomato; SST+ or VIP+

neurons, EYFP). Right, electrophysiological recording. **(b)** Fluorescence images showing PV⁺ neurons with enhancer-driven expression of tdTomato (red) and SST⁺ or VIP⁺ neurons with fDIO-induced expression of EYFP (green). Top, red arrowheads, tdTomato-expressing PV⁺ neurons; white arrowheads, EYFP-expressing SST⁺ neurons. Bottom, red arrowheads, tdTomato-expressing PV⁺ neurons; white arrowheads, EYFP-expressing VIP⁺ neurons. **(c)** Ipsilateral CPN-induced polysynaptic (without TTX+4-AP) and monosynaptic (with TTX+4-AP) EPSP amplitudes in different cell types. The ipsilateral CPN-induced monosynaptic EPSPs are weak and non-selective across three major types of cortical interneurons (PV⁺ vs. SST⁺, $P = 0.08$; PV⁺ vs. VIP⁺, $P = 0.21$; SST⁺ vs. VIP⁺, $P = 0.07$; Wilcoxon rank-sum test). All data are presented as the mean \pm SEM.”

14) Figure 6. Panel jkl: isn't it expected that PV_{cal} neurons would receive more input from contralateral side than other PV neurons? They are called PV_{cal} because they receive contralateral input. It would be good to clarify this matter.

RESPONSE:

We agree with the reviewer that PV_{cal} neurons would be expected to receive stronger callosal inputs than other PV⁺ neurons, yet the extent of contralateral input specificity is unknown. Seeking to clarify our message, we have replaced the framing question “We next examined whether PV_{cal} neurons receive stronger callosal inputs than other PV⁺ neurons” with “To investigate the contralateral input specificity of PV_{cal} neurons, we measured the strength of callosal inputs to PV_{cal} neurons compared to other PV⁺ neuron types.” (P. 16, Lines 14-15).

15) Figure 7. This figure shows the activity of PV_{cal} neurons and demonstrates convincingly that they are more ipsi-preferring. However, it does not show us what those neurons do in the incorrect trials. Showing this would help us distinguish visual responses from motor responses.

RESPONSE:

Kindly see our response to the initial comment regarding this topic, above (4th comment).

16) Figure 8. This figure could open with the effects of activation of PV cells during the task, which are currently in Supp Fig 11. They could be brought to the main text if the

graphics are simplified (using the format of Figure 4e,f, for figures in main text).

RESPONSE:

We thank the reviewer for this guidance and have now revised Fig. 8 accordingly. Fig.8 has been shown in our response to 3rd comment.

17) It is surprising that there is no effect of bilateral PV+ activation (Figure 8 d,e). And yet, it must shut down both sides of ACA. Please explain in the paper how this can be the case (if there is an explanation).

RESPONSE:

Kindly see our response to the initial comment regarding this topic, above (2nd comment).

OTHER COMMENTS

18) Style. Some statistics seem to appear both in main text and figure legends, sometimes with subtle variations. This is confusing. It would be best to choose one place and be consistent about it.

RESPONSE:

We thank the reviewer for the suggestion. We have removed explicated statistics from the main text whenever possible; the details are now presented in the figure legends.

19) page 8: “indicating a role for the ACA in mediating interhemispheric visuospatial competition.” This sentence is premature because at this point we don’t know that there is competition between the two stimuli, and that the competition between stimuli involves a competition between hemispheres.

RESPONSE:

We thank the reviewer for pointing out this issue. We have removed “In contrast to the similar left and right performance in control mice, ACA lesion led to prolonged deficits of contralesional change detection at multiple contrast levels (Fig. 1i-1j), indicating a role for the ACA in mediating interhemispheric visuospatial competition” and now use the following alternative: “In contrast to the similar left and right

performance in control mice, ACA lesion led to prolonged deficits of contralesional change detection at multiple contrast levels (Fig. 1i-1j), indicating a severe visuospatial bias towards the ipsilesional side following ACA lesion.” (P. 7, Lines 1-4).

20) Page 21 “which has been shown to suppress pyramidal neuron spiking”. The method of shutting down cortex by activating PV cells was demonstrated in Supplementary Figure 8 of this paper: Olsen, S. R., D. S. Bortone, H. Adesnik and M. Scanziani (2012). "Gain control by layer six in cortical circuits of vision." *Nature* 483(7387): 47-52.

RESPONSE:

We now cite this paper in the revised manuscript (P. 19, Line 14).

21) The literature contains studies where frontal cortex does not appear to be lateralized (work by the Svoboda laboratory) and studies where it does seem to be lateralized (e.g. this paper, and Zatzka-Haas et al, *eLife* 2021). It would be good to clarify where results agree and disagree with these and other previous studies.

RESPONSE:

We now understand the need to discuss more about the lateralization of cortex, and we have added a paragraph to the Discussion section (quoted below). For context, regarding Svoboda’s studies, some support that the frontal motor cortex (ALM) is lateralized. For example, they have shown that in a whisker-based discrimination task with a delayed response, unilateral inhibition of the ALM and thalamus during the whole delay period (1.3 s) results in an ipsilateral bias, supporting a role for a lateralized thalamocortical network in motor preparation (Guo et al., 2017; Guo et al., 2014; Li et al., 2015). However, another study from his lab showed (for the same task) that transient unilateral inhibition of ALM during the early delay period (first 400 ms) or late delay period (last 500 ms) results in distinct behavioral effects (Li et al., 2016). Unilateral inhibition of the ALM during the late delay period results in an ipsilateral bias, whereas unilateral inhibition of the ALM during the early delay period has no obvious behavioral effects. They ultimately concluded that the neuronal activity in the inhibited ALM recovers rapidly after the inhibition during the early delay period due to compensatory activity in the contralateral ALM. These results suggest that premotor networks in the ALM are functionally modular; that is, unperturbed parts remain functional during perturbations and can aid in recovery of the perturbed parts (*i.e.*, not lateralized).

The ALM is known as the motor cortex controlling tongue movement in mice, and normal tongue movement invariably requires control commands from both

hemispheres, resulting in the left and right parts of the tongue invariably move together. Therefore, the information of motor action encoded by the two ALMs is similar. In our study, we observed that both ACA CPNs and PV_{+cal} neurons exhibit selective visual-related activity but do not display selective motor-related activity, indicating that they encode visual stimulus but not motor action.

Given this context, we want to highlight two major differences between ACA transcallosal interactions and ALM transcallosal interactions.

1. The contralateral bias in the population responses of callosal-projection neurons (IT neurons) during the task: contralateral bias vs. no contralateral bias.

Our study identified significantly more contralateral-preferring CPNs (IT neurons) than ipsilateral-preferring CPNs in the ACA, and the averaged activity of ACA CPNs exhibited a contralateral bias in response to visual changes in the visual change-detection task (Fig. 2).

In contrast, a previous Svoboda group study showed that the distribution of contralateral-preferring and ipsilateral-preferring neurons is roughly equal in the ALM, and found no significant contralateral bias in the overall spike count across the recorded population of ALM neurons in response to tongue movements in the whisker-based discrimination task (Li N, et al., 2015; Fig. 2c,e and Extended Data Fig. 1). Further cell-type-specific analysis revealed more contralateral-preferring than ipsilateral-preferring neurons in the PT neuron population, but not in the IT neuron population in the ALM (Li N, et al., 2015; Fig. 3 and 4).

Viewed together, these results indicate that callosal inputs (generated by IT neurons) from the ACA are stronger with contralateral visual change, whereas callosal inputs from the ALM are roughly equal for contralateral and ipsilateral tongue movements.

2. The net effect of callosal inputs: inhibitory vs. excitatory

Our study indicated that the net effect of ACA callosal inputs is inhibitory, because the behavioral changes induced by inhibiting callosal-projection axon terminals is similar to that caused by the activating CPNs innervated by these callosal projections (Fig. 3 and 4, and Extended Data Table1).

In contrast, Li N, et al. showed that the net effect of ALM callosal inputs is excitatory. Unilaterally inhibiting ALM results in a small reduction of contralateral ALM neuron spike rates (Li N, et al., 2016; Fig. 2c,d), indicating

a mild net excitatory effect of ALM callosal inputs.

Collectively, these findings from our study about ACA callosal projections, the studies from Svoboda's lab about ALM callosal projections, and the studies regarding the function of the V1 callosal projections (Conde-Ocazonez et al., 2018a; Conde-Ocazonez et al., 2018b; Ramachandra et al., 2020; Rochefort et al., 2007; Wunderle et al., 2015; Wunderle et al., 2013) support that callosal projections can induce net excitation or inhibition in their target areas. And no matter whether such projections are excitatory or inhibitory, unilateral inhibition of the sensory, motor, and frontal cortices results in an ipsilateral bias in an animal's performance on sensory-motor tasks, thus supporting consistent functional consequences from imbalanced interhemispheric interaction. Future studies delineating the area-specific callosal input strengths on different types of postsynaptic neurons across multiple cortical areas should bring clarity to the ongoing debate about whether callosal projections are excitatory or inhibitory in cortical circuits, and will likely resolve how they exert distinct functions in various sensory and behavioral contexts.

Regarding the first major difference between our work and Svoboda's work, we have added more content about the differential contralateral bias in IT and PT neurons in the revised Discussion section (P. 22, Lines 13-17):

“A previous study of the mouse anterior lateral motor cortex (ALM) showed that the proportion of contralateral-lick preferring neurons was higher among PT neurons than among IT neurons (Li et al., 2015). Although PT and CT neurons do not directly project to the contralateral hemisphere, they may affect interhemispheric balance through indirect pathways, e.g., cortico-thalamic-cortical pathways.”

Regarding the second major difference, we have added content to the revised Discussion section about the lateralization of cortical areas and the excitatory/inhibitory functions of callosal inputs between different cortical areas. The updated content in the revised manuscript reads as follows (P. 21, Lines 3-20, and P. 22, Lines 1-2):

“In mammals, both sensory and motor systems are highly lateralized. Unilateral lesions reduce ipsilesional network activity, disrupt the interaction balance between the two hemispheres, and bias the sensory and motor processing towards the ipsilesional side (Corbetta and Shulman, 2011; Iacoboni and Zaidel, 2004; Schulte and Muller-Oehring, 2010; Seabrook et al., 2017). Our results showed that unilateral inactivation of

the ACA in mice biased their performance toward the ipsilateral side in a visual-change detection task, consistent with previous studies on unilateral inhibition of sensory and motor cortices in various sensory-motor tasks (Burgess et al., 2017; Guo et al., 2014; Zatzka-Haas et al., 2021; Znamenskiy and Zador, 2013), thus supporting consistent functional consequences from imbalanced interhemispheric interaction.

Previous studies have shown that activation of callosal projections can drive excitation and/or inhibition in cortical circuits (Anastasiades et al., 2018; Bloom and Hynd, 2005; Karayannis et al., 2007; Lee et al., 2014; Palmer et al., 2012; Rock and Apicella, 2015; Slater and Isaacson, 2020). While the callosal projections between the primary visual cortices provide interhemispheric facilitation of visual response gain (Bocci et al., 2014; Conde-Ocazone et al., 2018; Ramachandra et al., 2020; Rochefort et al., 2007; Wunderle et al., 2015; Wunderle et al., 2013), the callosal connections between the frontal cortices allow interhemispheric inhibition in visuospatial processing (Corbetta and Shulman, 2011; Innocenti et al., 2022). Consistent with the callosal projections between the visual, somatosensory, auditory, retrosplenial, and prefrontal cortices (Adaikkan et al., 2022; Anastasiades et al., 2018; Karayannis et al., 2007; Naskar et al., 2021; Palmer et al., 2012; Slater and Isaacson, 2020), our results support that the callosal projections between the frontal cortices directly innervate both excitatory and inhibitory neurons. Since the behavioral effects induced by inhibition of callosal-projection axon terminals from the opposite ACA are similar to those caused by activating CPNs in the same hemisphere, the net impact of ACA callosal projections is inhibitory. Future studies delineating the area-specific callosal input strengths on different types of postsynaptic neurons across multiple cortical areas should bring clarity to the ongoing debate about whether callosal projections are excitatory or inhibitory in cortical circuits, and will likely resolve how they exert distinct functions.”

22) Typos: in page 12, Fig 6a and Fig 6b should be Fig 4a and Fig 4b

RESPONSE:

We thank the reviewer for point out these errors and have corrected them in the revision.

References

- Adaikkan, C., Wang, J., Abdelaal, K., Middleton, S.J., Bozzelli, P.L., Wickersham, I.R., McHugh, T.J., and Tsai, L.H. (2022). Alterations in a cross-hemispheric circuit associates with novelty discrimination deficits in mouse models of neurodegeneration. *Neuron* *110*, 3091-3105.
- Anastasiades, P.G., Marlin, J.J., and Carter, A.G. (2018). Cell-Type Specificity of Callosally Evoked Excitation and Feedforward Inhibition in the Prefrontal Cortex. *Cell. Rep.* *22*, 679-692.
- Bocci, T., Pietrasanta, M., Cerri, C., Restani, L., Caleo, M., and Sartucci, F. (2014). Visual callosal connections: role in visual processing in health and disease. *Rev. Neurosci.* *25*, 113-127.
- Burgess, C.P., Lak, A., Steinmetz, N.A., Zátka-Haas, P., Bai Reddy, C., Jacobs, E.A.K., Linden, J.F., Paton, J.J., Ranson, A., Schroder, S., *et al.* (2017). High-Yield Methods for Accurate Two-Alternative Visual Psychophysics in Head-Fixed Mice. *Cell. Rep.* *20*, 2513-2524.
- Conde-Ocazonez, S., Altavini, T.S., Wunderle, T., and Schmidt, K.E. (2018a). Motion contrast in primary visual cortex: a direct comparison of single neuron and population encoding. *Eur. J. Neurosci.* *47*, 358-369.
- Conde-Ocazonez, S.A., Jungen, C., Wunderle, T., Eriksson, D., Neuenschwander, S., and Schmidt, K.E. (2018b). Callosal Influence on Visual Receptive Fields Has an Ocular, an Orientation-and Direction Bias. *Front. Syst. Neurosci.* *12*, 11.
- Corbetta, M., and Shulman, G.L. (2011). Spatial neglect and attention networks. *Annu. Rev. Neurosci.* *34*, 569-599.
- Cruikshank, S.J., Lewis, T.J., and Connors, B.W. (2007). Synaptic basis for intense thalamocortical activation of feedforward inhibitory cells in neocortex. *Nat. Neurosci.* *10*, 462-468.
- D'Souza, R.D., Meier, A.M., Bista, P., Wang, Q., and Burkhalter, A. (2016). Recruitment of inhibition and excitation across mouse visual cortex depends on the hierarchy of interconnecting areas. *eLife* *5*, e19332.
- Dambeck, N., Sparing, R., Meister, I.G., Wienemann, M., Weidemann, J., Topper, R., and Boroojerdi, B. (2006). Interhemispheric imbalance during visuospatial attention investigated by unilateral and bilateral TMS over human parietal cortices. *Brain. Res.* *1072*, 194-199.
- Fishell, G., and Kepecs, A. (2020). Interneuron Types as Attractors and Controllers. *Annu. Rev. Neurosci.* *43*, 1-30.
- Guo, Z.V., Inagaki, H.K., Daie, K., Druckmann, S., Gerfen, C.R., and Svoboda, K. (2017). Maintenance of persistent activity in a frontal thalamocortical loop. *Nature* *545*, 181-186.
- Guo, Z.V., Li, N., Huber, D., Ophir, E., Gutnisky, D., Ting, J.T., Feng, G., and Svoboda, K. (2014). Flow of cortical activity underlying a tactile decision in mice. *Neuron* *81*, 179-194.
- Harris, K.D., and Shepherd, G.M. (2015). The neocortical circuit: themes and variations. *Nat. Neurosci.* *18*, 170-181.
- Hilgetag, C.C., Theoret, H., and Pascual-Leone, A. (2001). Enhanced visual spatial attention ipsilateral to rTMS-induced 'virtual lesions' of human parietal cortex. *Nat. Neurosci.* *4*, 953-957.
- Iacoboni, M., and Zaidel, E. (2004). Interhemispheric visuo-motor integration in humans: the role of the superior parietal cortex. *Neuropsychologia* *42*, 419-425.
- Innocenti, G.M., Schmidt, K., Milleret, C., Fabri, M., Knyazeva, M.G., Battaglia-Mayer, A., Aboitiz, F., Ptito, M., Caleo, M., Marzi, C.A., *et al.* (2022). The functional characterization of callosal connections. *Prog. Neurobiol.* *208*, 102186.
- Isaacson, J.S., and Scanziani, M. (2011). How inhibition shapes cortical activity. *Neuron* *72*, 231-

243.

Jiang, X., Shen, S., Cadwell, C.R., Berens, P., Sinz, F., Ecker, A.S., Patel, S., and Tolias, A.S. (2015). Principles of connectivity among morphologically defined cell types in adult neocortex. *Science* 350, aac9462.

Karayannis, T., Huerta-Ocampo, I., and Capogna, M. (2007). GABAergic and pyramidal neurons of deep cortical layers directly receive and differently integrate callosal input. *Cereb. Cortex* 17, 1213-1226.

Kim, Y., Yang, G.R., Pradhan, K., Venkataraju, K.U., Bota, M., Garcia Del Molino, L.C., Fitzgerald, G., Ram, K., He, M., Levine, J.M., *et al.* (2017). Brain-wide Maps Reveal Stereotyped Cell-Type-Based Cortical Architecture and Subcortical Sexual Dimorphism. *Cell* 171, 456-469.

Lee, A.T., Gee, S.M., Vogt, D., Patel, T., Rubenstein, J.L., and Sohal, V.S. (2014). Pyramidal neurons in prefrontal cortex receive subtype-specific forms of excitation and inhibition. *Neuron* 81, 61-68.

Lee, S., Kruglikov, I., Huang, Z.J., Fishell, G., and Rudy, B. (2013). A disinhibitory circuit mediates motor integration in the somatosensory cortex. *Nat. Neurosci.* 16, 1662-1670.

Li, N., Chen, T.W., Guo, Z.V., Gerfen, C.R., and Svoboda, K. (2015). A motor cortex circuit for motor planning and movement. *Nature* 519, 51-56.

Li, N., Daie, K., Svoboda, K., and Druckmann, S. (2016). Robust neuronal dynamics in premotor cortex during motor planning. *Nature* 532, 459-464.

Lomber, S.G., and Payne, B.R. (1996). Removal of two halves restores the whole: reversal of visual hemineglect during bilateral cortical or collicular inactivation in the cat. *Vis. Neurosci.* 13, 1143-1156.

Lomber, S.G., Payne, B.R., Hilgetag, C.C., and Rushmore, J. (2002). Restoration of visual orienting into a cortically blind hemifield by reversible deactivation of posterior parietal cortex or the superior colliculus. *Exp. Brain. Res.* 142, 463-474.

Ma, G. *et al.* Hierarchy in sensory processing reflected by innervation balance on cortical interneurons. *Sci. Adv.* 7, abf5676 (2021). Naskar, S., Qi, J., Pereira, F., Gerfen, C.R., and Lee, S. (2021). Cell-type-specific recruitment of GABAergic interneurons in the primary somatosensory cortex by long-range inputs. *Cell Rep.* 34, 108774.

Palmer, L.M., Schulz, J.M., Murphy, S.C., Ledergerber, D., Murayama, M., and Larkum, M.E. (2012). The cellular basis of GABA(B)-mediated interhemispheric inhibition. *Science* 335, 989-993.

Payne, B.R., Lomber, S.G., Villa, A.E., and Bullier, J. (1996). Reversible deactivation of cerebral network components. *Trends Neurosci.* 19, 535-542.

Pfeffer, C.K., Xue, M., He, M., Huang, Z.J., and Scanziani, M. (2013). Inhibition of inhibition in visual cortex: the logic of connections between molecularly distinct interneurons. *Nat. Neurosci.* 16, 1068-1076.

Pi, H.J., Hangya, B., Kvitsiani, D., Sanders, J.I., Huang, Z.J., and Kepecs, A. (2013). Cortical interneurons that specialize in disinhibitory control. *Nature* 503, 521-524.

Ramachandra, V., Pawlak, V., Wallace, D.J., and Kerr, J.N.D. (2020). Impact of visual callosal pathway is dependent upon ipsilateral thalamus. *Nat. Commun.* 11, 1889.

Rochefort, N.L., Buzas, P., Kisvarday, Z.F., Eysel, U.T., and Milleret, C. (2007). Layout of transcallosal activity in cat visual cortex revealed by optical imaging. *Neuroimage* 36, 804-821.

Rock, C., and Apicella, A.J. (2015). Callosal projections drive neuronal-specific responses in the mouse auditory cortex. *J. Neurosci.* 35, 6703-6713.

Rushmore, R.J., DeSimone, C., and Valero-Cabre, A. (2013). Multiple sessions of transcranial direct

current stimulation to the intact hemisphere improves visual function after unilateral ablation of visual cortex. *Eur. J. Neurosci.* *38*, 3799–3807.

Schulte, T., and Muller-Oehring, E.M. (2010). Contribution of callosal connections to the interhemispheric integration of visuomotor and cognitive processes. *Neuropsychol. Rev.* *20*, 174–190.

Seabrook, T.A., Burbridge, T.J., Crair, M.C., and Huberman, A.D. (2017). Architecture, Function, and Assembly of the Mouse Visual System. *Annu. Rev. Neurosci.* *40*, 499–538.

Slater, B.J., and Isaacson, J.S. (2020). Interhemispheric Callosal Projections Sharpen Frequency Tuning and Enforce Response Fidelity in Primary Auditory Cortex. *eNeuro* *7*.

Sprague, J.M. (1966). Interaction of cortex and superior colliculus in mediation of visually guided behavior in the cat. *Science* *153*, 1544–1547.

Tremblay, R., Lee, S., and Rudy, B. (2016). GABAergic Interneurons in the Neocortex: From Cellular Properties to Circuits. *Neuron* *91*, 260–292.

Valero-Cabre, A., Amengual, J.L., Stengel, C., Pascual-Leone, A., and Coubard, O.A. (2017). Transcranial magnetic stimulation in basic and clinical neuroscience: A comprehensive review of fundamental principles and novel insights. *Neurosci. Biobehav. Rev.* *83*, 381–404.

Valero-Cabre, A., Toba, M.N., Hilgetag, C.C., and Rushmore, R.J. (2020). Perturbation-driven paradoxical facilitation of visuo-spatial function: Revisiting the 'Sprague effect'. *Cortex* *122*, 10–39.

Vuilleumier, P., Hester, D., Assal, G., and Regli, F. (1996). Unilateral spatial neglect recovery after sequential strokes. *Neurology* *46*, 184–189.

Weddell, R.A. (2004). Subcortical modulation of spatial attention including evidence that the Sprague effect extends to man. *Brain. Cogn.* *55*, 497–506.

Wehr, M., and Zador, A.M. (2003). Balanced inhibition underlies tuning and sharpens spike timing in auditory cortex. *Nature* *426*, 442–446.

Wunderle, T., Eriksson, D., Peiker, C., and Schmidt, K.E. (2015). Input and output gain modulation by the lateral interhemispheric network in early visual cortex. *J. Neurosci.* *35*, 7682–7694.

Wunderle, T., Eriksson, D., and Schmidt, K.E. (2013). Multiplicative mechanism of lateral interactions revealed by controlling interhemispheric input. *Cereb. Cortex* *23*, 900–912.

Yao, S., Wang, Q., Hirokawa, K.E., Ouellette, B., Ahmed, R., Bomben, J., Brouner, K., Casal, L., Caldejon, S., Cho, A., *et al.* (2023). A whole-brain monosynaptic input connectome to neuron classes in mouse visual cortex. *Nat. Neurosci.* *26*, 350–364.

Zatka-Haas, P., Steinmetz, N.A., Carandini, M., and Harris, K.D. (2021). Sensory coding and the causal impact of mouse cortex in a visual decision. *Elife* *10*, e63163.

Zhang, S., Xu, M., Kamigaki, T., Hoang Do, J.P., Chang, W.C., Jenvay, S., Miyamichi, K., Luo, L., and Dan, Y. (2014). Selective attention. Long-range and local circuits for top-down modulation of visual cortex processing. *Science* *345*, 660–665.

Znamenskiy, P., and Zador, A.M. (2013). Corticostriatal neurons in auditory cortex drive decisions during auditory discrimination. *Nature* *497*, 482–485.

Reviewer #2 (reviewers' comments in italic):

NCOMMS-23-04432-T

A Frontal Transcallosal Inhibition Loop Mediates Interhemispheric Visuospatial Competition

Wang et al.

General Comments

While I can appreciate that brevity is often welcomed, this should not be at the expense of clarity. The authors often employ forms of expression, particularly in describing results, in which terms such as “ipsilateral” and “contralateral” are used in a “shorthand” manner. Although the intended interpretation may be apparent to the authors, this will not be the case for many readers. It will therefore be necessary for the authors to expand several descriptions, to reduce the burden on the reader, and remove the scope for ambiguity.

There is a good deal of repetition in the Introduction and Results sections. The former contains material that I would have expected to see in the Results section. The latter contains material that would be more appropriately located in the Introduction section.

In many ways, the reporting of the results of the statistical tests does not accord with the indications provided in the Reporting Summary. For example, neither the test statistics nor the associated degrees of freedom are provided. Effect sizes are not given. The Reporting Summary also appears to stipulate that the standard deviations, rather than the standard error) should be reported. This has not been done.

RESPONSE:

We are very grateful to the reviewer for the detailed revision comments, which have been very helpful as we revised our manuscript.

Specific Comments

Abstract

1) The authors state that in the context of a change-detection task “ACA callosal projection neurons (CPNs) were more active with contralateral changes than with ipsilateral changes” Without prior knowledge of this specific paradigm, it will not be possible for the reader to appreciate the dimension along which “contralateral changes”

are differentiated from “ipsilateral changes”. The text should be revised to make transparent the nature of the behavioural phenomenon that is manifest in a change-detection task and, in particular, how “contralateral” and “ipsilateral” are to be understood in this context.

RESPONSE:

We thank the reviewer for pointing out this issue. To enhance the clarity of the information regarding the behavioral task and the differences between contralateral changes and ipsilateral changes, we have specified in the Abstract section that this is a visual change-detection task, in which changes can occur in either the contralateral or ipsilateral visual field. We have replaced “In a change-detection task, ACA callosal-projection neurons (CPNs) were more active with contralateral changes than with ipsilateral changes.” with “In a visual change-detection task, ACA callosal-projection neurons (CPNs) were more active with contralateral visual field changes than with ipsilateral changes.” (P. 2, Lines 6-7).

2) *There is scope to enhance the precision of the writing, particularly with respect to statements such as “improved ipsilateral detection via callosal projections”, which imply a lack of conceptual clarity. In this instance, “ipsilateral detection” is a behavioural phenomenon, whereas “via callosal projections” is a statement concerning anatomy. That which is omitted is reference to the process that relates the two. Although treatment of process follows, the authors should give further consideration to the sequence in which information is provided, and make a greater effort to provide explanation, rather than a set of observations, with the reader being left to expend too much effort in order to draw the intended inferences. In this context, the concluding statement cannot be seen to follow in an obvious way from the preceding material.*

RESPONSE:

Thank you for the feedback on the need to provide more detailed descriptions and eliminate ambiguities in our manuscript. We have made several changes, including expanded the phrase “via callosal projections” to “via altering interhemispheric interaction through CPN callosal projections” to clarify that the CPN callosal projections modify interhemispheric interaction and are responsible for the observed behavioral changes.

The revised Abstract reads as follows:

“Interhemispheric communication through the corpus callosum is required for both sensory and cognitive processes. Impaired transcallosal inhibition causing interhemispheric imbalance is believed to underlie

visuospatial bias after frontoparietal cortical damage, but the synaptic circuits involved remain largely unknown. Here, we show that lesions in the mouse anterior cingulate area (ACA) cause severe visuospatial bias mediated by a transcallosal inhibition loop. In a visual change-detection task, ACA callosal-projection neurons (CPNs) were more active with contralateral visual field changes than with ipsilateral changes. Unilateral CPN inhibition impaired contralateral change detection but improved ipsilateral detection by altering interhemispheric interaction through callosal projections. CPNs strongly activated contralateral parvalbumin-positive (PV+) neurons, and callosal-input-driven PV+ neurons preferentially inhibited ipsilateral CPNs, thus providing transcallosal inhibition. Unilateral PV+ neuron activation caused a similar behavioral bias to contralateral CPN activation and ipsilateral CPN inhibition, and bilateral PV+ neuron activation eliminated this bias. Notably, restoring interhemispheric balance by activating contralesional PV+ neurons significantly improved contralesional detection in ACA-lesion animals. Thus, a frontal transcallosal inhibition loop comprising CPNs and callosal-input-driven PV+ neurons mediates interhemispheric balance in visuospatial processing, and enhancing contralesional transcallosal inhibition restores interhemispheric balance while also reversing lesion-induced bias.”

Introduction (text commencing on page 3)

3) The opening statement, “Animals receive a large amount of environmental information at any given moment”, implies a particular conception of the relationship between perception, action (and cognition), in which the organism is a passive recipient (rather than there being an active reciprocal relationship between perception and action). While it may be the case that the authors indeed adhere to the view that animals “receive a large amount of ... information” is the appropriate way in which to conceive of perception, they should perhaps consider that many readers will not share this view, and that the opening statement will thus have a rather jarring effect.

RESPONSE:

We thank the reviewer for pointing out this issue. We understand that the animals are not passively receiving environmental information and are rather actively processing it. We have accordingly replaced “Animals receive a large amount of environmental information at any given moment” with “Animals actively process a large amount of environmental information at any given moment” (P. 3, Lines 1-2).

4) (page 3, para 1). Some further work is required to explain the meaning of “along an anterior-posterior gradient” in the context of “callosal fibers mostly project to homotopic regions of the hemispheres”. Are the authors implying that the density (or some other property) of the callosal fibres varies along an anterior-posterior axis? If so, the property in question, and the polarity of the variation, should be stated.

RESPONSE:

We now understand the problem with the ambiguous “gradient” diction. Our intent here was to indicate that the callosal fibers connecting anterior cortices are positioned at the anterior part of the corpus callosum, while the fibers connecting posterior cortices are positioned at the posterior part of the corpus callosum. The revised text reads as follows (P. 3, Lines 5-6):

“The callosal fibers mostly project to homotopic regions of the hemispheres (e.g., frontal lobe to frontal lobe, etc.) along an anterior-posterior axis (Innocenti et al., 2022).”

5) (page 3, para 1). The expression “interhemispheric facilitation from redundant information” is cryptic and fails to convey useful information.

RESPONSE:

We thank the reviewer for pointing out this issue. To improve the clarity of this sentence, we have replaced “interhemispheric facilitation from redundant information” with “interhemispheric facilitation of binocular vision”. The full revised sentence reads as follows (P. 3, Lines 6-10):

“Clinical observations of patients with callosal degradation suggested distinctions between posterior and anterior regions, with the posterior callosal areas linking sensory regions of the two hemispheres were related to interhemispheric facilitation of binocular vision, while anterior callosal regions linking frontal and parietal regions were related to maintaining balanced interhemispheric inhibition in visuospatial processing (Bocci et al., 2014; Corbetta and Shulman, 2011; Innocenti et al., 2022).”

6) (page 3, para 1). In a similar vein, the expression “interhemispheric visuospatial competition” could be interpreted in very many ways. Greater precision is required.

RESPONSE:

Since we focused on the synaptic circuits underlying interhemispheric balance via transcallosal inhibition between the frontal cortices of the brain's two hemispheres, we have substituted "interhemispheric visuospatial competition" with "interhemispheric inhibition in visuospatial processing". The full revised sentence reads as follows (P. 3, Lines 6-10):

"Clinical observations of patients with callosal degradation suggested distinctions between posterior and anterior regions, with the posterior callosal areas linking sensory regions of the two hemispheres were related to interhemispheric facilitation of binocular vision, while anterior callosal regions linking frontal and parietal regions were related to maintaining balanced interhemispheric inhibition in visuospatial processing (Bocci et al., 2014; Corbetta and Shulman, 2011; Innocenti et al., 2022)."

7) (page 3, para 2). *Rather than using terms such as "right-sided sensory inputs" and "left-sided sensory inputs", perhaps the authors might refer to visual fields.*

RESPONSE:

Based on the reviewer's suggestion, we have replaced "the left cerebral hemisphere processing right-sided sensory inputs and the right cerebral hemisphere processing left-sided sensory inputs" with "the left cerebral hemisphere processing visual information from the right visual field and the right cerebral hemisphere processing information from the left visual field". The full revised sentence reads as follows (P. 3, Lines 11-13):

"In mammals, the function of the primary visual cortex (V1) are highly lateralized, with the left cerebral hemisphere processing visual information from the right visual field and the right cerebral hemisphere processing information from the left visual field (Seabrook et al., 2017)."

8) (page 4, para 2). *Since it is central to the entire endeavour, it is essential that the authors provided a clear description of the defining features of interhemispheric visuospatial competition", upon first use of this term. It cannot be assumed that the characteristics of this phenomenon will be known to most readers.*

This is a particular requirement, since the reader is then informed that a consequences of "unilateral damage of the parietal and frontal cortices" is to "bias visuospatial competition".

RESPONSE:

We thank the reviewer for pointing out this issue. Since we focused on the synaptic circuits underlying interhemispheric balance via transcallosal inhibition between the frontal cortices of the brain's two hemispheres, we have substituted "interhemispheric visuospatial competition" with "interhemispheric balance in visuospatial processing" to enhance message precision. We have made attendant changes throughout the revised manuscript.

Title: "A Frontal Transcallosal Inhibition Loop Mediates Interhemispheric Balance in Visuospatial Processing"

Abstract: "...Impaired transcallosal inhibition causing interhemispheric imbalance is believed to underlie visuospatial bias after frontoparietal cortical damage, but the synaptic circuits involved remain largely unknown. ...Notably, restoring interhemispheric balance by activating contralesional PV+ neurons significantly improved contralesional detection in ACA-lesion animals. Thus, a frontal transcallosal inhibition loop comprising CPNs and callosal-input-driven PV+ neurons mediates interhemispheric balance in visuospatial processing, and enhancing contralesional transcallosal inhibition restores interhemispheric balance while also reversing lesion-induced bias." (P. 2, Lines 2-4, and P. 2, Lines 13-18).

Introduction:

1. "...anterior callosal regions linking frontal and parietal regions were related to maintaining balanced interhemispheric inhibition in visuospatial processing." (P. 3, Lines 9-10).
2. "...However, very little is known about which transcallosal synaptic circuits mediate interhemispheric inhibition in visuospatial processing ..." (P. 4, Lines 4-5).
3. "...Impaired mutual transcallosal inhibition causing interhemispheric imbalance has been proposed as a pathophysiological mechanism underlying visuospatial bias after frontoparietal cortical damage. ..." (P. 4, Lines 12-14).
4. "...Here, we examined the transcallosal synaptic circuits mediating the interhemispheric inhibition underlying unilateral cortical-damage-induced visuospatial bias in the mouse brain. ..." (P. 5, Lines 1-2).
5. "...we found a transcallosal loop wherein ACA CPNs strongly activate contralateral PV+ neurons and callosal-input-driven PV+ (PV_{+cal}) neurons preferentially inhibit nearby CPNs within the same hemisphere, thus mediating transcallosal inhibition in visuospatial processing. Additionally, we show

that lesion-induced visuospatial bias can be reversed by restoring interhemispheric balance through increased transcallosal inhibition in the contralesional hemisphere.” (P. 5, Lines 8-12).

Results:

1. “...These results indicate that ACA lesion induces more severe visuospatial bias than RSP and PTLp lesions, so we subsequently focused on the interhemispheric balance between the ACA in the two hemispheres. ” (P. 6, Lines 17-19).
2. “...To assess the physiological contributions of callosal projections from the ACA to interhemispheric balance in visuospatial processing, we measured the task-related activity of ACA CPNs. ...” (P. 7, Lines 7-8).
3. “...ACA CPNs mediate interhemispheric inhibition in visuospatial processing via callosal projections...” (P. 9, Lines 8-9).
4. “...CPNs and PV_{+cal} neurons form a transcallosal inhibition loop mediating interhemispheric balance” (P. 13, Lines 18-19).
5. “...Together, our results demonstrate that ACA CPNs and PV_{+cal} neurons form a transcallosal loop providing transcallosal inhibition to maintain interhemispheric balance (Fig. 6m). Equal CPN activity in each hemisphere enable balanced transcallosal inhibition through activating contralateral PV_{+cal} neurons. However, if one hemisphere has higher CPN activity, it will provide stronger transcallosal inhibition to the other hemisphere, resulting in a bias in the interhemispheric balance.” (P. 17, Lines 2-7).
6. “...The behavioral bias induced by PV_{+cal} neuron activation was similar to that of contralateral CPN activation and ipsilateral CPN inhibition (Extended Data Table1), supporting the notion that contralateral CPNs activate PV_{+cal} neurons via callosal projections to generate transcallosal inhibition and in turn inhibit ipsilateral CPNs, thus mediate interhemispheric balance in visuospatial processing.” (P. 19, Lines 3-7).

Discussion:

1. “In this study, we found that a transcallosal inhibition loop in the mouse ACA—a frontal cortical area providing top-down modulation to visual cortices—mediates interhemispheric balance in visuospatial processing. ...” (P. 20, Lines 14-16).
2. “...Our successful demonstration of reversing lesion-induced deficits based on short-term activation of contralesional PV+

neurons to restore interhemispheric balance may inspire the development of innovative treatments for lesion-induced visuospatial bias based on enhancing transcallosal inhibition.” (P. 24, Lines 19-22).

9) (page 4, para 2). *The description of the behavioural consequences is fair enough (i.e., “... even leading to neglect of visual stimuli on the contralesional side ...”, however, appended to this is an assertion with respect to process (“due to the interrupted top-down control of contralesional visual information processing”), which is not argued, and cannot be seen to follow from the characteristics of the behavioural phenomenon.*

Indeed, it is not clear why this is necessary, given that follows is a more clearly specified proposition that relates to, “impaired mutual transcallosal inhibition” (which is much more closely related to the study that is then described”.

RESPONSE:

We have replaced “... unilateral damage of the parietal and frontal cortices can strongly bias visuospatial competition towards the ipsilesional side, even leading to neglect of visual stimuli on the contralesional side, largely due to the interrupted top-down contralesional visual information processing ...” with “... unilateral damage in a dorsal frontoparietal network that provides top-down control of visual processing results in interhemispheric imbalance and leads to severe visuospatial bias towards the ipsilesional side, even leading to neglect of visual stimuli on the contralesional side ...” (P. 4, Lines 8-10).

10) (page 4, para 3). *The authors then go on to state that, “we examined the transcallosal synaptic circuits mediating interhemispheric visuospatial competition in the mouse brain”. How does this relate to the behavioural consequences of “unilateral damage of the parietal and frontal cortices” just described? The key problem is that the authors have not defined the nature of interhemispheric visuospatial competition, nor have they made clear the hypothesised relationship between unilateral damage of the parietal and frontal cortices and interhemispheric visuospatial competition. The logic of the implied association may be apparent to the authors, however, it is not explained to the reader.*

RESPONSE:

We thank the reviewer for pointing out this problem. We have extensively re-worked the ending of the Introduction section and now explain to readers that unilateral

lesions of the parietal and frontal cortices induce interhemispheric imbalance and thereby bias visuospatial processing.

The revised text reads as follows (P. 4, Lines 7-20, and P. 5, Lines 1-12):

“Clinical observations in stroke patients, as well as lesion studies in monkeys, have shown that unilateral damage in a dorsal frontoparietal network that provides top-down control of visual processing results in interhemispheric imbalance and leads to severe visuospatial bias towards the ipsilesional side, even leading to neglect of visual stimuli on the contralesional side (Clark et al., 2015; Corbetta and Shulman, 2011; Gregoriou et al., 2014; Heilman and Valenstein, 1972; Monosov et al., 2011; Wardak et al., 2004). Specifically, unilateral lesions were found to reduce ipsilesional network activity, and a contralesional network with relatively higher activity can lead to a visuospatial bias towards the ipsilesional side (Buschman and Kastner, 2015; Corbetta and Shulman, 2011; Zebhauser et al., 2019). Impaired mutual transcallosal inhibition causing interhemispheric imbalance has been proposed as a pathophysiological mechanism underlying visuospatial bias after frontoparietal cortical damage (Corbetta and Shulman, 2011; Kinsbourne, 1977; van der Knaap and van der Ham, 2011).

Previous studies examining the medial and frontal cortices in the mouse visual network have established that the retrosplenial (RSP), posterior parietal (PTLp), and cingulate (ACA) cortices directly innervate visual cortices, an arrangement similar to the cortical areas known for the primate dorsal frontoparietal network (Harris et al., 2019; Ma et al., 2021; Oh et al., 2014; Yao et al., 2023; Zhang et al., 2014; Zingg et al., 2014). These higher-level cortical areas provided top-down modulation of visual information processing, and modulate different aspects of visually-guided goal-directed behavior in mice (Hu et al., 2019; Huda et al., 2020; Kim et al., 2021; Leinweber et al., 2017a; Licata et al., 2017; Makino and Komiyama, 2015; Norman et al., 2021; Song et al., 2017; Vann et al., 2009; Zhang et al., 2014).

Here, we examined the transcallosal synaptic circuits mediating the interhemispheric inhibition underlying unilateral cortical-damage-induced visuospatial bias in the mouse brain. Examining all of the medial and frontal cortices directly innervating the V1 revealed that unilateral lesion of the ACA induced the most severe detected visuospatial bias in a visual change-detection task. We then examined the roles of the ACA callosal-projection neurons (CPNs) and their downstream contralateral local circuits that enable mutual transcallosal inhibition between the two hemispheres in a healthy brain. Combining cell-type-specific calcium imaging, optogenetic manipulation, and synaptic-circuit dissection, we found a transcallosal loop wherein ACA CPNs strongly activate contralateral PV⁺ neurons and callosal-input-driven PV⁺ (PV^{+cal})

neurons preferentially inhibit nearby CPNs within the same hemisphere, thus mediating transcallosal inhibition in visuospatial processing. Additionally, we show that lesion-induced visuospatial bias can be reversed by restoring interhemispheric balance through increased transcallosal inhibition in the contralesional hemisphere.”

11) (page 5, para 1). It does make sense to assert that the cingulate cortex (ACA) is, within the mouse frontal cortex, the “the region with the most severe detected visuospatial bias following unilateral lesion”. Visuospatial bias is a behavioural phenomenon that is defined in terms of choices made by an animal. It is not a property of brain regions. The authors should give greater consideration to the manner in which they conceive of and describe the mechanisms that mediate the behavioural phenomena under consideration.

RESPONSE:

We appreciate the reviewer for bringing this mistake to our attention. We have now reconsidered how we conceive of and describe the mechanisms underlying the behavioral phenomena throughout the manuscript. For instance, we have replaced “Examining all of the medial and frontal cortices directly innervating the V1 revealed the ACA as the region with the most severe detected visuospatial bias following unilateral lesion.” with “... Examining all of the medial and frontal cortices directly innervating the V1 revealed that unilateral lesion of the ACA induced the most severe detected visuospatial bias in a visual change-detection task. ...” (P. 5, Lines 2-4).

12) Even though a larger number of words will be required to make the same point, I would urge the authors to rephrase/expand upon statements such as “were more active with contralateral changes”. For example, it might be stated that “ACA CPNs were more active when the stimulus that changed was located in the contralateral visual field”. This may seem a pedantic point, however, the aim should be to eliminate ambiguity. This point applies to all statements of a similar nature made throughout the manuscript.

RESPONSE:

Based on guidance from Reviewers #1 and #3, we have compressed the summary of our results in the Introduction section into one paragraph, and have dropped this ambiguous statement. In addition, we have changed the phrasing in the Abstract and Results sections.

Abstract:

“... In a visual change-detection task, ACA callosal-projection neurons (CPNs) were more active with contralateral visual field changes than with ipsilateral changes. ...” (P. 2, Lines 6-7).

Results:

1. “... The differences in neuronal responses to contralateral versus ipsilateral visual change were significantly greater in hit trials compared to miss trials ...” (P. 8, Lines 21-22).
2. “... Similar to CPNs, the differences in the responses of PV_{+cal} neurons to contralateral versus ipsilateral visual change were significantly greater in hit trials compared to miss trials ...” (P. 18, Lines 8-10).

13) (page 5, para 2). Please expand the statement, “callosal-input-driven PV₊ (PV_{+cal}) neurons preferentially inhibit ipsilateral CPNs”, to make the meaning of “inhibit ipsilateral” entirely unambiguous.

RESPONSE:

We have replaced “callosal-input-driven PV₊ (PV_{+cal}) neurons preferentially inhibit ipsilateral CPNs” with “callosal-input-driven PV₊ (PV_{+cal}) neurons preferentially inhibit nearby CPNs within the same hemisphere”. The full revised sentence reads as follows (P. 5, Lines 7-10):

“... Combining cell-type-specific calcium imaging, optogenetic manipulation, and synaptic-circuit dissection, we found a transcallosal loop wherein ACA CPNs strongly activate contralateral PV₊ neurons and callosal-input-driven PV₊ (PV_{+cal}) neurons preferentially inhibit nearby CPNs within the same hemisphere, thus mediating transcallosal inhibition in visuospatial processing. ...”

14) (page 5, para 2). As per previous comments, while I can appreciate that the authors are seeking economy of expression, this should not be at the expense of clarity. Please rephrase, “were more active with ipsilateral changes than with contralateral changes”. In this case, how is the reader to understand the meaning of “ipsi-preferring neurons”? This is not a statement with physiological meaning.

RESPONSE:

Kindly see our response to the initial comment regarding this topic, above (12th

comment).

Based on guidance from Reviewers #1 and #3, we have compressed the summary of our results in the Introduction section into one paragraph, and have dropped this ambiguous statement. In addition, we have changed the phrasing in the Abstract and Results sections. For instance, in Abstract section "... In a visual change-detection task, ACA callosal-projection neurons (CPNs) were more active with contralateral visual field changes than with ipsilateral changes. ..." (P. 2, Lines 6-7). We have also replaced "contra-preferring" and "ipsi-preferring" with "contralateral-preferring" and "ipsilateral-preferring", respectively throughout the revised manuscript.

Results

15) (page 6). Please "speak" more plainly. The precise intended meaning of statements such as "disrupt the balance of interaction" and "biased the visuospatial competition" cannot be determined.

RESPONSE:

Kindly note that we have taken guidance from Reviewer #1 to move related content into the Introduction section, and we have updated the phrasing to speak more plainly. The moved and revised content now reads as follows (P. 4, Lines 7-14):

"Clinical observations in stroke patients, as well as lesion studies in monkeys, have shown that unilateral damage in a dorsal frontoparietal network that provides top-down control of visual processing results in interhemispheric imbalance and leads to severe visuospatial bias towards the ipsilesional side, even leading to neglect of visual stimuli on the contralesional side (Clark et al., 2015; Corbetta and Shulman, 2011; Gregoriou et al., 2014; Heilman and Valenstein, 1972; Monosov et al., 2011; Wardak et al., 2004). Specifically, unilateral lesions were found to reduce ipsilesional network activity, and a contralesional network with relatively higher activity can lead to a visuospatial bias towards the ipsilesional side (Buschman and Kastner, 2015; Corbetta and Shulman, 2011; Zebhauser et al., 2019). Impaired mutual transcallosal inhibition causing interhemispheric imbalance has been proposed as a pathophysiological mechanism underlying visuospatial bias after frontoparietal cortical damage (Corbetta and Shulman, 2011; Kinsbourne, 1977; van der Knaap and van der Ham, 2011)."

16) (page 8). The meaning of the "±" and the figures that follow is nor made apparent.

Do the values represent the range, the standard deviation, the standard error, the confidence interval)?

RESPONSE:

We thank the reviewer for pointing out this issue. We have now explicated our use of standard error reporting in all of the relevant figure captions.

17) (page 8). Please provide the test statistic (e.g., F ratio), the associated degrees of freedom, and the exact associated p value. Ideally, an effect size statistic should be provided along with the corresponding confidence interval. This information was not provided in the Supplementary Materials. In the absence of such, it is not possible to determine the veracity of the statistical analyses. It is simply not sufficient to provide only probability values.

I note that the reporting of the results of the statistical tests does not accord with the indications provided in the Reporting Summary (which also appears to stipulate that the standard deviations, rather than the standard error) should be reported.

RESPONSE:

We have re-examined the statistical analysis methods and corrected previous errors. Briefly, the ANOVA are now performed across mice, and details regarding ANOVA, including the *F* ratio, the associated degrees of freedom, the exact associated *P* value, and partial η^2 for ANOVA are now provided in Extended Data Table 2. The new ANOVA done across mice concur with our initial conclusions.

As to the use of standard deviation (SD) and standard error (SE). To our understanding, the current *Reporting Summary* for statistics does not insist on use of SD (Reviewer Fig. 3), and SE appears frequently in many recent *Nature Communications* papers (Li et al., 2023; Lia et al., 2023; Printz et al., 2023; Riva et al., 2023; Yu et al., 2023). Considering *Nature Communications* doesn't specifically demand the use of SD, we've elected to maintain the use of SE.

Reporting Summary

Nature Portfolio wishes to improve the reproducibility of the work that we publish. This form provides structure for consistency and transparency in reporting. For further information on Nature Portfolio policies, see our Editorial Policies and the Editorial Policy Checklist.

Please do not complete any field with "not applicable" or n/a. Refer to the help text for what text to use if an item is not relevant to your study. For final submission: please carefully check your responses for accuracy; you will not be able to make changes later.

Statistics

For all statistical analyses, confirm that the following items are present in the figure legend, table legend, main text, or Methods section.

- | n/a | Confirmed |
|--------------------------|---|
| [ ] | [ ] The exact sample size (n) for each experimental group/condition, given as a discrete number and unit of measurement |
| [ ] | [ ] A statement on whether measurements were taken from distinct samples or whether the same sample was measured repeatedly |
| [ ] | [ ] The statistical test(s) used AND whether they are one- or two-sided
Only common tests should be described solely by name; describe more complex techniques in the Methods section. |
| [ ] | [ ] A description of all covariates tested |
| [ ] | [ ] A description of any assumptions or corrections, such as tests of normality and adjustment for multiple comparisons |
| [ ] | [ ] A full description of the statistical parameters including central tendency (e.g. means) or other basic estimates (e.g. regression coefficient) AND variation (e.g. standard deviation) or associated estimates of uncertainty (e.g. confidence intervals) |
| [ ] | [ ] For null hypothesis testing, the test statistic (e.g. F , t , r) with confidence intervals, effect sizes, degrees of freedom and P value noted
Give P values as exact values whenever suitable. |
| [ ] | [ ] For Bayesian analysis, information on the choice of priors and Markov chain Monte Carlo settings |
| [ ] | [ ] For hierarchical and complex designs, identification of the appropriate level for tests and full reporting of outcomes |
| [ ] | [ ] Estimates of effect sizes (e.g. Cohen's d , Pearson's r), indicating how they were calculated |

Our web collection on statistics for biologists contains articles on many of the points above.

Reviewer Fig. 3| Reporting summary for statistics.

18) (Figure 1 and supplementary figures). It is recommended that the error bars represent standard deviations, rather than standard error of the means. The Reporting Summary appears to indicate that this is a requirement.

RESPONSE:

Kindly see our response to the initial comment regarding this topic, above (17th comment).

19) In some cases, the number of animals included was extremely small. For example, in the legend to Fig 3, it is stated that, “ $n = 14$ mice” and “ $n = 8$ mice”. In Fig 4, it is “ $n = 5$ mice”. In Fig 5, “(Pyr, $n = 3$ mice; PV, $n = 5$; SST, $n = 3$; VIP, $n = 5$)”. In Extended Data Fig. 2, it is stated that “ $n = 8$ mice”, in Extended Data Fig. 5, it is “ $n = 6$ mice”, in Extended Data Fig. 6, it is “ $n = 3$ mice”, in Extended Data Fig. 10, it is “ $n = 7$ mice”, in Extended Data Fig. 11, it is “ $n = 6$ mice”. While it can be appreciated that there is a general desire to minimise the number of animals used in laboratory

research, the challenges to replicability that arise from small sample sizes, have been well documented.

RESPONSE:

We understand the concern about the use of small sample sizes in some parts of our study, yet the number of animals used in our research (ranging from 3 to 20 mice per experiment) aligns with established practices in mouse studies investigating sensory-motor behaviors like sensory-cue-guided decision-making.

To our understanding, the sample sizes we used fall within the typical range of mouse studies in the field, which is considerably fewer than psychophysical human studies (typically > 50 participants) but more than non-human primate studies (usually 2-3 monkeys). For instance, similar optogenetic studies of sensory-motor tasks, such as those by Adam *et al.* (2022), Hu and Dan (2022), Liu *et al.* (2021), Norman *et al.* (2021), and Yang *et al.* (2022), used comparable numbers of mice in each group as in our study's Fig. 3 and 4, Extended Data Fig. 5 and 11 (Adam *et al.*, 2022; Hu and Dan, 2022; Liu *et al.*, 2021; Norman *et al.*, 2021; Yang *et al.*, 2022). Likewise, studies involving RV-mediated retrograde tracing (as shown in our study's Fig. 5) or single-cell calcium imaging in sensory-motor tasks (as shown in our study's Extended Data Fig. 13) also used a similar number of mice (Breton-Provencher *et al.*, 2022; Chen *et al.*, 2022; Cho *et al.*, 2023; Farrell *et al.*, 2021; Krabbe *et al.*, 2019; Liu *et al.*, 2022; Mohammad *et al.*, 2021; Narikiyo *et al.*, 2020; Pardi *et al.*, 2020; Tan *et al.*, 2020; Yang *et al.*, 2023; Zhang *et al.*, 2021).

We appreciate the importance of minimizing the number of animals used in laboratory research, and our aim has been to balance this with the need for statistically robust results.

We would again like to express our gratitude to the reviewer for the helpful guidance about how to improve our study.

References

- Adam, E., Johns, T., and Sur, M. (2022). Dynamic control of visually guided locomotion through corticosubthalamic projections. *Cell Rep.* *40*, 111139.
- Bocci, T., Pietrasanta, M., Cerri, C., Restani, L., Caleo, M., and Sartucci, F. (2014). Visual callosal connections: role in visual processing in health and disease. *Rev. Neurosci.* *25*, 113-127.
- Breton-Provencher, V., Drummond, G., Feng, J., Li, Y., and Sur, M. (2022). Spatiotemporal dynamics of noradrenaline during learned behaviour. *Nature* *606*, 732-738.
- Buschman, T.J., and Kastner, S. (2015). From Behavior to Neural Dynamics: An Integrated Theory of Attention. *Neuron* *88*, 127-144.
- Chen, A., Malgady, J., Chen, L., Shi, K., Cheng, E., Plotkin, J., Ge, S., and Xiong, Q. (2022). Nigrostriatal dopamine pathway regulates auditory discrimination behavior. *Nat. Commun.* *13*, 5942.
- Cho, K., Shi, J., Phensy, A., Turner, M., and Sohal, V. (2023). Long-range inhibition synchronizes and updates prefrontal task activity. *Nature* *617*, 548-554.
- Clark, K., Squire, R.F., Merriki, Y., and Noudoost, B. (2015). Visual attention: Linking prefrontal sources to neuronal and behavioral correlates. *Prog. Neurobiol.* *132*, 59-80.
- Corbetta, M., and Shulman, G.L. (2011). Spatial neglect and attention networks. *Annu. Rev. Neurosci.* *34*, 569-599.
- Farrell, J., Lovett-Barron, M., Klein, P., Sparks, F., Gschwind, T., Ortiz, A., Ahanonu, B., Bradbury, S., Terada, S., Oijala, M., *et al.* (2021). Supramammillary regulation of locomotion and hippocampal activity. *Science* *374*, 1492-1496.
- Gregoriou, G.G., Rossi, A.F., Ungerleider, L.G., and Desimone, R. (2014). Lesions of prefrontal cortex reduce attentional modulation of neuronal responses and synchrony in V4. *Nat. Neurosci.* *17*, 1003-1011.
- Harris, J.A., Mihalas, S., Hirokawa, K.E., Whitesell, J.D., Choi, H., Bernard, A., Bohn, P., Caldejon, S., Casal, L., Cho, A., *et al.* (2019). Hierarchical organization of cortical and thalamic connectivity. *Nature* *575*, 195-202.
- Heilman, K.M., and Valenstein, E. (1972). Frontal lobe neglect in man. *Neurology* *22*, 660-664.
- Hu, F., and Dan, Y. (2022). An inferior-superior colliculus circuit controls auditory cue-directed visual spatial attention. *Neuron* *110*, 109-119.
- Hu, F., Kamigaki, T., Zhang, Z., Zhang, S., Dan, U., and Dan, Y. (2019). Prefrontal Corticotectal Neurons Enhance Visual Processing through the Superior Colliculus and Pulvinar Thalamus. *Neuron* *104*, 1141-1152.
- Huda, R., Sipe, G.O., Breton-Provencher, V., Cruz, K.G., Pho, G.N., Adam, E., Gunter, L.M., Sullins, A., Wickersham, I.R., and Sur, M. (2020). Distinct prefrontal top-down circuits differentially modulate sensorimotor behavior. *Nat. Commun.* *11*, 6007.
- Iacoboni, M., and Zaidel, E. (2004). Interhemispheric visuo-motor integration in humans: the role of the superior parietal cortex. *Neuropsychologia* *42*, 419-425.
- Innocenti, G.M., Schmidt, K., Milleret, C., Fabri, M., Knyazeva, M.G., Battaglia-Mayer, A., Aboitiz, F., Ptito, M., Caleo, M., Marzi, C.A., *et al.* (2022). The functional characterization of callosal connections. *Prog. Neurobiol.* *208*, 102186.
- Kim, J.H., Ma, D.H., Jung, E., Choi, I., and Lee, S.H. (2021). Gated feedforward inhibition in the frontal cortex releases goal-directed action. *Nat. Neurosci.* *24*, 1452-1464.
- Kinsbourne, M. (1977). Hemi-neglect and hemisphere rivalry. *Adv. Neurol.* *18*, 41-49.

Krabbe, S., Paradiso, E., d'Aquin, S., Bitterman, Y., Courtin, J., Xu, C., Yonehara, K., Markovic, M., Müller, C., Eichlisberger, T., *et al.* (2019). Adaptive disinhibitory gating by VIP interneurons permits associative learning. *Nat. Neurosci.* *22*, 1834-1843.

Leinweber, M., Ward, D.R., Sobczak, J.M., Attinger, A., and Keller, G.B. (2017). A Sensorimotor Circuit in Mouse Cortex for Visual Flow Predictions. *Neuron* *95*, 1420-1432.

Li, C., Saliba, N.B., Martin, H., Losurdo, N.A., Kolahdouzan, K., Siddiqui, R., Medeiros, D., and Li, W. (2023). Purkinje cell dopaminergic inputs to astrocytes regulate cerebellar-dependent behavior. *Nat. Commun.* *14*, 1613.

Lia, A., Sansevero, G., Chiavegato, A., Sbrissa, M., Pendin, D., Mariotti, L., Pozzan, T., Berardi, N., Carmignoto, G., Fasolato, C., *et al.* (2023). Rescue of astrocyte activity by the calcium sensor STIM1 restores long-term synaptic plasticity in female mice modelling Alzheimer's disease. *Nat. Commun.* *14*, 1590.

Licata, A.M., Kaufman, M.T., Raposo, D., Ryan, M.B., Sheppard, J.P., and Churchland, A.K. (2017). Posterior Parietal Cortex Guides Visual Decisions in Rats. *J. Neurosci.* *37*, 4954-4966.

Liu, S., Ye, M., Pao, G., Song, S., Jhang, J., Jiang, H., Kim, J., Kang, S., Kim, D., and Han, S. (2022). Divergent brainstem opioidergic pathways that coordinate breathing with pain and emotions. *Neuron* *110*, 857-873.

Liu, Y., Xin, Y., and Xu, N. (2021). A cortical circuit mechanism for structural knowledge-based flexible sensorimotor decision-making. *Neuron* *109*, 2009-2024.

Ma, G. *et al.* Hierarchy in sensory processing reflected by innervation balance on cortical interneurons. *Sci. Adv.* *7*, abf5676 (2021).

Makino, H., and Komiyama, T. (2015). Learning enhances the relative impact of top-down processing in the visual cortex. *Nat. Neurosci.* *18*, 1116-1122.

Mohammad, H., Senol, E., Graf, M., Lee, C., Li, Q., Liu, Q., Yeo, X., Wang, M., Laskaratos, A., Xu, F., *et al.* (2021). A neural circuit for excessive feeding driven by environmental context in mice. *Nat. Neurosci.* *24*, 1132-1141.

Monosov, I.E., Sheinberg, D.L., and Thompson, K.G. (2011). The effects of prefrontal cortex inactivation on object responses of single neurons in the inferotemporal cortex during visual search. *J. Neurosci.* *31*, 15956-15961.

Narikiyo, K., Mizuguchi, R., Ajima, A., Shiozaki, M., Hamanaka, H., Johansen, J., Mori, K., and Yoshihara, Y. (2020). The claustrum coordinates cortical slow-wave activity. *Nat. Neurosci.* *23*, 741-753.

Norman, K.J., Riceberg, J.S., Koike, H., Bateh, J., McCraney, S.E., Caro, K., Kato, D., Liang, A., Yamamuro, K., Flanigan, M.E., *et al.* (2021). Post-error recruitment of frontal sensory cortical projections promotes attention in mice. *Neuron* *109*, 1202-1213.

Oh, S.W., Harris, J.A., Ng, L., Winslow, B., Cain, N., Mihalas, S., Wang, Q., Lau, C., Kuan, L., Henry, A.M., *et al.* (2014). A mesoscale connectome of the mouse brain. *Nature* *508*, 207-214.

Pardi, M., Vogenstahl, J., Dalmy, T., Spanò, T., Pu, D., Naumann, L., Kretschmer, F., Sprekeler, H., and Letzkus, J. (2020). A thalamocortical top-down circuit for associative memory. *Science* *370*, 844-848.

Printz, Y., Patil, P., Mahn, M., Benjamin, A., Litvin, A., Levy, R., Bringmann, M., and Yizhar, O. (2023). Determinants of functional synaptic connectivity among amygdala-projecting prefrontal cortical neurons in male mice. *Nat. Commun.* *14*, 1667.

Riva, M., Moriceau, S., Morabito, A., Dossi, E., Sanchez-Bellot, C., Azzam, P., Navas-Olive, A., Gal, B., Dori, F., Cid, E., *et al.* (2023). Aberrant survival of hippocampal Cajal-Retzius cells leads to

memory deficits, gamma rhythmopathies and susceptibility to seizures in adult mice. *Nat. Commun.* *14*, 1531.

Schulte, T., and Muller-Oehring, E.M. (2010). Contribution of callosal connections to the interhemispheric integration of visuomotor and cognitive processes. *Neuropsychol. Rev.* *20*, 174-190.

Seabrook, T.A., Burbridge, T.J., Crair, M.C., and Huberman, A.D. (2017). Architecture, Function, and Assembly of the Mouse Visual System. *Annu. Rev. Neurosci.* *40*, 499-538.

Song, Y.H., Kim, J.H., Jeong, H.W., Choi, I., Jeong, D., Kim, K., and Lee, S.H. (2017). A Neural Circuit for Auditory Dominance over Visual Perception. *Neuron* *93*, 1236-1237.

Tan, H., Sisti, A., Jin, H., Vignovich, M., Villavicencio, M., Tsang, K., Goffer, Y., and Zuker, C. (2020). The gut-brain axis mediates sugar preference. *Nature* *580*, 511-516.

van der Knaap, L.J., and van der Ham, I.J. (2011). How does the corpus callosum mediate interhemispheric transfer? A review. *Behav. Brain. Res.* *223*, 211-221.

Vann, S.D., Aggleton, J.P., and Maguire, E.A. (2009). What does the retrosplenial cortex do? *Nat. Rev. Neurosci.* *10*, 792-802.

Wardak, C., Olivier, E., and Duhamel, J.R. (2004). A deficit in covert attention after parietal cortex inactivation in the monkey. *Neuron* *42*, 501-508.

Yang, T., Yu, K., Zhang, X., Xiao, X., Chen, X., Fu, Y., and Li, B. (2023). Plastic and stimulus-specific coding of salient events in the central amygdala. *Nature* *616*, 510-519.

Yang, W., Tipparaju, S., Chen, G., and Li, N. (2022). Thalamus-driven functional populations in frontal cortex support decision-making. *Nat. Neurosci.* *25*, 1339-1352.

Yao, S., Wang, Q., Hirokawa, K.E., Ouellette, B., Ahmed, R., Bomben, J., Brouner, K., Casal, L., Caldejon, S., Cho, A., *et al.* (2023). A whole-brain monosynaptic input connectome to neuron classes in mouse visual cortex. *Nat. Neurosci.* *26*, 350-364.

Yu, Y., Qiu, Y., Li, G., Zhang, K., Bo, B., Pei, M., Ye, J., Thompson, G.J., Cang, J., Fang, F., *et al.* (2023). Sleep fMRI with simultaneous electrophysiology at 9.4 T in male mice. *Nat. Commun.* *14*, 1651.

Zebhauser, P.T., Vernet, M., Unterburger, E., and Brem, A.K. (2019). Visuospatial Neglect - a Theory-Informed Overview of Current and Emerging Strategies and a Systematic Review on the Therapeutic Use of Non-invasive Brain Stimulation. *Neuropsychol. Rev.* *29*, 397-420.

Zhang, G., Shen, L., Tao, C., Jung, A., Peng, B., Li, Z., Zhang, L., and Tao, H. (2021). Medial preoptic area antagonistically mediates stress-induced anxiety and parental behavior. *Nat. Neurosci.* *24*, 516-528.

Zhang, S., Xu, M., Kamigaki, T., Hoang Do, J.P., Chang, W.C., Jenvay, S., Miyamichi, K., Luo, L., and Dan, Y. (2014). Selective attention. Long-range and local circuits for top-down modulation of visual cortex processing. *Science* *345*, 660-665.

Zingg, B., Hintiryan, H., Gou, L., Song, M.Y., Bay, M., Bienkowski, M.S., Foster, N.N., Yamashita, S., Bowman, I., Toga, A.W., *et al.* (2014). Neural networks of the mouse neocortex. *Cell* *156*, 1096-1111.

Reviewer #3 (reviewers comments in italic):

Review of manuscript NCOMMS-23-04432-T

The present manuscript describes experiments manipulating callosal projecting neurons (CPN) and their targets in the anterior cingulate area (ACA) of awake mice through optogenetic excitation and inhibition while performing a two-alternative unforced-change detection task involving a motor response (turning a wheel) to indicate the discrimination of gratings versus noise stimuli placed in either of the two monocular visual fields of the mouse.

Besides the behavior, the activity of the neurons of the awake animal was monitored by calcium-imaging to demonstrate that ACA-CPNs are indeed associated with the visuomotor task. Unilateral ACA lesion or inhibition of ACA-CPN impaired contralateral visual change detection (independent of the contrast, i.e. visual neglect) and while improving ipsilateral change detection indicative of release of transcallosally mediated inhibition during this manipulation.

Using transmonosynaptic viral tracing, the ACA-CPN were found to project preferentially (60 % of the contralateral targets) to the contralateral ACA innervating both excitatory and inhibitory targets, i.e. pyramidal neurons, parvalbumin-positive (PV+), somatostatin-positive (SST+) and vasoactive intestinal peptide-positive (VIP+) neurons.

Subsequently, a CPN driven loop activating contralateral PV+ neurons which in turn inhibit ipsilateral excitatory CPNs was revealed by cell-type specific dissection of synaptic circuits in coronal brain slices.

Conformingly, unilateral activation of PV+ neurons in the ACA had similar behavioral effects as the unilateral inhibition of the excitatory CPNs, namely the contralateral neglect-like visuospatial impairment, a bias which was eliminated during bilateral PV+ neuron activation. Finally, a short-term activation of the contralateral PV+ neurons during learning of the change detection task improved the performance of the ACA-lesioned animals both during and after learning.

The authors conclude that they challenged and identified with their experiments a transcallosal loop including ipsilateral excitatory neurons in the anterior cingulate cortex impinging directly and indirectly on contralateral PV+ interneurons which modulate in turn the excitatory ipsilateral tonus and thus again the contralateral inhibition. They assume that the mouse ACA circuit provides top-down modulation on visual cortices and mediates interhemispheric visuospatial “competition” via the demonstrated transcallosal inhibitory loop.

The study is methodologically sound, comprehensive and elegant, the results are convincing and well documented as far as they concern the involvement of the mouse

anterior cingulate cortex in visuomotor integration or visuospatial attention. The intended link to interhemispheric competition in humans might not be appropriate.

Major critiques

1. Title, abstract and following text. "...visuospatial competition". I would not use the word competition, neither in the title nor in the abstract or discussion.

The authors claim to identify a loop mediating visuospatial competition but their task is actually not suited to characterize competition between the two hemifields. In my view, that would be a task demonstrating that two visual stimuli of equal salience (i.e. grating of the same SF and contrast) (such as in binocular rivalry, which does not exist here) compete for visual attention, or because of being represented in different hemispheres (such as in the Stroop-match-to sample task where the hemispheres are supposed to be in a "competitive" situation).

Being trained to detect a grating and rotate a wheel, I rather assume that the mouse deals with top-down visuomotor integration and visuospatial attention, and the loop which is identified is a reciprocal callosal excitatory-inhibitory loop.

The idea of visuospatial competition is driven from the human point of view who has access to the same central visual information in both hemispheres. The mouse might not even have this, and moreover, the task actively avoids the tiny rodent binocular visual field. It is thus possibly better to speak of interhemispheric cooperation (involving transcallosal inhibition) or excitatory-inhibitory balance instructing downstream motor, extrastriate and subcortical areas.

RESPONSE:

We appreciate this input, and since we focused on the synaptic circuits underlying interhemispheric balance via transcallosal inhibition between the frontal cortices of the brain's two hemispheres, we have now replaced "interhemispheric visuospatial competition" phrasing with "interhemispheric balance in visuospatial processing". We have made attendant changes throughout the revised manuscript.

Title: "A Frontal Transcallosal Inhibition Loop Mediates Interhemispheric Balance in Visuospatial Processing"

Abstract: "...Impaired transcallosal inhibition causing interhemispheric imbalance is believed to underlie visuospatial bias after frontoparietal cortical damage, but the synaptic circuits involved remain largely unknown. ...Notably, restoring interhemispheric balance by activating contralesional PV+ neurons significantly improved

contralesional detection in ACA-lesion animals. Thus, a frontal transcallosal inhibition loop comprising CPNs and callosal-input-driven PV+ neurons mediates interhemispheric balance in visuospatial processing, and enhancing contralesional transcallosal inhibition restores interhemispheric balance while also reversing lesion-induced bias.” (P. 2, Lines 2-4, and P. 2, Lines 13-18).

Introduction:

1. “...anterior callosal regions linking frontal and parietal regions were related to maintaining balanced interhemispheric inhibition in visuospatial processing.” (P. 3, Lines 9-10).
2. “...However, very little is known about which transcallosal synaptic circuits mediate interhemispheric inhibition in visuospatial processing ...” (P. 4, Lines 4-5).
3. “...Impaired mutual transcallosal inhibition causing interhemispheric imbalance has been proposed as a pathophysiological mechanism underlying visuospatial bias after frontoparietal cortical damage. ...” (P. 4, Lines 12-14).
4. “...Here, we examined the transcallosal synaptic circuits mediating the interhemispheric inhibition underlying unilateral cortical-damage-induced visuospatial bias in the mouse brain. ...” (P. 5, Lines 1-2).
5. “...we found a transcallosal loop wherein ACA CPNs strongly activate contralateral PV+ neurons and callosal-input-driven PV+ (PV_{+cal}) neurons preferentially inhibit nearby CPNs within the same hemisphere, thus mediating transcallosal inhibition in visuospatial processing. Additionally, we show that lesion-induced visuospatial bias can be reversed by restoring interhemispheric balance through increased transcallosal inhibition in the contralesional hemisphere.” (P. 5, Lines 8-12).

Results:

1. “...These results indicate that ACA lesion induces more severe visuospatial bias than RSP and PTLp lesions, so we subsequently focused on the interhemispheric balance between the ACA in the two hemispheres. ” (P. 6, Lines 17-19).
2. “...To assess the physiological contributions of callosal projections from the ACA to interhemispheric balance in visuospatial processing, we measured the task-related activity of ACA CPNs. ...” (P. 7, Lines 7-8).
3. “...ACA CPNs mediate interhemispheric inhibition in visuospatial processing via callosal projections...” (P. 9, Lines 8-9).

4. "...CPNs and PV_{+cal} neurons form a transcallosal inhibition loop mediating interhemispheric balance" (P. 13, Lines 18-19).
5. "...Together, our results demonstrate that ACA CPNs and PV_{+cal} neurons form a transcallosal loop providing transcallosal inhibition to maintain interhemispheric balance (Fig. 6m). Equal CPN activity in each hemisphere enable balanced transcallosal inhibition through activating contralateral PV_{+cal} neurons. However, if one hemisphere has higher CPN activity, it will provide stronger transcallosal inhibition to the other hemisphere, resulting in a bias in the interhemispheric balance." (P. 17, Lines 2-7).
6. "...The behavioral bias induced by PV_{+cal} neuron activation was similar to that of contralateral CPN activation and ipsilateral CPN inhibition (Extended Data Table1), supporting the notion that contralateral CPNs activate PV_{+cal} neurons via callosal projections to generate transcallosal inhibition and in turn inhibit ipsilateral CPNs, thus mediate interhemispheric balance in visuospatial processing." (P. 19, Lines 3-7).

Discussion:

1. "In this study, we found that a transcallosal inhibition loop in the mouse ACA—a frontal cortical area providing top-down modulation to visual cortices—mediates interhemispheric balance in visuospatial processing. ..." (P. 20, Lines 14-16).
2. "...Our successful demonstration of reversing lesion-induced deficits based on short-term activation of contralesional PV₊ neurons to restore interhemispheric balance may inspire the development of innovative treatments for lesion-induced visuospatial bias based on enhancing transcallosal inhibition." (P. 24, Lines 19-22).

Most of the references in the introduction about an assumed visuospatial competition are rather general and human-specific and do not introduce what is known about the role of the anterior cingulate cortex in mice.

RESPONSE:

In our revised Introduction, we now provide more context on the roles of the ACA, RSP, and PTLp cortices in the mouse visual network. In the revised Discussion, we present ideas about ACA functions in greater depth to help contextualize our findings that ACA lesions cause more severe visuospatial bias as compared to RSP and PTLp lesions.

The revised content in the Introduction reads as follows (P. 4, Lines 15-20):

“Previous studies examining the medial and frontal cortices in the mouse visual network have established that the retrosplenial (RSP), posterior parietal (PTLp), and cingulate (ACA) cortices directly innervate visual cortices, an arrangement similar to the cortical areas known for the primate dorsal frontoparietal network (Harris et al., 2019; Ma et al., 2021; Oh et al., 2014; Yao et al., 2023; Zhang et al., 2014; Zingg et al., 2014). These higher-level cortical areas provided top-down modulation of visual information processing, and modulate different aspects of visually-guided goal-directed behavior in mice (Hu et al., 2019; Huda et al., 2020; Kim et al., 2021; Leinweber et al., 2017a; Licata et al., 2017; Makino and Komiyama, 2015; Norman et al., 2021; Song et al., 2017; Vann et al., 2009; Zhang et al., 2014).”

The revised content in the Discussion reads as follows (P. 23, Lines 19-21, and P. 24, Lines 1-8):

“The mouse ACA is anatomically well-positioned to provide top-down control signals for visual information processing. It is the only frontal cortical area that receives direct inputs from the primary visual cortex and provided a substantial amount of direct axon innervation to the primary visual cortex (Harris et al., 2019; Ma et al., 2021; Zhang et al., 2016). The ACA has been shown to modulate different aspects of visually-guided goal-directed behavior in mice (Hu et al., 2019; Huda et al., 2020; Kim et al., 2021; Leinweber et al., 2017a; Norman et al., 2021; Zhang et al., 2014). Activation of the ACA improved visual feature detection—a hallmark feature of visual attention (Hu et al., 2019; Leinweber et al., 2017b; Norman et al., 2021; Zhang et al., 2014). Our results show that activation of ACA CPNs improves the ability to detect contralateral visual changes and decreases the reaction time to contralateral changes, findings consistent with the effects of attention modulation in primates (Desimone and Duncan, 1995; Fiebelkorn and Kastner, 2020; Moore and Zirnsak, 2017). The functional characterization of callosal projections between frontal areas is still in its infancy (Innocenti et al., 2022), and our study paves the way for subsequent detailed analyses of specific functional impacts of frontal callosal projections on higher cognitive functions, potentially with cell-type-specific resolution.”

Citations 10-12 do not fit for “visuospatial competition” either. Note that the almost identical argument is repeated in the beginning of the discussion on page 22.

RESPONSE:

We have now revised the citation errors in the manuscript. As the reviewer pointed out in a previous comment, the concept of visuospatial competition, while well-studied in primates, is not well defined in mice. Since we focused on the synaptic circuits underlying interhemispheric balance via transcallosal inhibition between the frontal cortices of the brain's two hemispheres, we have replaced "... anterior callosal regions linking frontal and parietal regions were related to interhemispheric visuospatial competition" with "... anterior callosal regions linking frontal and parietal regions were related to maintain balanced interhemispheric inhibition in visuospatial processing, e.g., visuospatial attention". We change the citations here to three papers reviewing the functions of posterior and anterior callosal connections in visual information processing.

To address the redundancy of arguments, we have revised the text in the Discussion section to focus on callosal functions revealed from animal models.

The revised Introduction text reads as follows (P. 3, Lines 6-10):

"Clinical observations of patients with callosal degradation suggested distinctions between posterior and anterior regions, with the posterior callosal areas linking sensory regions of the two hemispheres were related to interhemispheric facilitation of binocular vision, while anterior callosal regions linking frontal and parietal regions were related to maintaining balanced interhemispheric inhibition in visuospatial processing (Bocci et al., 2014; Corbetta and Shulman, 2011; Innocenti et al., 2022)."

And the revised Discussion text reads as follows (P. 21, Lines 10-13):

"Previous studies have shown that activation of callosal projections can drive excitation and/or inhibition in cortical circuits (Anastasiades et al., 2018; Bloom and Hynd, 2005; Karayannis et al., 2007; Lee et al., 2014; Palmer et al., 2012; Rock and Apicella, 2015; Slater and Isaacson, 2020). While the callosal projections between the primary visual cortices provide interhemispheric facilitation of visual response gain (Bocci et al., 2014; Conde-Ocazonez et al., 2018; Ramachandra et al., 2020; Rochefort et al., 2007; Wunderle et al., 2015; Wunderle et al., 2013), the callosal connections between the frontal cortices allow interhemispheric inhibition in visuospatial processing (Corbetta and Shulman, 2011; Innocenti et al., 2022)."

2. Page 3, last paragraph: I suggest to reformulate the sentence “... enhance visual information processing ...” to “Recent studies have shown that also in rodents VI callosal projections contribute to visual information processing....”.

RESPONSE:

We thank the reviewer for the suggestion and have adjusted that sentence accordingly (P. 3, Lines 18-19).

First of all functions, connections between the primary visual cortices in lateral-eyed mammals such as mice most likely and inevitably link the two halves of the visual field, because - in contrast to humans - more than 90% of the retinal fibres, i.e. the primary visual input, cross.

Second, noteworthy, many of the visual callosal functions listed here as specific for rodents, and more, have been detailed in frontal-eyed mammals such as cats and monkeys previously. Strangely, sometimes these studies are not put in the right context or not even cited in the more recent rodent studies, although they are much more similar to the situation in humans.

In this manuscript, it is important to identify and discuss the possible differences between the visual system of rodents and humans, and their implications for the present findings. Visual input from monocular and binocular fields in rodents and humans might not be of similar behavioral significance. It is likely that mice might not use the same visual strategies as primates. For one, rodent VI probably receives already much more multisensory input.

There exists the possibility that the slight performance deficit in the monocular field of a mouse induced by manipulating mouse cingulate cortex does not have a human homologue. Maybe it is not accurate to compare it to the human visual neglect situation following stroke.

RESPONSE:

We have added a paragraph in the Discussion section about the differences between the visual system of rodents and humans and cautioning the need for care in making any extrapolations from our results in mice to a human medical context. We have also dropped content from the originally submitted manuscript that compared our results to human visual neglect patients.

The revised Discussion text reads as follows (P. 24, Lines 9-22):

“Mouse models have been used to examine the roles of specific cell types and circuits in visual perception and behavior. While the visual systems of mice and humans share certain aspects, significant differences should be noted: humans have better visual acuity and color vision, while mice have a wider field of view and better low-light vision (Glickfeld and Olsen, 2017; Seabrook et al., 2017; Tehovnik et al., 2021). Owing to the lateral eye placement in mice, they have larger monocular zones yet a smaller binocular zone compared to humans (Seabrook et al., 2017; Tehovnik et al., 2021). Moreover, mice have fewer specialized brain areas for parallel visual information processing (Glickfeld and Olsen, 2017), and multisensory integration is known to occur at an earlier stage in mice (Laramee and Boire, 2014). Although results obtained from mice cannot be directly extrapolated to humans or non-human primates, the use of enhancer-driven viral tools for PV+ neurons allows for selective targeting and manipulation of specific interneuron types across species (Vormstein-Schneider et al., 2020), thereby providing the possibility to examine whether the cell-type-specific mechanisms of transcallosal inhibition in visuospatial processing revealed here in mice also function in primates. Our successful demonstration of reversing lesion-induced deficits based on short-term activation of contralesional PV+ neurons to restore interhemispheric balance may inspire the development of innovative treatments for lesion-induced visuospatial bias based on enhancing transcallosal inhibition.”

Minor critiques

- *Line numbers would be useful.*

RESPONSE:

Based on the reviewer’s suggestion, we have added line numbers in the revised manuscript.

- *In the first introduction of the abbreviation, you might want to state explicitly ACA for “anterior cingulate area”.*

RESPONSE:

We thank the reviewer for the helpful suggestion and have now explicated this abbreviation in the abstract (P. 2, Lines 4-5).

- Page 3, last paragraph: *...enhance the gain of the visual response*"18,19,20. For clarification, complete the sentence with "*...enhance the gain of the visual response in function of the retino-geniculo-cortical input*". 21. Cite here also Rochefort et al., 2007.

RESPONSE:

We have added Rochefort et al., 2007 citation to the revised text, and modified the sentence as "Previous studies in cats have shown that V1 callosal projections mainly activate contralateral excitatory neurons and enhance the gain of visual response in relation to the retino-geniculo-cortical input" (P. 3, Lines 15-17).

- From page 8 onward, the word "*p-value*" is written in different styles (*italic*) and sometimes *P* in capital letters, be consistent throughout.

RESPONSE:

We thank the reviewer for pointing this out and have unified the formatting in the revised manuscript.

- Figure 3b-d: *Some of the effects look tiny. The traces of the off-on effects refer to a single hit rate for a single session of a single mouse? In the legend of this figure and the others where this applies, you might state on how many hit rates in total the ANOVAs are based on, and the effect size.*

RESPONSE:

Based on Reviewer #1's suggestion, we have replaced the previous Fig. 3 with the format in Fig. 4e,f. The detailed breakdown session by session in previous Fig. 3 is in Extended Data Fig. 6 now. In the Extended Data Fig. 6, the traces of the off-on effects refer to a single hit rate for a single session of a single mouse. In the current version of Fig. 3, we have done the ANOVA across mice:

1. A three-way repeated ANOVA was used to assess single and crossover effects for side (contralateral and ipsilateral), contrast (100% and 12.5%), and laser (on and off).
2. A two-way repeated ANOVA was used to assess single and crossover effects for contrast and laser on each side.
3. A one-way repeated ANOVA was used to assess the laser effects on each side at each contrast level.

We have added detailed information for the statistical analysis, including the F ratio, the associated degrees of freedom, the exact associated P value, and the partial η^2 for ANOVA in the figure legends and in Extended Data Table 2. The revised Fig.3 and the statistical results for data in revised Fig. 3 are as follows. To keep the figure legends concise, the partial η^2 reflecting the effect size is shown in Extended Data Table 2.

“Fig. 3 | Manipulation of ACA CPN activity has opposite effects on contralateral and ipsilateral performance. **a**, Schematic diagram of the viral strategy for optogenetic manipulation of CPNs. **b**, Schematic diagram of optogenetic inhibition and activation of CPNs with yellow and blue laser, respectively. **c**, Effect of unilateral CPN inhibition on visual change detection ($n = 14$ mice). Left, changes in contralateral performance. Right, changes in ipsilateral performance. Inhibition of CPNs (yellow laser, 589 nm, 3-8 mW, constant light) had similar effects on hit rates and FA rates at the 100% and 12.5% contrast levels (hit rates, $F_{\text{contrast}*\text{laser}}(1,13) = 0.02$, $P_{\text{contrast}*\text{laser}} = 0.90$; FA rates, $F_{\text{contrast}*\text{laser}}(1,13) = 0.03$, $P_{\text{contrast}*\text{laser}} = 0.87$; three-way repeated ANOVA), while having significantly different effects on contralateral vs. ipsilateral hit rates and FA rates (hit rates, $F_{\text{side}*\text{laser}}(1,13) = 53.96$, $P_{\text{side}*\text{laser}} = 6 \times 10^{-6}$; FA rates, $F_{\text{side}*\text{laser}}(1,13) = 22.24$, $P_{\text{side}*\text{laser}} = 4 \times 10^{-4}$; three-way repeated ANOVA). Inhibition of CPNs significantly impaired contralateral detection (decreased hit rates, green bars, $F_{\text{laser}}(1,13) = 30.04$, $P_{\text{laser}} = 1 \times 10^{-4}$; increased FA rates, yellow bars, $F_{\text{laser}}(1,13) = 9.85$, $P_{\text{laser}} = 0.008$; two-way repeated ANOVA) and improved ipsilateral detection (increased hit rates, green bars,

$F_{\text{laser}}(1,13) = 40.11$, $P_{\text{laser}} = 3 \times 10^{-5}$; decreased FA rates, yellow bars, $F_{\text{laser}}(1,13) = 8.27$, $P_{\text{laser}} = 0.01$; two-way repeated ANOVA). **d**, Effect of unilateral CPN inhibition on reaction times in hit trials. Inhibition of CPNs induced significantly different effects on contralateral and ipsilateral reaction time ($F_{\text{side}*\text{laser}}(1,13) = 25.60$, $P_{\text{side}*\text{laser}} = 2 \times 10^{-4}$, three-way repeated ANOVA), showing a significant increase in the contralateral reaction time and a significant decrease in the ipsilateral reaction time at the 12.5% contrast level (contralateral reaction time, $F_{\text{laser}}(1,13) = 7.17$, $P_{\text{laser}} = 0.02$; ipsilateral reaction time, $F_{\text{laser}}(1,13) = 17.255$, $P_{\text{laser}} = 0.001$; one-way repeated ANOVA). **e,f**, Similar to **c,d**, but for optogenetic activation of CPNs ($n = 9$ mice). **e**, Activation of CPNs (blue laser, 473 nm, 1-3 mW, 10 Hz, 5 ms per pulse) had similar effects on hit rates and FA rates at the 100% and 12.5% contrast levels (hit rates, $F_{\text{contrast}*\text{laser}}(1,8) = 2.38$, $P_{\text{contrast}*\text{laser}} = 0.16$; FA rates, $F_{\text{contrast}*\text{laser}}(1,8) = 2.32$, $P_{\text{contrast}*\text{laser}} = 0.17$; three-way repeated ANOVA), while having significantly different effects on contralateral vs. ipsilateral hit rates and FA rates (hit rates, $F_{\text{side}*\text{laser}}(1,8) = 27.88$, $P_{\text{side}*\text{laser}} = 0.001$; FA rates, $F_{\text{side}*\text{laser}}(1,8) = 18.31$, $P_{\text{side}*\text{laser}} = 0.003$; three-way repeated ANOVA). Activation of CPNs significantly improved contralateral detection (increased hit rates, $F_{\text{laser}}(1,8) = 18.07$, $P_{\text{laser}} = 0.003$; decreased FA rates, $F_{\text{laser}}(1,8) = 14.37$, $P_{\text{laser}} = 0.005$; two-way repeated ANOVA) and impaired ipsilateral detection (decreased hit rates, $F_{\text{laser}}(1,8) = 15.49$, $P_{\text{laser}} = 0.004$; increased FA rates, $F_{\text{laser}}(1,13) = 14.34$, $P_{\text{laser}} = 0.005$; two-way repeated ANOVA). **f**, Activation of CPNs induced significantly different effects on contralateral and ipsilateral reaction time ($F_{\text{side}*\text{laser}}(1,8) = 5.90$, $P_{\text{side}*\text{laser}} = 0.04$, three-way repeated ANOVA), showing a significant decrease in the contralateral reaction time and a significant increase in the ipsilateral reaction time at the 12.5% contrast level (contralateral reaction time, $F_{\text{laser}}(1,8) = 10.46$, $P_{\text{laser}} = 0.01$; ipsilateral reaction time, $F_{\text{laser}}(1,8) = 5.71$, $P_{\text{laser}} = 0.04$; one-way repeated ANOVA). All data are presented as the mean \pm SEM. Also see Extended Data Table 2 for detailed parameters of ANOVA.”

- Page 4, second line: “... to facilitate stereoscopic vision...” Do not confound binocular with stereoscopic vision. The cited paper 22 is not about the latter.

RESPONSE:

We thank the reviewer for pointing out this error; we have now replaced “stereoscopic” with the correct “binocular” in the revised manuscript. The full revised sentence reads as follows (P. 3, Lines 19-21):

“First, V1 callosal projections drive the developmental elimination

of specific cortical interneurons (chandelier cells) in the binocular zone to facilitate binocular vision (Wang et al., 2021).”

- Page 22: “..comprising..” reads “comprises”

RESPONSE:

We thank the reviewer for pointing out this error, which we have corrected as follows (P. 20, Lines 16-17):

“This transcallosal loop comprises CPNs and PV_{cal} neurons in the left and right ACA.”

- Page 24 and else: “...ameliorate the behavioral bias...” might be contextually wrong. “...reverse the inactivation/lesion-induced behavioral bias ?”

RESPONSE:

We thank the reviewer for pointing out this error. We have now replaced “ameliorate the behavioral bias” with “reverse the inactivation/lesion-induced behavioral bias” throughout the revised manuscript.

- Page 24: “...support that reestablishing balanced interhemispheric interaction...after lesion”. Cite Lomber and Payne, 1996, for a similar restoration of a behavioral unilateral visual neglect by experimental bilateral inactivation of extrastriate or subcortical areas in an awake animal.

RESPONSE:

We now cite this paper in Discussion section. The revised text reads as follows (P. 23, Lines 6-9):

“Previous lesion studies in felines and clinical observations on stroke patients have shown that a contralateral brain lesion on a previously spared hemisphere can, paradoxically, cancel the deficits in visuospatial functions generated by a first lesion, e.g., recovery from visuospatial neglect-an attentional deficit (Corbetta and Shulman, 2011; Lomber and Payne, 1996; Valero-Cabre et al., 2020).”

Methods

Page 11: state the direction of movement of the vertical ? grating. This might be relevant as rodents have directional biases in the visual system.

RESPONSE:

Thank you for bringing this omission to our notice. To be clear, this was a horizontal grating moving downward. We have updated the statement for this grating, which now reads as “orientation 0°, direction 90°”. The full revised text reads as follows:

“After a random delay interval of 2.3-2.8 s, the white noise was replaced by the Go cue (sinusoidal drifting grating, orientation 0°, direction 90°, 2 Hz, 0.04 cycles/°) on either the left or right screen.”

- Page 12: “.. expert..”?

RESPONSE:

We have replaced “... After mice expert in the learning phase1 (ACA-lesion group, ipsilesional hit rate >80%; other groups, hit rate on both sides >80%), we shortened the response window to 2 s and introduced high contrast no-go trials. ...” with “... After mice mastered the task in the learning phase1 (ACA-lesion group, ipsilesional hit rate >80%; other groups, hit rate on both sides >80%), we shortened the response window to 2 s and introduced high contrast no-go trials. ...”.

- Page 15: Arduino board

RESPONSE:

We thank the reviewer for pointing out this error, which we have corrected in the revised manuscript.

- Page 16: “through an implanted canular..”

RESPONSE:

We thank the reviewer for pointing out this error and replace “canular” with “cannula” in the revised manuscript.

References

- Bocci, T., Pietrasanta, M., Cerri, C., Restani, L., Caleo, M., and Sartucci, F. (2014). Visual callosal connections: role in visual processing in health and disease. *Rev. Neurosci.* *25*, 113-127.
- Corbetta, M., and Shulman, G.L. (2011). Spatial neglect and attention networks. *Annu. Rev. Neurosci.* *34*, 569-599.
- Desimone, R., and Duncan, J. (1995). Neural mechanisms of selective visual attention. *Annu. Rev. Neurosci.* *18*, 193-222.
- Fiebelkorn, I.C., and Kastner, S. (2020). Functional Specialization in the Attention Network. *Annu. Rev. Psychol.* *71*, 221-249.
- Glickfeld, L.L., and Olsen, S.R. (2017). Higher-Order Areas of the Mouse Visual Cortex. *Annu. Rev. Vis. Sci.* *3*, 251-273.
- Harris, J.A., Mihalas, S., Hirokawa, K.E., Whitesell, J.D., Choi, H., Bernard, A., Bohn, P., Caldejon, S., Casal, L., Cho, A., *et al.* (2019). Hierarchical organization of cortical and thalamic connectivity. *Nature* *575*, 195-202.
- Hu, F., Kamigaki, T., Zhang, Z., Zhang, S., Dan, U., and Dan, Y. (2019). Prefrontal Corticotectal Neurons Enhance Visual Processing through the Superior Colliculus and Pulvinar Thalamus. *Neuron* *104*, 1141-1152.
- Huda, R., Sipe, G.O., Breton-Provencher, V., Cruz, K.G., Pho, G.N., Adam, E., Gunter, L.M., Sullins, A., Wickersham, I.R., and Sur, M. (2020). Distinct prefrontal top-down circuits differentially modulate sensorimotor behavior. *Nat. Commun.* *11*, 6007.
- Innocenti, G.M., Schmidt, K., Milleret, C., Fabri, M., Knyazeva, M.G., Battaglia-Mayer, A., Aboitiz, F., Ptito, M., Caleo, M., Marzi, C.A., *et al.* (2022). The functional characterization of callosal connections. *Prog. Neurobiol.* *208*, 102186.
- Kim, J.H., Ma, D.H., Jung, E., Choi, I., and Lee, S.H. (2021). Gated feedforward inhibition in the frontal cortex releases goal-directed action. *Nat. Neurosci.* *24*, 1452-1464.
- Leinweber, M., Ward, D.R., Sobczak, J.M., Attinger, A., and Keller, G.B. (2017a). A Sensorimotor Circuit in Mouse Cortex for Visual Flow Predictions. *Neuron* *95*, 1420-1432.
- Leinweber, M., Ward, D.R., Sobczak, J.M., Attinger, A., and Keller, G.B. (2017b). A Sensorimotor Circuit in Mouse Cortex for Visual Flow Predictions. *Neuron* *96*, 1204.
- Licata, A.M., Kaufman, M.T., Raposo, D., Ryan, M.B., Sheppard, J.P., and Churchland, A.K. (2017). Posterior Parietal Cortex Guides Visual Decisions in Rats. *J. Neurosci.* *37*, 4954-4966.
- Lomber, S.G., and Payne, B.R. (1996). Removal of two halves restores the whole: reversal of visual hemineglect during bilateral cortical or collicular inactivation in the cat. *Vis. Neurosci.* *13*, 1143-1156.
- Ma, G. *et al.* Hierarchy in sensory processing reflected by innervation balance on cortical interneurons. *Sci. Adv.* *7*, abf5676 (2021).
- Makino, H., and Komiyama, T. (2015). Learning enhances the relative impact of top-down processing in the visual cortex. *Nat. Neurosci.* *18*, 1116-1122.
- Moore, T., and Zirnsak, M. (2017). Neural Mechanisms of Selective Visual Attention. *Annu. Rev. Psychol.* *68*, 47-72.
- Norman, K.J., Riceberg, J.S., Koike, H., Bateh, J., McCraney, S.E., Caro, K., Kato, D., Liang, A., Yamamuro, K., Flanigan, M.E., *et al.* (2021). Post-error recruitment of frontal sensory cortical projections promotes attention in mice. *Neuron* *109*, 1202-1213.
- Oh, S.W., Harris, J.A., Ng, L., Winslow, B., Cain, N., Mihalas, S., Wang, Q., Lau, C., Kuan, L., Henry,

A.M., *et al.* (2014). A mesoscale connectome of the mouse brain. *Nature* 508, 207-214.

Seabrook, T.A., Burbridge, T.J., Crair, M.C., and Huberman, A.D. (2017). Architecture, Function, and Assembly of the Mouse Visual System. *Annu. Rev. Neurosci.* 40, 499-538.

Song, Y.H., Kim, J.H., Jeong, H.W., Choi, I., Jeong, D., Kim, K., and Lee, S.H. (2017). A Neural Circuit for Auditory Dominance over Visual Perception. *Neuron* 93, 1236-1237.

Tehovnik, E.J., Froudarakis, E., Scala, F., Smirnakis, S.M., Patel, S.S., and Tolias, A.S. (2021). Visuomotor control in mice and primates. *Neurosci. Biobehav. Rev.* 130, 185-200.

Valero-Cabre, A., Toba, M.N., Hilgetag, C.C., and Rushmore, R.J. (2020). Perturbation-driven paradoxical facilitation of visuo-spatial function: Revisiting the 'Sprague effect'. *Cortex* 122, 10-39.

Vann, S.D., Aggleton, J.P., and Maguire, E.A. (2009). What does the retrosplenial cortex do? *Nat. Rev. Neurosci.* 10, 792-802.

Vormstein-Schneider, D., Lin, J.D., Pelkey, K.A., Chittajallu, R., Guo, B., Arias-Garcia, M.A., Allaway, K., Sakopoulos, S., Schneider, G., Stevenson, O., *et al.* (2020). Viral manipulation of functionally distinct interneurons in mice, non-human primates and humans. *Nat. Neurosci.* 23, 1629-1636.

Wang, B.S., Bernardez Sarria, M.S., An, X., He, M., Alam, N.M., Prusky, G.T., Crair, M.C., and Huang, Z.J. (2021). Retinal and Callosal Activity-Dependent Chandelier Cell Elimination Shapes Binocularity in Primary Visual Cortex. *Neuron* 109, 502-515.

Yao, S., Wang, Q., Hirokawa, K.E., Ouellette, B., Ahmed, R., Bomben, J., Brouner, K., Casal, L., Caldejon, S., Cho, A., *et al.* (2023). A whole-brain monosynaptic input connectome to neuron classes in mouse visual cortex. *Nat. Neurosci.* 26, 350-364.

Zhang, S., Xu, M., Chang, W.C., Ma, C., Hoang Do, J.P., Jeong, D., Lei, T., Fan, J.L., and Dan, Y. (2016). Organization of long-range inputs and outputs of frontal cortex for top-down control. *Nat. Neurosci.* 19, 1733-1742.

Zhang, S., Xu, M., Kamigaki, T., Hoang Do, J.P., Chang, W.C., Jenvay, S., Miyamichi, K., Luo, L., and Dan, Y. (2014). Selective attention. Long-range and local circuits for top-down modulation of visual cortex processing. *Science* 345, 660-665.

Zingg, B., Hintiryan, H., Gou, L., Song, M.Y., Bay, M., Bienkowski, M.S., Foster, N.N., Yamashita, S., Bowman, I., Toga, A.W., *et al.* (2014). Neural networks of the mouse neocortex. *Cell* 156, 1096-1111.

REVIEWERS' COMMENTS

Reviewer #1 (Remarks to the Author):

The paper has much improved. It was a strong paper to begin with and it is now even stronger.

It is nice that the paper now focuses on “interhemispheric communication” rather than “competition” (per the suggestion by Reviewer 3) as that interpretation of competition was at times strained. The abstract is now clearer; the introduction is more concise; the results section no longer contains introductory material. Also, it is good that the statistics no longer assume that measurements obtained in the same mouse are independent of each other. And it is great to see images showing where the measurements were made in the brain.

Finally, as far as I can tell, my remaining comments have been satisfactorily addressed. I say “as far as I can tell” because it was frankly not possible to go carefully through 60 pages of paper and 87 pages of rebuttal. This is too much to ask of a reviewer. If a reviewer writes “please explain X”, that’s a request for the revised paper to explain X, not a request for a lengthy explanation of X in a rebuttal. Indeed, a paper should speak for itself.

If there is a further round of review, please limit the rebuttal to a few sentences for each comment, and if relevant use them to point to changes made in the paper.

Further suggestions:

When describing Figs 1h-j in the main text, please clarify what is meant by False Alarm in this version of the task. Is it moving the wheel in any direction when the wheel should have been stationary? Or is it moving the wheel in the wrong direction? The text in page 9 line 4 seems to indicate the former but it would be good to be sure.

Reviewer Figure 1: this arguably belongs in the paper, as a supplementary figure. Or it can be easily summarized in words.

The fact that shutting down both sides of ACA does not affect behavior seems like a major issue. It indicates that ACA is not crucial to the behavior. To address this matter, the revision has added a paragraph in Discussion that mentions the Sprague effect. This is fine, but it does not make the effect less mysterious. It seems that ACA is not necessary for the task, but if only one side of it is present then it interferes with the task. This could be made clearer. It is certainly not the impression one gets from the beginning of the paper, when one thinks that ACA is crucial for the task.

When referring to optogenetic inactivation (e.g., Abstract, lines 7 and 13 and many other places in the text) it would be best to use words such as “inactivation” or “suppression” rather than “inhibition”, because the latter is a synaptic mechanism. Indeed, when referring to the opposite manipulation the

paper correctly says “activation” and not “excitation”.

Additional details:

Abstract, line 11: providing -> mediating

Abstract, line 15 (and in all other relevant places in text): lesion -> lesioned

Page 3 line 7. The sentence does not work. To make it work, replace “, with” with a colon.

Page 6 line 20. It would help to remove “with visual stimuli at 100% contrast” as this is confusing (one thinks it refers to the no-go trials).

Page 7 line 9. GCamp6s -> GCaMP6s

Page 14 line 11 “supporting that” -> “supporting the view that”. Similarly, page 21 line 15 “support that” -> “support the view that” (or “the notion that”). Similar suggestions apply to other places in the paper that say “support that” or “supporting that”.

Reviewer #2 (Remarks to the Author):

It remains my impression that the results are interesting and, in so much as they pertain to a functional role of local inhibition mediated by transcallosal projection, they may have a significance that extend beyond the specific experimental context in which they were obtained.

In my initial review I focussed primarily on conceptual issues, and on the extent to which key information has been communicated effectively. It is evident that the authors attempted to address these issues. Although I consider that there remains scope for further enhancement, to extend the appeal of the study to those who employ different experimental paradigms (and study other behavioural phenomenon), it is not my role as reviewer to insist on an emphasis that I favour.

Reviewer #3 (Remarks to the Author):

The authors have thoroughly addressed my concerns, responded the questions and changed the title of the manuscript as recommended.

Reviewer #1 (reviewer's comments in italic):

The paper has much improved. It was a strong paper to begin with and it is now even stronger.

It is nice that the paper now focuses on “interhemispheric communication” rather than “competition” (per the suggestion by Reviewer 3) as that interpretation of competition was at times strained. The abstract is now clearer; the introduction is more concise; the results section no longer contains introductory material. Also, it is good that the statistics no longer assume that measurements obtained in the same mouse are independent of each other. And it is great to see images showing where the measurements were made in the brain.

Finally, as far as I can tell, my remaining comments have been satisfactorily addressed. I say “as far as I can tell” because it was frankly not possible to go carefully through 60 pages of paper and 87 pages of rebuttal. This is too much to ask of a reviewer. If a reviewer writes “please explain X”, that’s a request for the revised paper to explain X, not a request for a lengthy explanation of X in a rebuttal. Indeed, a paper should speak for itself.

If there is a further round of review, please limit the rebuttal to a few sentences for each comment, and if relevant use them to point to changes made in the paper.

Further suggestions:

When describing Figs 1h-j in the main text, please clarify what is meant by False Alarm in this version of the task. Is it moving the wheel in any direction when the wheel should have been stationary? Or is it moving the wheel in the wrong direction? The text in page 9 line 4 seems to indicate the former but it would be good to be sure.

RESPONSE:

We thank the reviewer for this guidance and have made revisions accordingly. The revised content now reads as:

“In no-go trials, constant visual white noise was presented, and mice were punished with a timeout (2s) for turning the wheel in any direction (false alarm [FA]; clockwise turn, left-FA; counter-clockwise turn, right-FA).” (P. 6, Lines 8-10).

Reviewer Figure 1: this arguably belongs in the paper, as a supplementary figure. Or it can be easily summarized in words.

RESPONSE:

As the Reviewer Figure 1 used the same data as in Fig. 6f, we now summarize the information shown in Reviewer Figure 1 in the revised text, which reads as:

“Notably, the strength of callosal inputs was found to be comparable between SST+ and VIP+ neurons across different layers, but in each layer, PV+ neurons exhibited much stronger callosal inputs (greater than 2.8-fold) compared to SST+ and VIP+ neurons. (Fig. 6f).” (P. 15, Lines 1-4).

The fact that shutting down both sides of ACA does not affect behavior seems like a major issue. It indicates that ACA is not crucial to the behavior. To address this matter, the revision has added a paragraph in Discussion that mentions the Sprague effect. This is fine, but it does not make the effect less mysterious. It seems that ACA is not necessary for the task, but if only one side of it is present then it interferes with the task. This could be made clearer. It is certainly not the impression one gets from the beginning of the paper, when one thinks that ACA is crucial for the task.

RESPONSE:

We understand the reviewer’s concern regarding the role of ACA in our two-alternative unforced-choice (2AUC) change-detection task and the seemingly contradictory effects of unilateral and bilateral ACA inactivation on behavioral bias. As you noted in this comment, we have considered this apparent contradiction in our discussion, and we currently think the most plausible explanation lies in the interplay of multiple processes mediated by a set of distinct brain structures for visual processing across both hemispheres, in a scenario potentially similar to the Sprague effect.

For some additional context, previous studies have shown that ACA modulates different aspects of visually-guided goal-directed behavior in mice (Hu et al., 2019; Huda et al., 2020; Kim et al., 2021; Leinweber et al., 2017; Norman et al., 2021; Zhang et al., 2014). Activation of the ACA improves visual feature detection, a hallmark feature of visual attention (Hu et al., 2019; Leinweber et al., 2017; Norman et al., 2021; Zhang et al., 2014). Consistent with studies of attention modulation in primates (Desimone and Duncan, 1995; Fiebelkorn and Kastner, 2020; Moore and Zirnsak, 2017), our mouse results from the 2AUC task establish that activation of ACA CPNs i) enhances the detection of contralateral visual changes and ii) reduces the reaction time to such changes. These lines of evidence empirically support that the ACA does function in visual processing and suggest its involvement in visual attention.

While recalling our results showing that inactivating/lesioning the ACA on one side does exert a functional impact on task performance, the reviewer rightly notes that when both sides of the ACA are inactivated/lesioned, the deficit evident from a single lesion seems to be canceled, so the behavioral readout we can observe from the task appears to be unaffected.

Three points bear consideration together here:

- 1) Note that in our initial exploratory profiling of the medial and frontal

cortices in the mouse visual network using the 2AUC task, we did detect some extent of bias upon lesion of multiple cortical areas, including the ACA, the RSP, and the PTLp. That is, while the most obvious impact was detected upon ACA lesion, there was bias evident with the other examined higher-level cortical areas in the visual network.

- 2) Given the current understanding that multiple cortical and subcortical brain structures exhibit selective visual-related activity in visuomotor tasks (Peters et al., 2021; Steinmetz et al., 2019; Zatka-Haas et al., 2021), it is unlikely that a specific contribution of the ACA to visual processing would be solely causal for the observed behaviors.
- 3) The behaviors we are able to monitor with the 2AUC task must be conceptualized as higher-order readouts of the test animal's overall brain function; accordingly, it is plausible that some compensatory activity (*i.e.*, some form of rebalancing, as mediated by different structures or some combination of structures) could in theory be engaged upon bilateral ACA lesion that could explain the apparent canceling (*NOTA BENE*: assessed only as the final behavioral readout) of the bias that was uniquely evident upon unilateral lesion.

Thus, while our evidence does unequivocally support a function for the ACA in visual processing, our study does not exclude potential contributions from other structures in the overall sequence of visually-guided decision-making events leading to the behaviors. We have now added mention of both our RSP/PTLp profiling findings and of the potential for multi-structure impacts to our revised discussion section. The revised content now reads as (P. 23, Lines 19-22, and P. 24, Lines 1-3):

“Previous studies have shown that multiple cortical and subcortical brain structures exhibit task-related activity in visually-guided decision-making^{45,66,81}. In the current study, we also observed some extent of bias upon lesion of the RSP and PTLp. It is plausible that some compensatory activity (*i.e.*, some form of rebalancing, as mediated by different structures or some combination of structures) could in theory be engaged upon bilateral ACA lesion that could explain the apparent canceling (*NOTA BENE*: assessed only as the final behavioral readout) of the bias that was uniquely evident upon unilateral lesion.”

When referring to optogenetic inactivation (e.g., Abstract, lines 7 and 13 and many other places in the text) it would be best to use words such as “inactivation” or “suppression” rather than “inhibition”, because the latter is a synaptic mechanism. Indeed, when referring to the opposite manipulation the paper correctly says

“activation” and not “excitation”.

RESPONSE:

We thank the reviewer for pointing out these inconsistencies; we have now replaced “inhibition” with “inactivation” in the revision.

Additional details:

Abstract, line 11: providing -> mediating

RESPONSE:

Based on the reviewer’s guidance, this content now reads as:

“CPNs strongly activated contralateral parvalbumin-positive (PV+) neurons, and callosal-input-driven PV+ neurons preferentially inhibited ipsilateral CPNs, thus mediating transcallosal inhibition.”
(Abstract, Lines 9-11).

Abstract, line 15 (and in all other relevant places in text): lesion -> lesioned

RESPONSE:

Based on the reviewer’s guidance, we have now employed the properly inflected ‘lesioned’ diction throughout the text (see for examples: Abstract, Line 15; P. 6, Lines, 19 and 20; etc).

Page 3 line 7. The sentence does not work. To make it work, replace “, with” with a colon.

RESPONSE:

We thank the reviewer for pointing out this error. The full revised sentence reads as follows (P. 3, Lines 6-10):

“Clinical observations of patients with callosal degradation suggested distinctions between posterior and anterior regions: the posterior callosal areas linking sensory regions of the two hemispheres were related to interhemispheric facilitation of binocular vision, while anterior callosal regions linking frontal and parietal regions were related to maintaining balanced interhemispheric inhibition in visuospatial

processing.”

Page 6 line 20. It would help to remove “with visual stimuli at 100% contrast” as this is confusing (one thinks it refers to the no-go trials).

RESPONSE:

Based on the reviewer’s guidance, we have removed “with visual stimuli at 100% contrast”. The full revised sentence reads as follows (P. 7, Lines 3):

“After learning phase1, we introduced no-go trials to control mice and ACA-lesioned mice.”

Page 7 line 9. GCamp6s -> GCaMP6s

RESPONSE:

We thank the reviewer for pointing out this error. We have replaced “GCamp6s” with “GCaMP6s” throughout the text.

Page 14 line 11 “supporting that” -> “supporting the view that”. Similarly, page 21 line 15 “support that” -> “support the view that” (or “the notion that”). Similar suggestions apply to other places in the paper that say “support that” or “supporting that”.

RESPONSE:

Based on the reviewer’s guidance, we have replaced “supporting that” with “supporting the view that” in the revised text.

References

- Desimone, R., and Duncan, J. (1995). Neural mechanisms of selective visual attention. *Annu Rev Neurosci* *18*, 193-222.
- Fiebelkorn, I.C., and Kastner, S. (2020). Functional Specialization in the Attention Network. *Annu Rev Psychol* *71*, 221-249.
- Hu, F., Kamigaki, T., Zhang, Z., Zhang, S., Dan, U., and Dan, Y. (2019). Prefrontal Corticotectal Neurons Enhance Visual Processing through the Superior Colliculus and Pulvinar Thalamus. *Neuron* *104*, 1141-1152 e1144.
- Huda, R., Sipe, G.O., Breton-Provencher, V., Cruz, K.G., Pho, G.N., Adam, E., Gunter, L.M., Sullins, A., Wickersham, I.R., and Sur, M. (2020). Distinct prefrontal top-down circuits differentially modulate sensorimotor behavior. *Nat Commun* *11*, 6007.
- Kim, J.H., Ma, D.H., Jung, E., Choi, I., and Lee, S.H. (2021). Gated feedforward inhibition in the frontal cortex releases goal-directed action. *Nat Neurosci* *24*, 1452-1464.
- Leinweber, M., Ward, D.R., Sobczak, J.M., Attinger, A., and Keller, G.B. (2017). A Sensorimotor Circuit in Mouse Cortex for Visual Flow Predictions. *Neuron* *96*, 1204.
- Moore, T., and Zirnsak, M. (2017). Neural Mechanisms of Selective Visual Attention. *Annu Rev Psychol* *68*, 47-72.
- Norman, K.J., Riceberg, J.S., Koike, H., Bateh, J., McCraney, S.E., Caro, K., Kato, D., Liang, A., Yamamuro, K., Flanigan, M.E., *et al.* (2021). Post-error recruitment of frontal sensory cortical projections promotes attention in mice. *Neuron* *109*, 1202-1213 e1205.
- Peters, A.J., Fabre, J.M.J., Steinmetz, N.A., Harris, K.D., and Carandini, M. (2021). Striatal activity topographically reflects cortical activity. *Nature* *591*, 420-425.
- Steinmetz, N.A., Zatka-Haas, P., Carandini, M., and Harris, K.D. (2019). Distributed coding of choice, action and engagement across the mouse brain. *Nature* *576*, 266-273.
- Zatka-Haas, P., Steinmetz, N.A., Carandini, M., and Harris, K.D. (2021). Sensory coding and the causal impact of mouse cortex in a visual decision. *Elife* *10*.
- Zhang, S., Xu, M., Kamigaki, T., Hoang Do, J.P., Chang, W.C., Jenvay, S., Miyamichi, K., Luo, L., and Dan, Y. (2014). Selective attention. Long-range and local circuits for top-down modulation of visual cortex processing. *Science* *345*, 660-665.

Reviewer #2 (Remarks to the Author):

It remains my impression that the results are interesting and, in so much as they pertain to a functional role of local inhibition mediated by transcallosal projection, they may have a significance that extend beyond the specific experimental context in which they were obtained.

In my initial review I focussed primarily on conceptual issues, and on the extent to which key information has been communicated effectively. It is evident that the authors attempted to address these issues. Although I consider that there remains scope for further enhancement, to extend the appeal of the study to those who employ different experimental paradigms (and study other behavioural phenomenon), it is not my role as reviewer to insist on an emphasis that I favour.

Reviewer #3 (Remarks to the Author):

The authors have thoroughly addressed my concerns, responded the questions and changed the title of the manuscript as recommended.